# How to Guide Your Flow:
# Few-Step Alignment via Flow Map Reward Guidance

**Jerry Y. Huang** [* 1]   **Justin Lin** [* 1]   **Sheel Shah** [1]   **Kartik Nair** [1]   **Nicholas M. Boffi** [1]

## Abstract

In generative modeling, we often wish to produce samples that maximize a user-specified reward such as aesthetic quality or alignment with human preferences, a problem known as *guidance*. Despite their widespread use, existing guidance methods either require expensive multi-particle, many-step schemes or rely on poorly understood approximations. We reformulate guidance as a *deterministic optimal control problem*, yielding a hierarchy of algorithms that subsumes existing approaches at the coarsest level. We show that the *flow map*, an object of significant recent interest for its role in fast inference, arises naturally in the optimal solution. Based on this observation, we propose **Flow Map Reward Guidance (FMRG)**: a training-free, *single-trajectory* framework that uses the flow map to both integrate and guide the flow. At text-to-image scale, FMRG matches or surpasses baselines across inverse problems and reward-guided generation with **as few as 3 NFEs**, giving at least an order-of-magnitude speedup in comparison to prior state of the art.

## 1. Introduction

Flow- and diffusion-based (Lipman et al., 2022; Albergo et al., 2023; Song et al., 2021) generative models have emerged as state-of-the-art methods for high-fidelity generation across continuous modalities such as images (Rombach et al., 2022), video (Blattmann et al., 2023), and molecular data (Watson et al., 2023). In practice, however, we rarely seek arbitrary samples from a learned distribution. Instead, applications in creative generation, molecular design, and inverse problems require samples that satisfy additional criteria, such as high aesthetic quality (Clark et al., 2024), measurement agreement (Daras et al., 2024), physical plau-

sibility (Wu et al., 2023a), or alignment with human user intent. These criteria are often encoded as a hand-designed or learned reward function $r$, and *guidance* methods aim to steer the sampling process toward high-reward samples without additional training (Singhal et al., 2025).

The prevailing theoretical framework for guidance frames the problem as sampling from a *reward-tilted* distribution $\tilde{\rho}(x) \propto e^{r(x)}\rho(x)$, where $\rho$ denotes the distribution induced by a pre-trained generative model (Domingo-Enrich et al., 2025; Sabour et al., 2025a; Uehara et al., 2025). Despite its mathematical appeal, this target is remarkably difficult to sample from, and recent work has shown that doing so is computationally intractable even for simple reward functions (Moitra et al., 2026). Multi-particle methods such as sequential Monte Carlo (SMC) can in principle sample from $\tilde{\rho}$, but generically require a prohibitive number of particles and integration steps (Del Moral et al., 2006). To get around this difficulty, widely-used methods such as diffusion posterior sampling (DPS) (Chung et al., 2024) sidestep the ensemble by using a single particle, and additionally leverage heuristic approximations to the tilted score. However, their reliance on such approximations leaves the distribution they actually produce poorly characterized. In particular, it is unclear what object DPS actually approximates, or whether this object even relates to the reward tilt.

Compounding these difficulties, the reward-tilt formulation does not sit naturally with modern deterministic samplers. Its theoretical underpinnings lie in stochastic processes, where reward alignment is typically achieved by augmenting a noising SDE with a score term that transports samples toward $\tilde{\rho}$ (Domingo-Enrich et al., 2025; Uehara et al., 2024b). Modern systems such as Stable Diffusion 3 (Esser et al., 2024) and FLUX (Labs et al., 2025), however, increasingly favor *deterministic* probability flows, which recent work on flow maps (Boffi et al., 2025b;a; Song et al., 2023; Geng et al., 2025a; Kim et al., 2024) further accelerate to just a few sampling steps. Recent efforts have ported tilt-based alignment to this regime by simulating stochastic transitions within an ODE sampler (Holderrieth et al., 2025), but such approaches inherit the structure of the stochastic formulation rather than treating the deterministic sampler as the primary object. As a result, a principled guidance methodology na-

---
[*]Equal contribution   [1]Carnegie Mellon University, Pittsburgh, PA, USA. Correspondence to: Jerry Y. Huang <jerry-hua@andrew.cmu.edu>.

*Proceedings of the 43rd International Conference on Machine Learning*, Seoul, South Korea. PMLR 306, 2026. Copyright 2026 by the author(s).

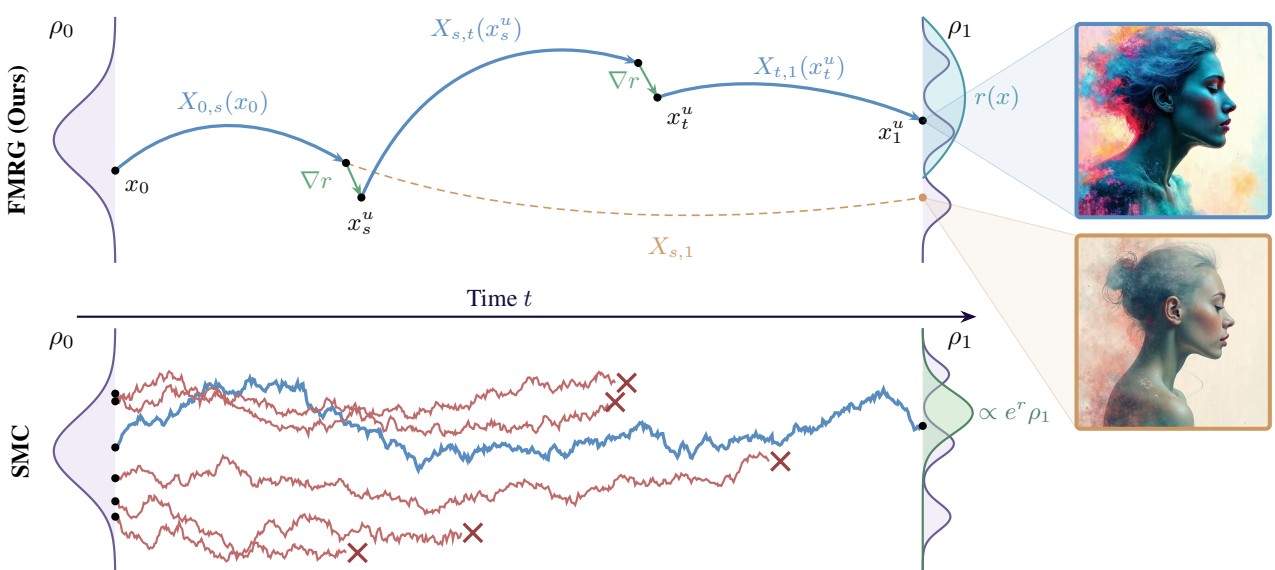

*Figure 1.* **Overview.** FMRG guides a *single* generative trajectory by alternating flow map steps, which integrate the base dynamics exactly, with gradient steps that steer toward high reward. This *optimization-centric* perspective contrasts with methods that explicitly target *sampling* the exponential reward tilt $\tilde{\rho} \propto e^r \rho$, which typically require many particles with resampling (e.g., SMC) and are often based on diffusion processes requiring many steps. Right: an unguided FLUX sample (orange) and a guided FMRG sample (blue) under a human-preference aesthetic reward.

tive to deterministic, few-step, single-trajectory generation does not yet exist. This motivates the central question:

> *Is there a principled framework for guiding*
> *flow-based generative models*
> *with very few function evaluations?*

Here we answer in the affirmative, proposing that the natural tool to formalize such a framework is *deterministic* optimal control. This runs contrary to the typical *stochastic* optimal control framework used for guidance and finetuning of flows and diffusions, and in particular it strays entirely from the reward tilt formulation of guidance (Domingo-Enrich et al., 2025; Uehara et al., 2024b;a). We show that this departure yields a natural optimization problem whose critical points we characterize in closed form, and which depend only on the *present* trajectory rather than an intractable average over an ensemble of additional particles. Crucially, the flow map appears explicitly in the optimal feedback, making it the central mathematical object of the solution. Its compositional structure enables a single pre-trained flow map to both integrate the base dynamics and compute an accurate guidance signal along one trajectory, leading to an extremely efficient guidance algorithm. Analyzing the design space of this approach leads to our *Flow Map Reward Guidance (FMRG)* framework, which to our knowledge is the first guidance methodology native to the few-step, single-particle regime.

Together, our **main contributions** can be summarized as:

- We propose an alternative perspective to reward tilting,

formulating guidance as a deterministic optimal control problem and showing that the *flow map* emerges naturally in the closed-form solution. This yields a tractable single-trajectory guidance algorithm in which a single pre-trained flow map drives both integration and guidance.

- We characterize the behavior of FMRG in an analytically-tractable setting and analyze the role of algorithmic interventions such as early stopping and the choice of reward gradient. This yields a unifying framework that subsumes existing single-trajectory methods such as DPS, as well as seed-optimization methods, as special cases (Appendix D).

- We demonstrate FMRG enables highly efficient, training-free guidance on FLUX-scale text-to-image models, matching or surpassing baselines across a wide range of latent-space inverse problems and reward-guided image generation settings with **as few as** 3 **NFEs**. This represents at least an order of magnitude speedup over competing approaches, and in some cases is up to $70\times$ more efficient.

## 2. Theoretical Framework

### 2.1. Flow-Based Generative Models

We consider a pre-trained flow-based generative model specified by a velocity field $b : [0, 1] \times \mathbb{R}^d \to \mathbb{R}^d$. Given a target distribution $\rho_1 \in \mathcal{P}(\mathbb{R}^d)$ and a Gaussian base $\rho_0 = \mathsf{N}(0, I)$,

samples from $\rho_1$ are drawn by numerically integrating the probability flow ordinary differential equation

$$\dot{x}_t = b_t(x_t), \qquad x_0 \sim \rho_0, \tag{1}$$

from $t = 0$ to $t = 1$, at which point $x_1 \sim \rho_1$. The velocity $b_t$ is typically learned from a dataset $\{x_1^i\}_{i=1}^n$ of target samples via flow matching or equivalent objectives based on stochastic interpolants $I_t = \alpha_t x_0 + \beta_t x_1$, with $\alpha_0 = \beta_1 = 1$ and $\alpha_1 = \beta_0 = 0$ (Lipman et al., 2022; Liu et al., 2022a; Albergo et al., 2023).

**Flow maps.** While generative models based on the probability flow (1) perform well in practice, inference requires solving the differential equation numerically, which typically necessitates 50–100 network evaluations. To accelerate inference, recent effort has centered around learning the *flow map* $X : [0,1]^2 \times \mathbb{R}^d \to \mathbb{R}^d$ (Boffi et al., 2025a;b), which is the solution operator of the probability flow (1). The flow map satisfies the *jump condition*

$$X_{s,t}(x_s) = x_t, \tag{2}$$

so that sampling only requires a single function evaluation via $x_1 = X_{0,1}(x_0)$. Further detail on the flow map formulation is provided in Appendix A, which includes many methods for accelerated generative modeling as special cases, including consistency models (Kim et al., 2024; Geng et al., 2024; Song et al., 2023; Li & He, 2024; Lu & Song, 2024), mean flows (Geng et al., 2025a;b), shortcut models (Frans et al., 2024), and terminal velocity matching (Zhou et al., 2025).

## 2.2. The Guidance Problem

Given a pre-trained flow-based model $b_t$ and a reward function $r : \mathbb{R}^d \to \mathbb{R}$, the goal of guidance is to modify the sampling process (1) to produce samples with high reward that "remain close" to the learned distribution $\rho_1$ (Uehara et al., 2025). In contrast to finetuning methods (Uehara et al., 2024b), which seek a similar goal via an additional training phase, guidance takes place purely at inference and does not require any additional training.

**Reward tilting.** The standard theoretical framework targets the *reward-tilted* distribution $\tilde{\rho}_1(x) \propto e^{r(x)} \rho_1(x)$, sampled by augmenting the velocity field with the value-function gradient

$$\nabla_x U(x) = \nabla_x \log \mathbb{E}\left[e^{r(x_1)} \mid x_t = x\right], \tag{3}$$

which is intractable in high dimensions and motivates SMC-based methods that maintain finite particle populations and perform resampling (Domingo-Enrich, 2024; Wu et al., 2023a). While principled, such methods sacrifice the efficiency of single-trajectory inference; moreover, while finite

sample estimators of the gradient (3) are *consistent*, they are biased away from the population limit, and can be high variance due to rare samples with high reward (Chetrite & Touchette, 2015). See Appendix B.

## 2.3. An Optimal Control Formulation

If we commit to guiding a *single deterministic trajectory*, what is the right problem to solve? We propose to depart from reward tilting entirely and instead formulate guidance as a *deterministic optimal control* problem that trades off reward maximization against deviation from the base flow:

$$\min_u \mathcal{L}(u) = \min_u \int_0^1 \frac{\|u_t\|^2}{2\lambda} dt - r(x_1^u),$$
$$\text{s.t. } \dot{x}_t^u = b_t(x_t^u) + u_t, \quad x_0 \text{ given.} \tag{4}$$

Above, $u : [0,1] \to \mathbb{R}^d$ is a control that modifies the base dynamics $b_t$, and $\lambda \in \mathbb{R}_{>0}$ is an inverse temperature that weights reward maximization against control effort. Larger $\lambda$ permits more aggressive guidance at the cost of departing further from the base flow, while smaller $\lambda$ keeps samples close to $\rho_1$.

In general, characterizing the optimal control for (4) in closed-form is challenging without restrictive assumptions on the drift $b_t$ and the reward $r$ (Fleming & Rishel, 1975). To gain insight into its optimizers, the following result gives the necessary conditions for an extreme point.

**Proposition 2.1.** *Let $u_t^*$ be an optimal solution of (4), and let $x_t^*$ denote its trajectory. Then, along $x_t^*$,*

$$u_t^* = \lambda \nabla X_{t,1}^{u^*}(x_t^*)^\mathsf{T} \nabla r\left(X_{t,1}^{u^*}(x_t^*)\right), \tag{5}$$

*where $X_{s,t}^{u^*}$ is the flow map for the optimally controlled dynamics $\dot{x}_t^{u^*} = b_t(x_t^{u^*}) + u_t^*$.*

The proof is given in Appendix C.2, and proceeds via a standard application of the Pontryagin Maximum Principle. The guidance term (5) is circular, requiring access to the optimally-guided flow map $X_{s,t}^{u^*}$ to compute the endpoint $x_1^* = X_{t,1}^{u^*}(x_t^*)$. Unfortunately, this endpoint is precisely what we are trying to compute during inference, making direct application of (5) infeasible in practice.

## 2.4. A Tractable Approximation

To break this circularity, we study the control problem (4) in the small-$\lambda$ limit. This regime is both analytically tractable and practically meaningful, as it forces the controlled trajectory to stay close to the base flow. Moreover, large $\lambda$ values can be counterproductive, promoting reward hacking and mode collapse.

To carry out this analysis, we first introduce the *optimal*

*value function,*

$$V_t^\lambda(x) = \min_u \left\{ \int_t^1 \frac{\|u_\tau\|^2}{2\lambda} d\tau - r(x_1^u) \;\Big|\; x_t^u = x \right\}, \quad (6)$$

which gives the optimal cost starting from state $x$ at time $t$. The optimal feedback control law has the form

$$u_t^*(x) = -\lambda \nabla V_t^\lambda(x), \quad (7)$$

where the optimal value function solves a Hamilton-Jacobi-Bellman (HJB) equation (Bardi & Capuzzo-Dolcetta, 1997). In the limit $\lambda \to 0$, we can solve this equation exactly, yielding a tractable guidance signal given only a pre-trained flow map.

**Proposition 2.2** (Small-$\lambda$ expansion). *Under standard regularity assumptions, the following properties hold:*

*(i) The $\lambda \to 0$ solution is $V_t^0(x) = -r(X_{t,1}(x))$, where $X_{t,1}$ is the uncontrolled flow map.*

*(ii) The corresponding guidance*

$$u_t^J(x) = \lambda \nabla X_{t,1}(x)^\mathsf{T} \nabla r(X_{t,1}(x)), \quad (8)$$

*approximates the optimal feedback to second order:*

$$\|u_t^*(x) - u_t^J(x)\| = O(\lambda^2). \quad (9)$$

The proof, given more formally in Appendix C.3, shows that the HJB equation reduces to a transport equation as $\lambda \to 0$, which we solve exactly via the method of characteristics. In practice, we may wish to use the guidance signal (8) away from the small $\lambda$ limit. We turn to this extension now.

**A greedy perspective.** We now show that (8) also admits a complementary interpretation as a *greedy* approximation of the original optimal control problem for *any* choice of $\lambda$. Specifically, for each $t$ we consider guidance signals that are applied in a small window around the present time,

$$\mathcal{U}_t^{\delta t} = \{u : [0,1] \to \mathbb{R}^d \mid u_\tau = 0 \,\forall\, \tau \notin [t, t+\delta t]\}. \quad (10)$$

The following result shows that (8) is optimal within this restricted class.

**Proposition 2.3.** *Consider the restricted problem*

$$\min_{u \in \mathcal{U}_t^{\delta t}} \mathcal{L}(u) = \min_{u \in \mathcal{U}_t^{\delta t}} \int_0^1 \frac{\|u_\tau\|^2}{2\lambda} d\tau - r(x_1^u). \quad (11)$$

*Then, the optimal guidance signal is given by (8) as $\delta t \to 0$.*

The proof is given in Appendix C.4, and proceeds by Taylor expansion in the width of the interval. Proposition 2.2 and Proposition 2.3 thus arrive at the same guidance signal (8) via complementary arguments: it is the globally optimal control for small $\lambda$, as well as the optimal greedy correction for any $\lambda$. Together, they justify using (8) in practice beyond the small-$\lambda$ regime.

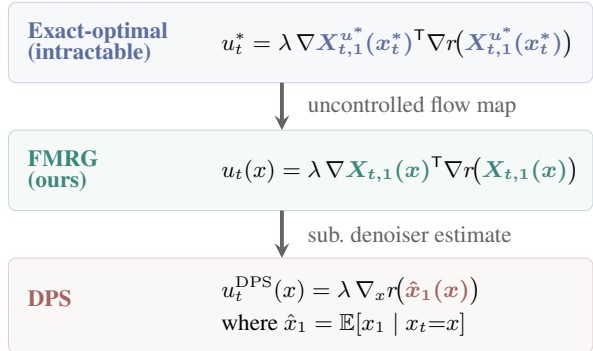

*Figure 2.* **Hierarchy of approximations.** The exact-optimal control requires the controlled flow map $X_{t,1}^{u^*}$. Our approaches leverages the uncontrolled flow map $X_{t,1}$, while DPS further approximates $X_{t,1}$ with a single Euler step.

**A hierarchy of approximations.** An immediate implication of the above results is that standard guidance schemes like DPS (Chung et al., 2024) can be more naturally understood as a single-step approximation of the greedy guidance (8), rather than as approximations of the reward-tilt gradient (3) (Moitra et al., 2026), which we formalize as follows. DPS is recovered from the greedy guidance (8) by replacing the exact flow-map endpoint $X_{t,1}(x)$ with the one-step Euler approximation $\hat{x}_1(x) := x + (1-t)\, b_t(x)$ and correspondingly approximating the Jacobian $\nabla X_{t,1}(x)^\mathsf{T}$ by $\left(I + (1-t)\,\nabla b_t(x)\right)^\mathsf{T}$ (Corollary D.1; see Appendix D.1 for the formal statement). For the linear interpolant, the posterior mean $\hat{x}_1$ coincides with a single Euler step of the probability flow, while the exact flow map $X_{t,1}$ corresponds to the limit of an infinite number of Euler steps; Corollary D.1 thus places DPS at the coarsest level of an approximation hierarchy (Figure 2). More broadly, many existing guidance methods, including FlowDPS (Kim et al., 2025), FlowChef (Patel et al., 2025), and MPGD (He et al., 2023), fall within this hierarchy (Table 3), with each method differing primarily in the choice of guidance weight. Similarly, seed-optimization methods such as ReNO (Eyring et al., 2024) and D-Flow (Ben-Hamu et al., 2024) can be viewed as restricting the greedy guidance to $t = 0$ (see Section 3 for additional details). Appendix D contains a detailed derivation of each reduction.

## 2.5. On-Manifold Guidance

Beyond its appearance in the optimal feedback (8), the Jacobian $\nabla X_{t,1}(x_t)^\mathsf{T}$ has a second, geometric role: it projects reward gradients onto the tangent space of the data manifold.

**Proposition 2.4** (Tangent space projection). *Suppose the data distribution $\rho_1$ is supported on a smooth manifold $\mathcal{M} \subset \mathbb{R}^d$. Let $x_1 = X_{t,1}(x_t) \in \mathcal{M}$ be the endpoint of the flow, and let $T_{x_1}\mathcal{M}$ denote the tangent space at $x_1$. Then,*

*for any $v \in \mathbb{R}^d$ with $v = v_\parallel + v_\perp$ where $v_\parallel \in T_{x_1}\mathcal{M}$ and $v_\perp \perp T_{x_1}\mathcal{M}$,*

$$\nabla X_{t,1}(x_t)^{\mathsf{T}} v = \nabla X_{t,1}(x_t)^{\mathsf{T}} v_\parallel. \qquad (12)$$

The proof is given in Appendix C.7. Proposition 2.4 shows that the Jacobian annihilates components of the reward gradient orthogonal to the data manifold, keeping the controlled trajectory on-manifold by construction (Figure 4, left). While exact manifold structure is an idealization, the intuition extends to settings where data concentrates *near* a low-dimensional structure, as widely believed for natural images: the Jacobian has small singular values in directions orthogonal to the effective data support. This geometric structure underlies the FMRG-J variant of our algorithm (Section 3), which uses the full Jacobian-projected guidance signal, in contrast to the cheaper FMRG-E variant, which drops the Jacobian for memory efficiency.

### 2.6. Terminal Distribution and Early Stopping

Given the characterization in Proposition 2.3, we now ask how this greedy approximation affects the terminal distribution obtained by evolving an ensemble of initial conditions. We address this in the Gaussian setting with a quadratic reward, where we find that all three guidance schemes (3), (5), and (8) become amenable to analytical treatment. While idealized, the resulting calculation allows us to understand the effect of each scheme on the diversity of samples and their rewards.

**Proposition 2.5.** *Consider a Gaussian base distribution $\rho_0 = \mathsf{N}(0, I)$, a Gaussian target $\rho_1 = \mathsf{N}(\mu_1, \sigma_1^2 I)$, and a quadratic reward $r(x) = -\|x - a\|^2$. Then the output distribution in each case is Gaussian with variances*

$$\sigma_{\text{tilt}}^2 = \frac{\sigma_1^2}{1 + 2\lambda\sigma_1^2},$$

$$\sigma_{\text{greedy}}^2 = \sigma_1^2 \exp(-2\pi\lambda\sigma_1), \qquad (13)$$

$$\sigma_{\text{exact}}^2 = \frac{\sigma_1^2}{(1 + \pi\lambda\sigma_1)^2},$$

*and expected terminal rewards $\bar{r}_s := \mathbb{E}[r(X_1^s)]$ given by*

$$\bar{r}_{\text{tilt}} = -\frac{\sigma_1^2(1 + 2\lambda\sigma_1^2) + (\mu_1 - a)^2}{(1 + 2\lambda\sigma_1^2)^2},$$

$$\bar{r}_{\text{greedy}} = -\left(\sigma_1^2 + (\mu_1 - a)^2\right) e^{-2\pi\lambda\sigma_1}, \qquad (14)$$

$$\bar{r}_{\text{exact}} = -\frac{\sigma_1^2 + (\mu_1 - a)^2}{(1 + \pi\lambda\sigma_1)^2}.$$

The proof is given in Appendix C.5. Inspecting (14), greedy guidance achieves the highest expected reward for a given $\lambda$, followed by exact optimal control and reward tilting. The three schemes exhibit qualitatively different variance scaling: reward tilting reduces variance as $O(1/(1 + c\lambda))$, exact

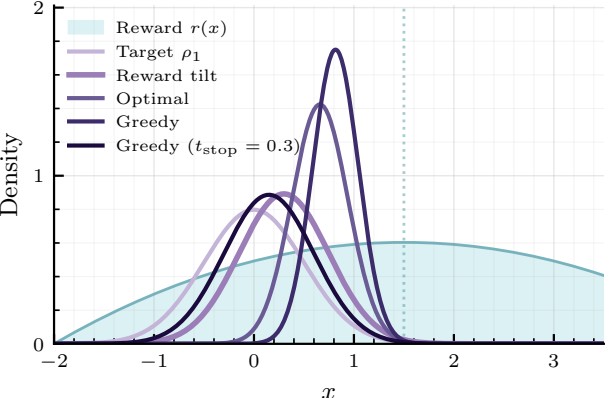

*Figure 3.* **Terminal distribution.** Greedy guidance produces a narrower distribution than reward tilting or the distribution produced by exactly solving the optimal control problem (5). Early stopping can be used to effectively mitigate this mode collapse, and when applied at $t_{\text{stop}} = 0.3$ recovers variance comparable to the reward tilt.

optimal control as $O\left(1/(1 + c\lambda)^2\right)$, and greedy guidance exponentially (Figure 3). This more aggressive contraction arises because greedy guidance neglects dependencies across time, yielding both higher expected reward and lower diversity for a given $\lambda$. In our image generation experiments, we find that this enables effective reward alignment at extremely low NFEs, though it necessitates careful control of the guidance strength to avoid mode collapse.

One effective way to control mode collapse is early stopping, in which we apply guidance only for $t \in [0, t_{\text{stop}}]$ and then integrate the uncontrolled flow to $t = 1$ via $X_{t_{\text{stop}},1}$. Here $t_{\text{stop}} \in (0, 1]$ controls how much of the trajectory is guided: smaller values stop guidance earlier (when the sample is still mostly noise), while $t_{\text{stop}} = 1$ corresponds to guiding the entire trajectory. In the Gaussian setting, we prove analytically that a suitable $t_{\text{stop}}(\lambda)$ recovers the polynomial scaling of exact optimal control. In the Gaussian setting, a unique $t_{\text{stop}}(\lambda) \in (0, 1]$ exists for which the terminal variance under greedy guidance on $[0, t_{\text{stop}}(\lambda)]$ followed by the uncontrolled flow on $[t_{\text{stop}}(\lambda), 1]$ matches $\sigma_{\text{exact}}^2$ (Proposition C.14); see Appendix C.5 for the formal statement, closed-form variance, and asymptotic behavior of $t_{\text{stop}}(\lambda)$ (Figure 3). We find empirically that these observations hold for high-resolution image synthesis at FLUX scale (Section 5; see Appendices E.4 and E.6 for ablations).

## 3. Algorithmic Framework

We now turn to a practical implementation of the guidance framework developed in Section 2. Substituting the greedy guidance term (8) into the controlled dynamics (4) gives the guided generative process

**Algorithm 1** Flow Map Reward Guidance (FMRG)

**Require:** Flow map $X$, reward $r$, time grid $\{t_k\}_{k=0}^N$, guidance strength $\lambda_t(x)$, gradient steps $n_{\text{opt}}$, gradient type $\in \{\text{Jacobian}, \text{Euclidean}\}$
1: Sample $x_0 \sim \rho_0$
2: **for** $k = 0, \ldots, N-1$ **do**
3:    $x \leftarrow X_{t_k, t_{k+1}}(x_{t_k})$ {Flow map step}
4:    **for** $j = 0, \ldots, n_{\text{opt}} - 1$ **do**
5:      **if** gradient type = Jacobian **then**
6:        $u \leftarrow \nabla X_{t_{k+1}, 1}(x)^\mathsf{T} \nabla r(X_{t_{k+1}, 1}(x))$
7:      **else**
8:        $u \leftarrow \nabla r(X_{t_{k+1}, 1}(x))$
9:      **end if**
10:     $x \leftarrow x + \frac{\Delta t_k}{n_{\text{opt}}} \lambda_{t_{k+1}}(x) \cdot u$ {Gradient step}
11:    **end for**
12:    $x_{t_{k+1}} \leftarrow x$
13: **end for**
14: **return** $x_1$

$$\dot{x}_t^u = b_t(x_t^u) + \lambda_t(x_t^u)\, \nabla X_{t,1}(x_t^u)^\mathsf{T} \nabla r(X_{t,1}(x_t^u)), \tag{15}$$

which forms the basis of our *Flow Map Reward Guidance (FMRG)* methodology. Implementing (15) effectively admits a design space of algorithmic decisions; in this section, we outline the conceptual role of each choice and make practical recommendations for high performance. Our complete procedure is summarized in Algorithm 1.

**Operator splitting.** For efficient inference, we propose an *operator splitting* (Strang, 1968) scheme to decompose (15) into the base flow $b_t(x_t)$, which we integrate exactly with the pre-trained flow map $X_{s,t}$, and the guidance term. Given a temporal grid $0 = t_0 < t_1 < \ldots < t_N = 1$ with $\Delta t_k = t_{k+1} - t_k$, each step proceeds as

$$\begin{aligned} \tilde{x}_{t_{k+1}} &= X_{t_k, t_{k+1}}(x_{t_k}), \\ x_{t_{k+1}} &= \tilde{x}_{t_{k+1}} + \Delta t_k \lambda_{t_{k+1}}(\tilde{x}_{t_{k+1}}) u_{t_{k+1}}(\tilde{x}_{t_{k+1}}), \end{aligned} \tag{16}$$

where $u$ denotes the guidance signal.

**Multiple gradient steps.** Within each time interval $[t_k, t_{k+1}]$, we may take multiple gradient steps to better optimize the reward before advancing the flow. To this end, let $n_{\text{opt}}$ denote the number of gradient steps per interval. The gradient part of the guidance update then becomes:

$$\begin{aligned} x^{(j+1)} &= x^{(j)} + \frac{\Delta t_k}{n_{\text{opt}}} \lambda_{t_{k+1}}(x^{(j)}) \cdot u_{t_{k+1}}(x^{(j)}), \\ &\quad j = 0, \ldots, n_{\text{opt}} - 1, \end{aligned} \tag{17}$$

where $x^{(0)} = \tilde{x}_{t_{k+1}}$ and $x_{t_{k+1}} = x^{(n_{\text{opt}})}$. These optimization steps may also be run before the first step in (16) for an

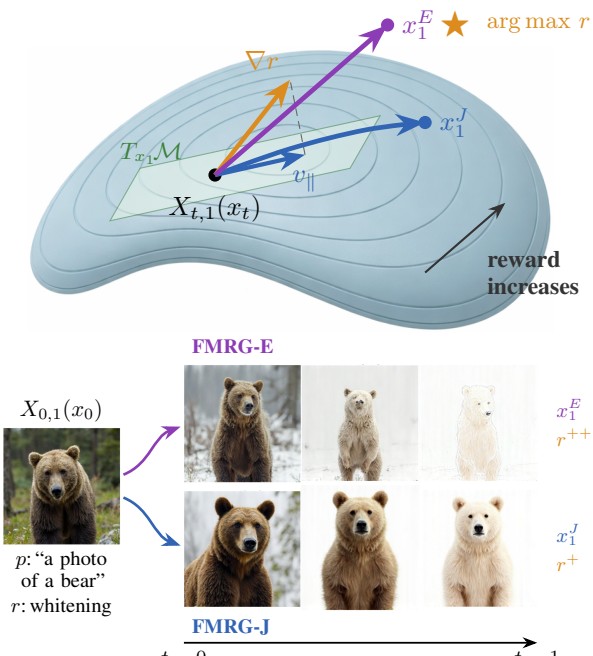

*Figure 4.* **Gradient options.** (Top) The flow map Jacobian $\nabla X_{t,1}(x)^\mathsf{T}$ projects the reward gradient $\nabla r$ onto $T_{x_1}\mathcal{M}$, keeping the trajectory on-manifold (blue, FMRG-J), while the Euclidean gradient follows $\nabla r$ off-manifold (purple, FMRG-E). (Bottom) FMRG-E achieves higher reward ($r$++) but produces artifacts because it can leave the data manifold, often leading to reward hacking; FMRG-J stays on-manifold and more robustly preserves features in the data ($r$+).

initial round of seed optimization (Eyring et al., 2024).

**Gradient options.** For the guidance signal $u$, we consider two primary choices. The first is the gradient (8), which backpropagates through the flow map and gives (15). The second is the *Euclidean gradient*

$$u^E(x,t) = \nabla r(X_{t,1}(x)), \tag{18}$$

which avoids backpropagation through the neural network and is commonly leveraged by denoiser-based methods to avoid the memory cost of the network Jacobian (Kim et al., 2025; Patel et al., 2025; He et al., 2023; Rout et al., 2024). By Proposition 2.4, the Jacobian-based guidance projects the reward gradient onto the data-manifold tangent space, which serves to transport the data-space gradient at the endpoint $X_{t,1}(x_t)$ back to time $t$. Prior works that avoid the Jacobian, such as FlowDPS (Kim et al., 2025) and MPGD (He et al., 2023), handle this transport by renoising to time $t$; we show in Table 3 and Appendix D that this effectively amounts to a change of gain $\lambda_t$. Our Euclidean variant applies the data-space gradient at the flow-map lookahead and treats $\lambda_t$ as a free parameter for tuning; this directly optimizes the reward without reprojection and may leave the data manifold as a result (Figure 4). These two choices

give rise to two variants of our algorithm (Algorithm 1), which we term **FMRG-E** (Euclidean) and **FMRG-J** (Jacobian), respectively. In practice, we find that FMRG-E is effective when the reward landscape is well-aligned with the data manifold, while FMRG-J tends to be more effective for neural-network-based rewards with more complex landscapes (see Section 5.5 for a systematic study).

**Reducing NFEs.** Each step of FMRG-E requires evaluating $X_{t_k,1}$ to compute the reward gradient and $X_{t_k,t_{k+1}}$ to advance the flow, for a total of two NFEs per step. In the very low NFE regime, we prefer to allocate this budget to additional guidance updates rather than to accurate intermediate integration. To this end, we reuse the endpoint evaluation via the linear interpolation $X_{t_k,t_{k+1}}(x) \approx x + \frac{t_{k+1}-t_k}{1-t_k}\big(X_{t_k,1}(x) - x\big)$, reducing the cost to a single NFE per step. On its own, this linearization would not form a valid sampler, since it discards the nonlinear structure of the learned flow during advancement. However, the reward gradient supplies a consistent correction back toward the data manifold, so the trajectory is kept on track by guidance rather than by accurate free integration. Crucially, we still compute the reward gradient via the full flow map lookahead $X_{t_{k+1},1}$, so the learned flow's nonlinearity is preserved for accurate reward estimation. When we leverage early stopping, we can complete the generation with a single flow map step $X_{t_{\text{stop}},1}$. A small number of independent reinitializations can further improve robustness; we discuss this in Appendix E.2.

# 4. Related Work

**Dynamical measure transport.** Flow- (Albergo et al., 2023; Lipman et al., 2022; Liu et al., 2022b) and diffusion-based (Song et al., 2021) generative models have achieved state-of-the-art performance across continuous modalities, and consistency models (Kim et al., 2024; Song et al., 2023) and flow maps (Boffi et al., 2025b;a; Geng et al., 2025a; Frans et al., 2024; Zhou et al., 2025) have recently accelerated sampling to a few steps.

**Inference-time guidance.** Guidance steers a pre-trained generative model toward user-specified rewards without retraining (Uehara et al., 2025). Single-trajectory methods including DPS (Chung et al., 2024), FlowDPS (Kim et al., 2025), FlowChef (Patel et al., 2025), and MPGD (He et al., 2023) apply heuristic gradient corrections to the sampler, while seed-optimization methods such as ReNO (Eyring et al., 2024) and flow map trajectory tilting (FMTT) (Sabour et al., 2025a) optimize the input noise or steer multi-particle ensembles. Our framework subsumes these single-trajectory approaches as coarse approximations to a deterministic optimal-control signal derived from the flow map (Appendix D), and replaces tilt-based approximations (Moitra

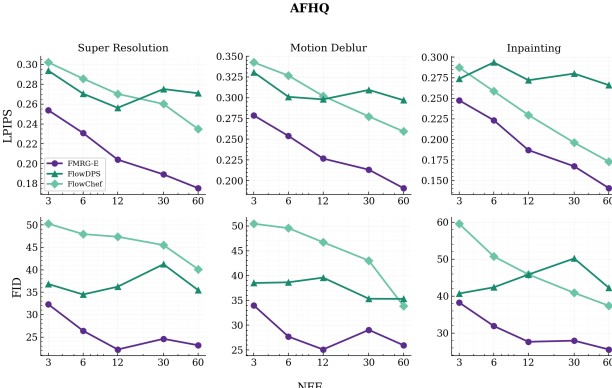

*Figure 5.* **Inverse problems.** FMRG outperforms baselines on super-resolution, motion deblurring, and inpainting at remarkably low NFEs.

et al., 2026) with an exact endpoint computation along a single trajectory. Feng et al. (2025) present a complementary unifying view based on reward-tilted velocity fields with a small-posterior-variance Taylor approximation; we instead reduce DPS-type methods directly through the exact flow-map endpoint, and require no small-variance assumption (Appendix F).

In concurrent work, Diamond Maps (Holderrieth et al., 2026), Meta Flow Maps (Potaptchik et al., 2026), and Variational Flow Maps (Mammadov et al., 2026) train new flow maps or auxiliary noise adapters to sample the reward tilt, whereas FMRG operates at inference on the existing pre-trained flow map.

See Appendix F for an extended discussion.

# 5. Experiments

We evaluate FMRG on reward functions of varying complexity, including (i) $\ell_2$ reconstruction losses for latent-space inverse problems, (ii) neural-network-based human preference rewards, and (iii) vision-language model rewards for text-to-image generation. For all experiments, we use a flow map distilled via Lagrangian distillation (Boffi et al., 2025b) from Flux.1-Dev (Black Forest Labs, 2024).[1] Inverse problems and style guidance are evaluated at $256 \times 256$ resolution, while GenEval and VLM guidance are evaluated at $512 \times 512$ resolution. We find that FMRG achieves competitive or superior sample quality in comparison to strong baselines while requiring up to $10\times$ fewer function evaluations (NFEs) on inverse problems and $70\times$ fewer on reward-guided generation. The breadth of reward types considered allows us to explore the interplay between the geometry of

---

[1]The pre-trained flow map checkpoint is publicly available at `https://huggingface.co/gabeguofanclub/flux-1-dev-flowmap-lsd`.

*Table 1.* **Quantitative comparison: Latent-space inverse problems.** We report PSNR, SSIM, LPIPS, FID, and KID for super-resolution (4×), motion deblurring, and inpainting. **Bold** indicates best, underline indicates second best. Results are averaged over 1000 images. Best hyperparameters per method are selected via grid search (see Appendix E).

| | Method | Super-Resolution | | | | | Motion Deblur | | | | | Inpainting | | | | |
|---|---|---|---|---|---|---|---|---|---|---|---|---|---|---|---|---|
| | | PSNR↑ | SSIM↑ | LPIPS↓ | FID↓ | KID↓ | PSNR↑ | SSIM↑ | LPIPS↓ | FID↓ | KID↓ | PSNR↑ | SSIM↑ | LPIPS↓ | FID↓ | KID↓ |
| **AFHQ** | DPS | 18.06 | .443 | .503 | 59.99 | .030 | 16.68 | .407 | .547 | 58.59 | .029 | 17.15 | .512 | .540 | 95.30 | .036 |
| | FlowChef | 26.87 | .767 | .243 | 43.29 | .018 | 26.75 | .749 | .250 | 38.58 | .015 | 24.12 | .828 | .160 | 35.10 | .013 |
| | FlowDPS | 27.02 | .778 | .250 | 34.58 | .013 | 26.07 | .739 | .279 | 36.77 | .014 | 26.11 | .798 | .239 | 44.10 | .018 |
| | FMRG-E (ours) | 27.12 | .772 | **.180** | **25.48** | **.009** | 27.26 | .771 | **.177** | **23.12** | **.007** | 26.12 | **.851** | **.126** | **24.73** | **.008** |
| | FMRG-J (ours) | **27.39** | **.795** | .193 | 29.51 | .012 | **27.36** | **.788** | .204 | 24.43 | .008 | **26.71** | .842 | .158 | 31.20 | .013 |
| **FFHQ** | DPS | 18.74 | .570 | .530 | 119.30 | .054 | 20.20 | .618 | .483 | 127.91 | .077 | 18.93 | .623 | .486 | 137.36 | .081 |
| | FlowChef | 26.71 | .782 | .237 | 117.04 | .106 | 24.53 | .713 | .296 | 109.30 | .081 | 25.41 | .830 | .164 | 76.52 | .057 |
| | FlowDPS | 27.70 | .818 | .205 | 61.85 | .040 | 26.85 | .789 | .233 | 64.55 | .042 | 27.90 | .844 | .195 | 73.30 | .054 |
| | FMRG-E (ours) | 27.53 | .799 | .171 | 62.63 | .042 | 28.10 | .807 | **.153** | **38.52** | **.015** | 28.66 | .883 | **.103** | **35.15** | **.014** |
| | FMRG-J (ours) | **28.23** | **.832** | **.154** | **55.99** | **.039** | **28.62** | **.834** | .155 | 41.33 | .022 | **29.48** | **.893** | .112 | 43.99 | .026 |

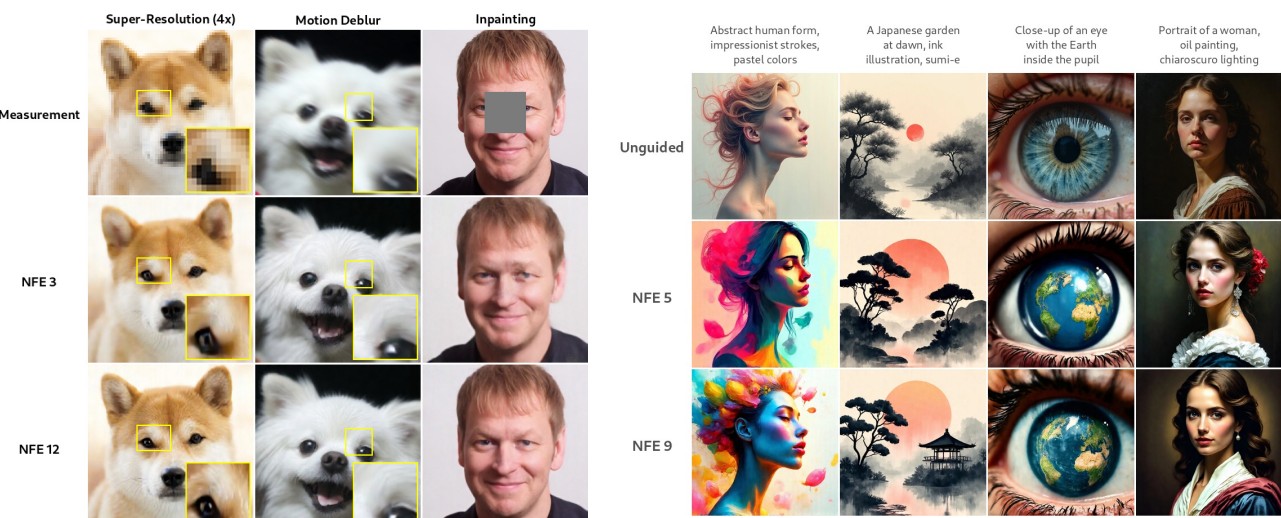

*Figure 6.* **Qualitative results.** (Left) Latent-space inverse problems: FMRG produces sharp reconstructions with as few as 3 NFEs. (Right) Reward-guided aesthetic enhancement: FMRG produces visually compelling enhancements with as few as 5 NFEs. Additional comparisons in Appendix E.6.

*Table 2.* GenEval accuracy for reward-guided generation. All methods use the same FLUX.1-dev flow map backbone. **Bold** indicates best.

| Method | NFE↓ | GenEval (Overall)↑ |
|---|---|---|
| FLUX.1-dev | 50 | 0.662 |
| Flow Map | 8 | 0.668 |
| Flow Map + Best-of-$N$ | 128 | 0.758 |
| ReNO | 58 | 0.716 |
| FMTT | 1400 | 0.771 |
| FMRG-E | 100 | 0.766 |
| FMRG-J | 20 | 0.770 |
| FMRG-J | 100 | **0.800** |

the reward landscape and the effectiveness of inference-time guidance—from simple reconstruction losses whose optima lie close to the data manifold to complex neural-network rewards whose optima may lie far from it (Section 5.5).

## 5.1. Latent-Space Inverse Problems

We compare FMRG-J and FMRG-E against FlowDPS (Kim et al., 2025), FlowChef (Patel et al., 2025), and DPS (Chung et al., 2024). We consider three latent-space inverse problems: (i) super-resolution with 4× downsampling to 64×64, (ii) motion deblurring with kernel size 61 and intensity 0.5, and (iii) box inpainting with a $64 \times 64$ mask. All methods minimize an $\ell_2$ reconstruction loss $r(x) = -\|Ax - y\|^2$ between the generated image $x \in \mathbb{R}^d$ and the observation $y \in \mathbb{R}^{d_o}$, where $A \in \mathbb{R}^{d_o \times d}$ denotes the (linear) measurement operator (see Appendix E for details).

Table 1 reports the peak performance of each method at best hyperparameters obtained via independent grid search. FMRG outperforms all baselines across the board on both AFHQ-Dog and FFHQ (Table 1). Figure 5 compares the NFE–performance trade-off for Euclidean gradient-based methods (FMRG-E, FlowDPS, FlowChef), which avoid backpropagation through the flow map. FMRG-E achieves

competitive performance even at very low NFEs (e.g., 3 and 6), matching or outperforming baselines using 2-10× more NFEs. At higher NFEs, the gap widens further, with FMRG-E and FMRG-J achieving the best results across all distortion and perception metrics (Table 1). This advantage is even more pronounced for unconditional generation (without text prompts), where FlowDPS and FlowChef degrade significantly while FMRG remains robust (see Appendix E). Wall-clock time and VRAM measurements are reported in Appendix E.8.

## 5.2. Style Guidance

We use style transfer to qualitatively illustrate the approximation hierarchy developed in Section 2. The style reward is computed via the Frobenius norm between the Gram matrices of CLIP ViT-B/16 features extracted from the reference and generated images, following MPGD (He et al., 2023). Quantitative results are provided in Table 7. Figure 20 compares all methods in the hierarchy on this task. Consistent with Proposition 2.4, FMRG-J better preserves image quality while incorporating the target style, whereas FMRG-E more directly optimizes the style reward. Both FMRG variants outperform their denoiser-based counterparts: DPS (denoiser approximation of FMRG-J) fails to incorporate the target style, while FlowChef (denoiser approximation of FMRG-E; see Appendix D) exhibits visible artifacts.

## 5.3. Reward-Guided Generation

We evaluate FMRG on human preference rewards for text-to-image generation. Following Eyring et al. (2024), we use a linear combination of human preference and text-image alignment reward models, including ImageReward (Xu et al., 2023), HPSv2 (Wu et al., 2023b), PickScore (Kirstain et al., 2023), and CLIP, as the guidance objective. The right panel of Figure 6 shows that FMRG produces visually compelling aesthetic enhancements with as few as 5 NFEs.

To quantitatively evaluate these gains, we benchmark on GenEval (Ghosh et al., 2023), an object-focused compositional benchmark. GenEval uses object detection methods that are independent of the reward models used for guidance, providing a disentangled evaluation that measures genuine compositional improvements rather than reward over-optimization (Eyring et al., 2024).

As shown in Table 2, FMRG-J achieves a GenEval score of 0.80 at NFE 100 and matches FMTT (0.77) at NFE 20, a 70× reduction in NFEs (Pareto frontier in Figure 21). We additionally compare against the base FLUX.1-Dev model (Black Forest Labs, 2024), best-of-$N$ reward selection, and seed optimization via ReNO. Wall-clock time and VRAM measurements are reported in Appendix E.8.

## 5.4. VLM Guidance

Beyond the fixed reward ensembles used above, FMRG can leverage complex vision-language model (VLM) rewards to guide generation toward detailed, compositional prompts that base models struggle to follow. We use a 7B-parameter VLM (Wang et al., 2025) that scores prompt-image alignment, representing the most complex reward landscape in our evaluation (see Appendix E.7 for details). Figure 25 shows examples where unguided FLUX generations fail to capture fine-grained compositional details, while FMRG-J successfully steers the output toward prompt-faithful images.

## 5.5. Analysis of Design Choices

We discuss two key design choices whose empirical behavior is consistent with our theoretical analysis. Full ablations are provided in Appendices E.4 and E.6.

**Early stopping.** Theory predicts that greedy guidance contracts variance exponentially without early stopping (Proposition 2.5); empirically, its importance tracks the reward's susceptibility to over-optimization. For inverse problems, the $\ell_2$ loss has a unique optimum at the ground truth, so early stopping mostly helps at low NFEs (Table 6). For human preference rewards, early stopping at $t_{\mathrm{stop}} = 0.25$ consistently improves GenEval accuracy (Table 9); without it, quality degrades due to over-optimization artifacts (Figure 22).

**Jacobian vs. Euclidean.** The Jacobian projects reward gradients onto the manifold tangent space (Proposition 2.4), which should matter more when reward optima lie off-manifold. For $\ell_2$ losses with on-manifold optima, FMRG-E matches FMRG-J at low NFEs (Figure 5); for human preference rewards (∼3.4B params) with potentially off-manifold optima, FMRG-J leads by 3.4 points at matched NFE (Table 2).

## 6. Conclusion and Limitations

We introduced Flow Map Reward Guidance (FMRG), a deterministic optimal-control framework for inference-time guidance of flow and flow-map generative models, yielding efficient sampling with strong empirical performance.

**Limitations.** FMRG-J requires backpropagation through the flow map (∼48 GB VRAM at 512px); FMRG-E avoids this and fits on a single L40S. Performance depends on the pre-trained flow map quality, and myopic guidance can over-optimize without early stopping (Appendix C.5).

## Acknowledgements

We thank Gabe Guo for providing the FLUX-distilled flow map checkpoint used in all experiments. We also thank Carles Domingo-Enrich, Peter Holderrieth, and Stephen Huan for helpful discussions.

## Impact Statement

This paper presents work whose goal is to advance the field of machine learning. As with all controllable generation methods, FMRG could be used to generate misleading or harmful content. We believe the benefits of efficient, principled guidance outweigh these risks, but encourage responsible deployment.

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

# A. Background on flow maps

In this section, we provide some brief further background on flow maps. For complete details, we refer the reader to (Boffi et al., 2025a;b). Given a probability flow $\dot{x}_t = b_t(x_t)$, the *flow map* $X : [0,1]^2 \times \mathbb{R}^d \to \mathbb{R}^d$ is the solution operator satisfying $X_{s,t}(x_s) = x_t$, i.e., $X_{s,t}$ maps the state at time $s$ to the state at time $t$ along the flow. Equivalently, we may write

$$X_{s,t}(x) = x + \int_s^t b_\tau(X_{s,\tau}(x))\, d\tau. \tag{19}$$

The flow map enjoys several fundamental properties:

**Proposition A.1.** *The flow map satisfies the following:*

1. *The* Semigroup property: *for all* $(s, u, t) \in [0,1]^3$ *and for all* $x \in \mathbb{R}^d$,

$$X_{s,t}(x) = X_{u,t}(X_{s,u}(x)). \tag{20}$$

2. *The* Lagrangian equation: *for all* $(s, t) \in [0,1]^2$ *and for all* $x \in \mathbb{R}^d$,

$$\partial_t X_{s,t}(x) = b_t(X_{s,t}(x)). \tag{21}$$

3. *The* Eulerian equation: *for all* $(s, t) \in [0,1]^2$ *and for all* $x \in \mathbb{R}^d$,

$$\partial_s X_{s,t}(x) + \nabla X_{s,t}(x)\, b_s(x) = 0. \tag{22}$$

Following recent work on accelerated sampling (Boffi et al., 2025b; Sabour et al., 2025b; Geng et al., 2025a), we parameterize the flow map as

$$X_{s,t}(x) = x + (t - s)v_{s,t}(x), \tag{23}$$

where $v : [0,1]^2 \times \mathbb{R}^d \to \mathbb{R}^d$ is a learned velocity function. On the diagonal $s = t$, the Lagrangian equation implies

$$v_{t,t}(x) = b_t(x), \tag{24}$$

i.e., the parameterized velocity recovers the probability flow drift. Equation (24) is known as the *tangent condition*, and enables us to compare to baselines that are based on a pre-trained flow with a single model.

Below, we collect a simple property of the flow map that we will use in later proofs.

**Proposition A.2** (Jacobian evolution). *The Jacobian* $\nabla X_{s,t}(x)$ *satisfies the variational equation*

$$\frac{d}{dt}\nabla X_{s,t}(x) = \nabla b_t(X_{s,t}(x))\, \nabla X_{s,t}(x), \qquad \nabla X_{s,s}(x) = I. \tag{25}$$

*Consequently, the transpose* $\nabla X_{s,t}(x)^\mathsf{T}$ *satisfies the adjoint equation*

$$\frac{d}{dt}\nabla X_{s,t}(x)^\mathsf{T} = \nabla X_{s,t}(x)^\mathsf{T}\, \nabla b_t(X_{s,t}(x))^\mathsf{T}, \qquad \nabla X_{s,s}(x)^\mathsf{T} = I. \tag{26}$$

*Proof.* Differentiating the Lagrangian equation (25) with respect to $x$ and permuting derivatives:

$$\frac{d}{dt}\nabla X_{s,t}(x) = \nabla_x\big[b_t(X_{s,t}(x))\big] = \nabla b_t(X_{s,t}(x))\, \nabla X_{s,t}(x), \tag{27}$$

where we applied the chain rule. The initial condition $\nabla X_{s,s}(x) = I$ follows from $X_{s,s}(x) = x$. The adjoint equation is obtained by transposing and using $(AB)^\mathsf{T} = B^\mathsf{T}A^\mathsf{T}$. $\square$

## B. Background on reward tilting and stochastic optimal control

We establish the connection between stochastic optimal control (SOC) and the reward-tilted distribution, following the viewpoint of Domingo-Enrich et al. (2025). Let $(X_t)_{t\in[0,1]}$ denote a *base* generative diffusion (the pre-trained sampler) with dynamics

$$dX_t = b_t^{\text{SDE}}(X_t)\,dt + \eta_t\,dW_t, \qquad X_0 \sim \rho_0, \tag{28}$$

where $\eta_t$ is a (scalar) diffusion schedule and the SDE drift is related to the probability flow ODE drift $b_t$ by

$$b_t^{\text{SDE}}(x) = b_t(x) + \frac{\eta_t^2}{2}\nabla\log\rho_t(x), \tag{29}$$

with $\nabla\log\rho_t$ the score of the marginal density at time $t$. We study the inverse temperature-$\lambda$ SOC problem

$$\min_u\ \mathbb{E}\left[\int_0^1 \frac{1}{2\lambda}\|u_t(X_t^u)\|^2\,dt\ -\ r(X_1^u)\right],$$
$$\text{s.t.}\ \ dX_t^u = \left(b_t^{\text{SDE}}(X_t^u) + \eta_t u_t(X_t^u)\right)dt + \eta_t\,dW_t, \tag{30}$$

where $r : \mathbb{R}^d \to \mathbb{R}_{\geqslant 0}$ is the terminal reward and $u : [0,1]\times\mathbb{R}^d \to \mathbb{R}^d$ is the control in Brownian coordinates.

**Doob $h$-transform.** Define the Doob $h$-function, $h : [0,1]\times\mathbb{R}^d \to \mathbb{R}^d$,

$$h_t(x) := \mathbb{E}\big[\exp(\lambda r(X_1))\,\big|\,X_t = x\big], \tag{31}$$

where the expectation is taken under the *base* diffusion (28) (i.e., $u \equiv 0$). Let

$$V_t(x) := -\log h_t(x). \tag{32}$$

Then the optimally-controlled drift is given by the Doob $h$-transform (Doob, 1957):

$$b_t^{\star}(x) = b_t^{\text{SDE}}(x) + \eta_t^2\nabla_x\log h_t(x) = b_t^{\text{SDE}}(x) - \eta_t^2\nabla_x V_t(x), \tag{33}$$

which can be naturally understood as adding to the base process a gradient flow term on the expected cost-to-go. Equivalently, the optimal feedback is

$$u_t^*(x) = \eta_t\nabla_x\log h_t(x) = -\eta_t\nabla_x V_t(x). \tag{34}$$

**Memoryless schedule.** Adjoint Matching defines the *memoryless* condition to mean that the endpoint is independent of the initial condition, $X_0 \perp X_1$, under the *base* process (28) (Domingo-Enrich et al., 2025). For stochastic interpolants with coefficients $(\alpha_t, \beta_t)$, the memoryless diffusion schedule is typically written in terms of the coefficient

$$\frac{\eta_t^2}{2} = \frac{\alpha_t^2\dot{\beta}_t}{\beta_t} - \dot{\alpha}_t\alpha_t, \tag{35}$$

which for the linear interpolant $\alpha_t = 1 - t$ and $\beta_t = t$ reduces to $\eta_t^2/2 = (1-t)/t$. This choice ensures that the two-time marginals $(X_t, X_{t'})$ of the base process match those of the interpolant, which by construction implies the memoryless condition.

Under the memoryless condition, marginalizing the path measure of the optimally-controlled process implies that the *optimal terminal marginal* is the reward tilt:

$$\rho_{\text{tilt}}(x)\ \propto\ e^{\lambda r(x)}\rho_1(x). \tag{36}$$

This provides the target distribution for tilt guidance, whose drift correction (3) is the gradient of the SOC value function.

## C. Omitted proofs

In this section, we develop some rigorous approximation theory for the optimal control problem (4).

## C.1. Setup and regularity assumptions

We work with the value function associated with the optimal control problem (4):

$$V_t^\lambda(x) := \inf_{u \in L^2([t,1];\mathbb{R}^d)} \left\{ \int_t^1 \frac{1}{2\lambda} \|u_\tau\|^2 \, d\tau \; - \; r(x_1^u) \right\}, \tag{37}$$

where $x^u$ solves the controlled dynamics $\dot{x}_\tau = b_\tau(x_\tau) + u_\tau$ starting from $x_t = x$. Throughout, $\|\cdot\|$ denotes the Euclidean norm and $\|\cdot\|_{\mathrm{op}}$ the operator norm. We make the following standard regularity assumptions.

**Assumption C.1** (Regularity). Fix a domain $\Omega \subseteq \mathbb{R}^d$ containing all relevant trajectories. There exist constants $M, K, G, L_r > 0$ such that for all $s \in [0,1]$ and $z, z' \in \Omega$:

(i) $\|\nabla b_s(z)\|_{\mathrm{op}} \leqslant M$ and $\|\nabla b_s(z) - \nabla b_s(z')\|_{\mathrm{op}} \leqslant K\|z - z'\|$,

(ii) $\|\nabla r(z)\| \leqslant G$ and $\|\nabla r(z) - \nabla r(z')\| \leqslant L_r\|z - z'\|$.

Under Assumption C.1, we have the following standard flow map Jacobian bounds.

**Lemma C.2** (Jacobian bound). *For $t \geqslant s$, the flow map Jacobian satisfies*

$$\|\nabla X_{s,t}(x)\|_{\mathrm{op}} \leqslant e^{M(t-s)}. \tag{38}$$

*Proof.* By Proposition A.2, the flow map Jacobian satisfies the variational equation (25). Since $\frac{d}{dt}\|A(t)\|_{\mathrm{op}} \leqslant \|\dot{A}(t)\|_{\mathrm{op}}$ for any differentiable matrix-valued function $A(t)$, we have

$$\frac{d}{dt}\|\nabla X_{s,t}\|_{\mathrm{op}} \leqslant \left\| \frac{d}{dt} \nabla X_{s,t} \right\|_{\mathrm{op}} = \|\nabla b_t(X_{s,t})\nabla X_{s,t}\|_{\mathrm{op}} \leqslant M\|\nabla X_{s,t}\|_{\mathrm{op}}. \tag{39}$$

Grönwall's inequality with initial condition $\|\nabla X_{s,s}\|_{\mathrm{op}} = 1$ yields the result. $\quad\square$

**Lemma C.3** (Hessian bound). *For $t \geqslant s$, the flow map Hessian satisfies*

$$\|\nabla^2 X_{s,t}(x)\| \leqslant \frac{K}{M}\big(e^{2M(t-s)} - e^{M(t-s)}\big). \tag{40}$$

*Proof.* Differentiating the variational equation (25) with respect to $x$:

$$\frac{d}{dt}\nabla^2 X_{s,t}(x) = \nabla^2 b_t(X_{s,t}(x))\langle \nabla X_{s,t}(x), \nabla X_{s,t}(x)\rangle + \nabla b_t(X_{s,t}(x))\,\nabla^2 X_{s,t}(x). \tag{41}$$

By the same reasoning as in Lemma C.2, taking norms, using $\|\nabla^2 b_t\|_{\mathrm{op}} \leqslant K$ from Assumption C.1, and applying the result of Lemma C.2:

$$\frac{d}{dt}\|\nabla^2 X_{s,t}\| \leqslant K\|\nabla X_{s,t}\|_{\mathrm{op}}^2 + M\|\nabla^2 X_{s,t}\| \leqslant Ke^{2M(t-s)} + M\|\nabla^2 X_{s,t}\|. \tag{42}$$

Applying Gronwall's inequality with initial condition $\nabla^2 X_{s,s} = 0$:

$$\begin{aligned}
\|\nabla^2 X_{s,t}\| &\leqslant \int_s^t Ke^{2M(\tau-s)}e^{M(t-\tau)} \, d\tau \\
&= Ke^{M(t-s)} \int_s^t e^{M(\tau-s)} \, d\tau \\
&= \frac{K}{M}\big(e^{2M(t-s)} - e^{M(t-s)}\big).
\end{aligned} \tag{43}$$

This completes the proof. $\quad\square$

These simple results will be used later to understand the error in our greedy approximation.

## C.2. Proof of Proposition 2.1

**Proposition 2.1.** *Let $u_t^*$ be an optimal solution of (4), and let $x_t^*$ denote its trajectory. Then, along $x_t^*$,*

$$u_t^* = \lambda \nabla X_{t,1}^{u^*}(x_t^*)^\mathsf{T} \nabla r\left(X_{t,1}^{u^*}(x_t^*)\right),\tag{5}$$

*where $X_{s,t}^{u^*}$ is the flow map for the optimally controlled dynamics $\dot{x}_t^{u^*} = b_t(x_t^{u^*}) + u_t^*$.*

*Proof.* The Lagrangian for the optimal control problem (4) is given by

$$\mathcal{L}(x, u, p) = \int_0^1 \left(\frac{\|u_t\|^2}{2\lambda(t)} + p_t^\mathsf{T}(b_t(x_t) + u_t - \dot{x}_t)\right) dt - r(x_1^u),\tag{44}$$

where $p : [0, 1] \to \mathbb{R}^d$ denotes the adjoint variable, or the Lagrange multiplier enforcing the controlled dynamics constraint $\dot{x}_t^u = b_t(x_t^u) + u_t(x_t^u)$. Taking the first variation with respect to the curves $x_t, u_t$, and $p_t$ gives the optimality conditions of the Pontryagin Maximum Principle (Fleming & Rishel, 1975),

$$\begin{aligned}
\frac{\delta}{\delta x} &: \quad \dot{p}_t^* = -\nabla b_t(x_t^*)^\mathsf{T} p_t^*, & p_1^* &= -\nabla r(x_1^*),\\
\frac{\delta}{\delta p} &: \quad \dot{x}_t^* = b_t(x_t^*) + u_t^*(x_t^*), & x_0^* &= x_0,\\
\frac{\delta}{\delta u} &: \quad u_t^* = -\lambda(t)p_t^*.
\end{aligned}\tag{45}$$

The second relation simply enforces the controlled dynamics constraint. The third relation demonstrates a simple scaling relationship between the optimal guidance $u_t^*$ and the adjoint $p_t^*$, reducing the problem to solving for $p_t^*$.

The adjoint equation $\dot{p}_t = -\nabla b_t(x_t)^\mathsf{T} p_t$ is a linear ordinary differential equation whose solution we now derive analytically. By the semigroup property (20), for any $s < t < 1$:

$$X_{s,1}^u(x_s^u) = X_{t,1}^u(X_{s,t}^u(x_s^u)).\tag{46}$$

Taking the Jacobian with respect to $x_s^u$ via the chain rule:

$$\nabla X_{s,1}^u(x_s^u) = \nabla X_{t,1}^u(x_t^u)\nabla X_{s,t}^u(x_s^u),\tag{47}$$

where $x_t^u = X_{s,t}^u(x_s^u)$. The left-hand side is independent of the intermediate time $t$, so differentiating both sides with respect to $t$ gives

$$0 = \frac{d}{dt}\left[\nabla X_{t,1}^u(x_t^u)\right]\nabla X_{s,t}^u(x_s^u) + \nabla X_{t,1}^u(x_t^u)\frac{d}{dt}\left[\nabla X_{s,t}^u(x_s^u)\right].\tag{48}$$

By the Jacobian evolution equation (25), the forward Jacobian satisfies $\frac{d}{dt}[\nabla X_{s,t}^u(x_s^u)] = \nabla b_t(x_t^u)\nabla X_{s,t}^u(x_s^u)$. Here, only the state-dependent part of the dynamics contributes because we treat the (open-loop controller) $u$ as independent of $x$ throughout the functional optimization. Substituting and using the invertibility of $\nabla X_{s,t}^u$ (which follows because $X_{s,t}$ is a diffeomorphism):

$$\frac{d}{dt}\left[\nabla X_{t,1}^u(x_t^u)\right] = -\nabla X_{t,1}^u(x_t^u)\nabla b_t(x_t^u).\tag{49}$$

Taking the transpose of both sides:

$$\frac{d}{dt}\left[\nabla X_{t,1}^u(x_t^u)^\mathsf{T}\right] = -\nabla b_t(x_t^u)^\mathsf{T}\nabla X_{t,1}^u(x_t^u)^\mathsf{T}.\tag{50}$$

This shows that $\nabla X_{t,1}^u(x_t^u)^\mathsf{T}$ exactly satisfies the adjoint equation, with terminal condition $\nabla X_{1,1}^u = I$. Therefore, the solution to the adjoint equation with terminal condition $p_1 = -\nabla r(x_1^u)$ is

$$p_t = \nabla X_{t,1}^u(x_t^u)^\mathsf{T} p_1 = -\nabla X_{t,1}^u(x_t^u)^\mathsf{T}\nabla r(x_1^u).\tag{51}$$

Substituting into the optimality condition $u_t^* = -\lambda(t)p_t^*$ yields the optimal control

$$u_t^* = \lambda(t)\nabla X_{t,1}^u(x_t^u)^\mathsf{T}\nabla r(x_1^u).\tag{52}$$

This completes the proof. □

### C.3. HJB characterization and small-$\lambda$ expansion

By Bellman's principle of optimality, the value function (37) satisfies the Hamilton–Jacobi–Bellman equation (Fleming & Rishel, 1975)

$$\partial_t V_t^\lambda(x) + b_t(x) \cdot \nabla V_t^\lambda(x) - \frac{\lambda}{2} \|\nabla V_t^\lambda(x)\|^2 = 0, \qquad V_1^\lambda(x) = -r(x), \tag{53}$$

with optimal feedback $u_t^*(x) = -\lambda \nabla V_t^\lambda(x)$. The following result, stated informally as Proposition 2.2 in the main text, shows that greedy guidance corresponds to the $\lambda = 0$ limit and is accurate to $O(\lambda^2)$.

**Proposition C.4.** *Under Assumption C.1, suppose that $V_t^\lambda$ admits an expansion uniformly for all $x \in \mathbb{R}^d$*

$$V_t^\lambda(x) = V_t^0(x) + \lambda V_t^1(x) + O(\lambda^2). \tag{54}$$

*Then we have that:*

*(i) The $\lambda = 0$ solution is $V_t^0(x) = -r(X_{t,1}(x))$, and the greedy guidance (8) is $u_t^J(x) = -\lambda \nabla V_t^0(x)$.*

*(ii) The optimal feedback satisfies*

$$\|u_t^*(x) - u_t^J(x)\| \leqslant C_1 \lambda^2 + O(\lambda^3), \qquad \forall x \in \mathbb{R}^d \tag{55}$$

*for a constant $C_1$ depending on $\Omega$, $M$, $K$, $G$, and $L_r$.*

*Proof.* Setting $\lambda = 0$ in (53) yields the transport equation

$$\partial_t V_t^0(x) + b_t(x) \cdot \nabla V_t^0(x) = 0, \qquad V_1^0(x) = -r(x). \tag{56}$$

We solve this transport equation via the method of characteristics. For any initial point $x$, consider the characteristic curve $\gamma(\tau) := X_{t,\tau}(x)$ for $\tau \in [t, 1]$. By the chain rule,

$$\begin{aligned}
\frac{d}{d\tau} V_\tau^0(\gamma(\tau)) &= \partial_\tau V_\tau^0(\gamma(\tau)) + \dot{\gamma}(\tau) \cdot \nabla V_\tau^0(\gamma(\tau)) \\
&= \partial_\tau V_\tau^0(\gamma(\tau)) + b_\tau(\gamma(\tau)) \cdot \nabla V_\tau^0(\gamma(\tau)) \\
&= 0,
\end{aligned} \tag{57}$$

where the last equality uses (56). Thus $V_\tau^0(\gamma(\tau))$ is constant along characteristics. Setting $\gamma(t) = x$ and using that $V_1^0(x) = -r(x)$ for all $x \in \mathbb{R}^d$ we have

$$V_t^0(x) = V_t^0(\gamma(t)) = V_1^0(\gamma(1)) = V_1^0(X_{t,1}(x)) = -r(X_{t,1}(x)). \tag{58}$$

Differentiating via the chain rule gives the gradient:

$$\nabla V_t^0(x) = -\nabla X_{t,1}(x)^\top \nabla r(X_{t,1}(x)). \tag{59}$$

Thus $u_t^J(x) := -\lambda \nabla V_t^0(x) = \lambda \nabla X_{t,1}(x)^\top \nabla r(X_{t,1}(x))$, which is exactly the greedy guidance (8).

For part (ii), insert $V_t^\lambda(x) = V_t^0(x) + \lambda V_t^1(x) + O(\lambda^2)$ into (53). Since $\nabla V_t^\lambda(x) = \nabla V_t^0(x) + \lambda \nabla V_t^1(x) + O(\lambda^2)$, we have $\|\nabla V_t^\lambda(x)\|^2 = \|\nabla V_t^0(x)\|^2 + O(\lambda)$. Substituting:

$$\begin{aligned}
0 &= \partial_t V_t^\lambda(x) + b_t(x) \cdot \nabla V_t^\lambda(x) - \frac{\lambda}{2} \|\nabla V_t^\lambda(x)\|^2 \\
&= \left(\partial_t V_t^0(x) + \lambda \partial_t V_t^1(x)\right) + b_t(x) \cdot \left(\nabla V_t^0(x) + \lambda \nabla V_t^1(x)\right) - \frac{\lambda}{2}\left(\|\nabla V_t^0(x)\|^2 + O(\lambda)\right) + O(\lambda^2) \\
&= \underbrace{\left(\partial_t V_t^0(x) + b_t(x) \cdot \nabla V_t^0(x)\right)}_{O(1)} + \lambda \underbrace{\left(\partial_t V_t^1(x) + b_t(x) \cdot \nabla V_t^1(x) - \frac{1}{2}\|\nabla V_t^0(x)\|^2\right)}_{O(\lambda)} + O(\lambda^2).
\end{aligned} \tag{60}$$

Matching orders, the $O(1)$ term recovers (56), while the $O(\lambda)$ term shows that $V_t^1$ solves

$$\partial_t V_t^1(x) + b_t(x) \cdot \nabla V_t^1(x) = \frac{1}{2}\|\nabla V_t^0(x)\|^2, \qquad V_1^1(x) = 0, \quad \forall x \in \mathbb{R}^d. \tag{61}$$

The boundary condition on $V_t^1(x)$ follows by observing that $V_1^0(x) = -r(x)$ already satisfies the boundary condition for $V_t^\lambda(x)$. This is an inhomogeneous transport equation, which we solve via Duhamel's principle. As before, consider the characteristic $\gamma(\tau) := X_{t,\tau}(x)$ for $\tau \in [t, 1]$. By the chain rule,

$$
\begin{aligned}
\frac{d}{d\tau} V_\tau^1(\gamma(\tau)) &= \partial_\tau V_\tau^1(\gamma(\tau)) + \dot{\gamma}(\tau) \cdot \nabla V_\tau^1(\gamma(\tau)) \\
&= \partial_\tau V_\tau^1(\gamma(\tau)) + b_\tau(\gamma(\tau)) \cdot \nabla V_\tau^1(\gamma(\tau)) \\
&= \frac{1}{2} \|\nabla V_\tau^0(\gamma(\tau))\|^2,
\end{aligned}
\tag{62}
$$

where the last equality uses (61). Unlike the homogeneous case, $V_\tau^1(\gamma(\tau))$ is not constant but accumulates the source term. Integrating from $t$ to 1 and using $\gamma(t) = x$, $\gamma(\tau) = X_{t,\tau}(x)$, and $V_1^1 = 0$:

$$
V_1^1(X_{t,1}(x)) - V_t^1(x) = \int_t^1 \frac{1}{2} \|\nabla V_\tau^0(X_{t,\tau}(x))\|^2 \, d\tau
$$

$$
\implies \quad V_t^1(x) = -\frac{1}{2} \int_t^1 \|\nabla V_\tau^0(X_{t,\tau}(x))\|^2 \, d\tau.
\tag{63}
$$

To derive (55), we use $u_t^*(x) = -\lambda \nabla V_t^\lambda(x)$ and differentiate the expansion:

$$
\begin{aligned}
u_t^*(x) &= -\lambda \nabla V_t^\lambda(x) \\
&= -\lambda \nabla \big(V_t^0(x) + \lambda V_t^1(x) + O(\lambda^2)\big) \\
&= -\lambda \nabla V_t^0(x) - \lambda^2 \nabla V_t^1(x) + O(\lambda^3) \\
&= u_t^J(x) - \lambda^2 \nabla V_t^1(x) + O(\lambda^3).
\end{aligned}
\tag{64}
$$

Thus $\|u_t^*(x) - u_t^J(x)\| = \lambda^2 \|\nabla V_t^1(x)\| + O(\lambda^3)$. It remains to show that $\|\nabla V_t^1(x)\|$ is uniformly bounded. From (59), we have $\nabla V_\tau^0(y) = -\nabla X_{\tau,1}(y)^\top \nabla r(X_{\tau,1}(y))$, so by Lemma C.2:

$$
\|\nabla V_\tau^0(y)\| \leqslant \|\nabla X_{\tau,1}(y)\|_{\mathrm{op}} \|\nabla r(X_{\tau,1}(y))\| \leqslant e^{M(1-\tau)} G.
\tag{65}
$$

Differentiating the integral representation (63) with respect to $x$ via the chain rule:

$$
\begin{aligned}
\nabla V_t^1(x) &= -\frac{1}{2} \int_t^1 \nabla_x \|\nabla V_\tau^0(X_{t,\tau}(x))\|^2 \, d\tau \\
&= -\int_t^1 \nabla X_{t,\tau}(x)^\top \nabla^2 V_\tau^0(X_{t,\tau}(x)) \, \nabla V_\tau^0(X_{t,\tau}(x)) \, d\tau.
\end{aligned}
\tag{66}
$$

Taking norms and using Lemma C.2:

$$
\begin{aligned}
\|\nabla V_t^1(x)\| &\leqslant \int_t^1 \|\nabla X_{t,\tau}(x)\|_{\mathrm{op}} \|\nabla^2 V_\tau^0(X_{t,\tau}(x))\|_{\mathrm{op}} \|\nabla V_\tau^0(X_{t,\tau}(x))\| \, d\tau \\
&\leqslant \int_t^1 e^{M(\tau-t)} \cdot H_\tau \cdot e^{M(1-\tau)} G \, d\tau,
\end{aligned}
\tag{67}
$$

It remains to bound $H_\tau := \sup_{y \in \Omega} \|\nabla^2 V_\tau^0(y)\|_{\mathrm{op}}$. Differentiating (59) via the product rule:

$$
\nabla^2 V_\tau^0(y) = -\nabla^2 X_{\tau,1}(y) \, \nabla r(X_{\tau,1}(y)) - \nabla X_{\tau,1}(y)^\top \nabla^2 r(X_{\tau,1}(y)) \, \nabla X_{\tau,1}(y).
\tag{68}
$$

For the second term, using Lemma C.2 and $\|\nabla^2 r\|_{\mathrm{op}} \leqslant L_r$ from Assumption C.1:

$$
\|\nabla X_{\tau,1}^\top \nabla^2 r \, \nabla X_{\tau,1}\|_{\mathrm{op}} \leqslant \|\nabla X_{\tau,1}\|_{\mathrm{op}}^2 L_r \leqslant e^{2M(1-\tau)} L_r.
\tag{69}
$$

For the first term, by Lemma C.3:

$$
\|\nabla^2 X_{\tau,1} \, \nabla r\|_{\mathrm{op}} \leqslant \|\nabla^2 X_{\tau,1}\| \cdot G \leqslant \frac{KG}{M} \big(e^{2M(1-\tau)} - e^{M(1-\tau)}\big).
\tag{70}
$$

Combining both terms:

$$H_\tau \leqslant \frac{KG}{M}\left(e^{2M(1-\tau)} - e^{M(1-\tau)}\right) + L_r e^{2M(1-\tau)}. \tag{71}$$

Substituting back and integrating yields $\|\nabla V_t^1(x)\| \leqslant C_1'$ for a constant $C_1'$ depending on $M$, $K$, $G$, and $L_r$. Therefore:

$$\|u_t^*(x) - u_t^J(x)\| = \lambda^2 \|\nabla V_t^1(x)\| + O(\lambda^3) \leqslant C_1' \lambda^2 + O(\lambda^3), \tag{72}$$

which gives (55) with $C_1 = C_1'$. $\qquad\square$

### C.4. Proof of Proposition 2.3

Before giving the proof, we record an identity relating the controlled and uncontrolled flow maps that we will use to compute first-order perturbations.

**Lemma C.5** (Eulerian perturbation of the flow map). *Let $X_{s,t}$ denote the flow map for the uncontrolled dynamics $\dot{x}_\tau = b_\tau(x_\tau)$, and let $X_{s,t}^u$ denote the flow map for the controlled dynamics $\dot{x}_\tau^u = b_\tau(x_\tau^u) + u_\tau(x_\tau^u)$. Then for every $x \in \mathbb{R}^d$ and every $(s,t) \in [0,1]^2$,*

$$X_{s,t}^u(x) = X_{s,t}(x) + \int_s^t \nabla X_{\tau,t}(x_\tau^u)\, u_\tau(x_\tau^u)\, d\tau, \tag{73}$$

*where $x_\tau^u = X_{s,\tau}^u(x)$ denotes the controlled trajectory initialized from $x$ at time $s$.*

*Proof.* Define $F_s := X_{s,t}(x_s^u)$, the uncontrolled flow map evaluated at the controlled trajectory, and view it as a function of the initial time $s$. Differentiating and using the Eulerian equation $\partial_s X_{s,t}(x) + \nabla X_{s,t}(x)\, b_s(x) = 0$ from (22), together with the controlled dynamics $\dot{x}_s^u = b_s(x_s^u) + u_s(x_s^u)$, yields

$$\dot{F}_s = \partial_s X_{s,t}(x_s^u) + \nabla X_{s,t}(x_s^u)\, \dot{x}_s^u = \nabla X_{s,t}(x_s^u)\, u_s(x_s^u). \tag{74}$$

Integrating from $s$ to $t$ and using $F_t = x_t^u = X_{s,t}^u(x)$ gives (73). $\qquad\square$

**Proposition 2.3.** *Consider the restricted problem*

$$\min_{u \in \mathcal{U}_t^{\delta t}} \mathcal{L}(u) = \min_{u \in \mathcal{U}_t^{\delta t}} \int_0^1 \frac{\|u_\tau\|^2}{2\lambda}\, d\tau - r(x_1^u). \tag{11}$$

*Then, the optimal guidance signal is given by (8) as $\delta t \to 0$.*

*Proof.* By definition, $u \in \mathcal{U}_t^{\delta t}$ is non-zero only on the window $[t, t+\delta t]$. The control cost on this class is

$$\int_0^1 \frac{\|u_\tau\|^2}{2\lambda_\tau}\, d\tau = \int_t^{t+\delta t} \frac{\|u_\tau\|^2}{2\lambda_\tau}\, d\tau = \frac{\|u_t\|^2}{2\lambda_t}\delta t + o(\delta t). \tag{75}$$

To relate the terminal reward $r(x_1^u)$ to the control $u$, we compute the first-order expansion of the terminal point $x_1^u$ in $\delta t$. Applying Lemma C.5 with $(s,t) \to (t,1)$ and using that $u$ vanishes outside $[t, t+\delta t]$,

$$X_{t,1}^u(x_t) = X_{t,1}(x_t) + \int_t^{t+\delta t} \nabla X_{\tau,1}(x_\tau^u)\, u_\tau(x_\tau^u)\, d\tau. \tag{76}$$

The integrand is continuous in $\tau$ and equals $\nabla X_{t,1}(x_t)\, u_t$ at $\tau = t$, since the controlled trajectory satisfies $x_t^u = x_t$ at the initial time. A left Riemann sum approximation on the window of length $\delta t$ therefore gives

$$X_{t,1}^u(x_t) = X_{t,1}(x_t) + \nabla X_{t,1}(x_t)\, u_t\, \delta t + o(\delta t). \tag{77}$$

Using (77) to Taylor expand the reward,

$$r(x_1^u) = r(X_{t,1}(x_t)) + \nabla r(X_{t,1}(x_t))^\mathsf{T} \nabla X_{t,1}(x_t)\, u_t\, \delta t + o(\delta t). \tag{78}$$

Substituting into the objective (11) and retaining leading-order terms in $\delta t$,

$$\min_{u_t} \frac{\|u_t\|^2}{2\lambda_t} - \nabla r(X_{t,1}(x_t))^{\mathsf{T}} \nabla X_{t,1}(x_t)\, u_t. \tag{79}$$

This is a convex quadratic in $u_t$, and setting its gradient to zero gives the optimal control

$$u_t^* = \lambda_t \, \nabla X_{t,1}(x_t)^{\mathsf{T}} \nabla r(X_{t,1}(x_t)), \tag{80}$$

which completes the proof. $\qquad\square$

## C.5. Gaussian case

Consider a setting in which the base distribution is a standard Gaussian $\rho_0 = \mathsf{N}(0,1)$, the target is $\rho_1 = \mathsf{N}(\mu_1, \sigma_1^2)$, and the reward is quadratic: $r(x) = -(x-a)^2$ for some target location $a$. This setting admits closed-form solutions for both the reward tilt and the optimal control problem (5). The analysis below also extends to the multivariate case with dense covariance matrices, but we present the scalar setting where the equations are simpler and more illuminating.

### C.5.1. REWARD-TILTED DISTRIBUTION

We begin by specializing the reward tilt $\rho_{\text{tilt}} \propto e^{\lambda r(x)} \rho_1(x)$ (see Appendix B) to the Gaussian target with quadratic reward.

**Proposition C.6** (Reward-tilted distribution). *The reward-tilted distribution $\rho_{\text{tilt}}(x) \propto e^{\lambda r(x)} \rho_1(x)$ is Gaussian, $\rho_{\text{tilt}} = \mathcal{N}(\mu_{\text{tilt}}, \sigma_{\text{tilt}}^2)$, with*

$$\sigma_{\text{tilt}}^2 = \frac{\sigma_1^2}{1 + 2\lambda\sigma_1^2}, \qquad \mu_{\text{tilt}} = \frac{\mu_1 + 2\lambda\sigma_1^2 a}{1 + 2\lambda\sigma_1^2}. \tag{81}$$

*Proof.* The proof is elementary, and proceeds by completing the square. The log-density is

$$\log \rho_{\text{tilt}}(x) = -\lambda(x-a)^2 - \frac{(x-\mu_1)^2}{2\sigma_1^2} + \text{const.} \tag{82}$$

Expanding and collecting terms in $x$:

$$\log \rho_{\text{tilt}}(x) = -\left(\lambda + \frac{1}{2\sigma_1^2}\right) x^2 + 2\left(\lambda a + \frac{\mu_1}{2\sigma_1^2}\right) x + \text{const} \tag{83}$$

$$= -\frac{1 + 2\lambda\sigma_1^2}{2\sigma_1^2} \left(x - \frac{2\lambda a \sigma_1^2 + \mu_1}{1 + 2\lambda\sigma_1^2}\right)^2 + \text{const.} \tag{84}$$

Reading off the Gaussian parameters gives (81). $\qquad\square$

Notably, reward tilting reduces variance (since $1 + 2\lambda\sigma_1^2 > 1$) and shifts the mean toward the reward maximum $a$.

### C.5.2. GREEDY GUIDANCE

As a prerequisite to analyzing greedy guidance, we first derive a closed-form expression for the flow map in the Gaussian setting.

**Lemma C.7** (Flow map for Gaussian targets). *For the linear interpolant $I_t = (1-t)x_0 + tx_1$ with $x_0 \sim \mathsf{N}(0,1)$ and $x_1 \sim \mathsf{N}(\mu_1, \sigma_1^2)$, the flow map is*

$$X_{t,1}(x) = \mu_1 + M_t(x - t\mu_1), \qquad M_t := \frac{\sigma_1}{\sqrt{C_t}}, \tag{85}$$

*where $C_t := (1-t)^2 + t^2\sigma_1^2$. The Jacobian is $\nabla X_{t,1}(x) = M_t$, which is state-independent.*

*Proof.* The probability flow velocity is given by (Albergo et al., 2023),

$$b_t(x) = \mu_1 + \frac{\dot{C}_t}{2C_t}(x - t\mu_1), \tag{86}$$

$$\dot{C}_t = -2(1-t) + 2t\sigma_1^2. \tag{87}$$

To find the flow map, we solve $\dot{x}_\tau = b_\tau(x_\tau)$ from time $t$ to time 1. Substituting $y_\tau := x_\tau - \tau\mu_1$ gives the linear ODE $\dot{y}_\tau = \frac{\dot{C}_\tau}{2C_\tau}y_\tau$, with solution

$$y_\tau = y_t \exp\left(\int_t^\tau \frac{\dot{C}_s}{2C_s}\,ds\right) = y_t \exp\left(\frac{1}{2}\log\frac{C_\tau}{C_t}\right) = y_t\sqrt{\frac{C_\tau}{C_t}}. \tag{88}$$

Transforming back to $x_\tau$ and evaluating at $\tau = 1$ with $C_1 = \sigma_1^2$ gives (85). $\qquad\square$

Using the flow map, we can now derive the greedy guidance in closed form.

**Proposition C.8** (Gaussian greedy guidance). *The greedy guidance* (8) *with constant* $\lambda$ *is*

$$u_t(x) = -2\lambda M_t^2(x - x_t^{\mathrm{M}}), \qquad x_t^{\mathrm{M}} := t\mu_1 + \frac{a - \mu_1}{M_t}. \tag{89}$$

*Proof.* Applying (8) with the flow map from Lemma C.7:

$$u_t(x) = \lambda M_t \nabla r(X_{t,1}(x)) = -2\lambda M_t\big(X_{t,1}(x) - a\big). \tag{90}$$

Substituting (85) and simplifying gives the result. $\qquad\square$

Since the greedy control is linear in $x$ and the initial distribution is Gaussian, the terminal distribution under guidance is also Gaussian. The following result, summarized as part of Proposition 2.5 in the main text, gives the mean and variance analytically.

**Proposition C.9** (Terminal distribution under the greedy guidance). *Under the greedy guidance* (89)*, the terminal distribution is* $\mathsf{N}(\mu_{\mathrm{greedy}}, \sigma_{\mathrm{greedy}}^2)$ *with the closed-form expressions*

$$\mu_{\mathrm{greedy}} = a + (\mu_1 - a)\,e^{-\pi\lambda\sigma_1}, \qquad \sigma_{\mathrm{greedy}}^2 = \sigma_1^2\,e^{-2\pi\lambda\sigma_1}. \tag{91}$$

*Proof.* Expanding the controlled dynamics $\dot{x}_t = b_t(x_t) + u_t(x_t)$ using (86) and (89) gives the linear system

$$\dot{x}_t = \tilde{A}_t x_t + f_t, \tag{92}$$

where $\tilde{A}_t := \frac{\dot{C}_t}{2C_t} - 2\lambda M_t^2$ and $f_t := \mu_1(1 - \frac{t\dot{C}_t}{2C_t}) + 2\lambda M_t^2 x_t^{\mathrm{M}}$. Since the dynamics are linear in $x$ and the initial condition is Gaussian, the terminal distribution is Gaussian. For the mean, consider the deviation $y_t := x_t - x_t^{\mathrm{M}}$ from the reference trajectory defined in (89). By definition, $x_t^{\mathrm{M}}$ is the state at time $t$ of the flow trajectory ending at $a$ at time 1, so that $x_t^{\mathrm{M}} = X_{1,t}(a)$. The Lagrangian equation (21) therefore gives $\dot{x}_t^{\mathrm{M}} = b_t(x_t^{\mathrm{M}})$. Subtracting the reference from (92) then yields

$$\dot{y}_t = \left(\frac{\dot{C}_t}{2C_t} - 2\lambda\frac{\sigma_1^2}{C_t}\right)y_t. \tag{93}$$

Integrating from 0 to 1 requires the two integrals $\int_0^1 \frac{\dot{C}_s}{2C_s}\,ds$ and $\int_0^1 \frac{\sigma_1^2}{C_s}\,ds$, which we will also use in subsequent proofs. The first is elementary. By the fundamental theorem of calculus,

$$\int_0^1 \frac{\dot{C}_s}{2C_s}\,ds = \tfrac{1}{2}\log(C_1/C_0) = \tfrac{1}{2}\log\sigma_1^2 = \log\sigma_1, \tag{94}$$

since $C_0 = 1$ and $C_1 = \sigma_1^2$. For the second, the substitution $z := s/(1-s)$ gives $ds = dz/(1+z)^2$ and $C_s = (1 + \sigma_1^2 z^2)/(1+z)^2$, so that $\sigma_1^2\,ds/C_s = \sigma_1^2\,dz/(1 + \sigma_1^2 z^2)$. As $s$ ranges over $[0,1]$, $z$ ranges over $[0,\infty)$, and

$$\int_0^1 \frac{\sigma_1^2}{C_s}\,ds = \sigma_1^2 \int_0^\infty \frac{dz}{1 + \sigma_1^2 z^2} = \sigma_1\arctan(\sigma_1 z)\Big|_0^\infty = \frac{\pi\sigma_1}{2}. \tag{95}$$

Using (94) and (95), we obtain $y_1 = \sigma_1 e^{-\pi\lambda\sigma_1} y_0$. Since $x_0 \sim \mathsf{N}(0,1)$ and $x_0^{\mathrm{M}} = (a - \mu_1)/\sigma_1$, we have $y_0 \sim \mathsf{N}\big(-(a - \mu_1)/\sigma_1,\, 1\big)$, so

$$y_1 \sim \mathsf{N}\big((\mu_1 - a)\,e^{-\pi\lambda\sigma_1},\, \sigma_1^2\,e^{-2\pi\lambda\sigma_1}\big). \tag{96}$$

Since $x_1 = a + y_1$, we obtain (91). $\qquad\square$

We next characterize the effect of early stopping on the greedy terminal variance. Anticipating the exact closed-form variance $\sigma^2_{\text{exact}} = \sigma_1^2/(1 + \pi\lambda\sigma_1)^2$ established in Proposition C.12 below, we show that early stopping can recover the rational-squared scaling of the exact optimal control.

**Proposition C.10** (Greedy variance under early stopping). *Applying greedy guidance only on $[0, t_{\text{stop}}]$ for $t_{\text{stop}} \in (0, 1]$ and integrating the uncontrolled flow on $[t_{\text{stop}}, 1]$ yields the terminal variance*

$$\sigma^2_{\text{greedy}}(t_{\text{stop}}) = \sigma_1^2 \exp\left(-4\lambda\sigma_1 \arctan\left(\frac{\sigma_1 t_{\text{stop}}}{1 - t_{\text{stop}}}\right)\right). \tag{97}$$

*Moreover, there exists a unique $t_{\text{stop}}(\lambda) \in (0, 1]$, depending continuously on $\lambda$, such that $\sigma^2_{\text{greedy}}(t_{\text{stop}}(\lambda)) = \sigma^2_{\text{exact}}$, with $t_{\text{stop}}(\lambda) \to 1$ as $\lambda \to 0$ and $t_{\text{stop}}(\lambda) \sim \log\lambda/\lambda$ as $\lambda \to \infty$.*

*Proof.* Truncating the variance integral at $t = t_{\text{stop}}$ in Proposition C.9 replaces the upper limit $z = \infty$ in (95) with $z_{\text{stop}} = t_{\text{stop}}/(1-t_{\text{stop}})$, which evaluates the arctangent at a finite argument and yields (97). The map $t_{\text{stop}} \mapsto \sigma^2_{\text{greedy}}(t_{\text{stop}})$ is continuous and monotone decreasing, with $\sigma^2_{\text{greedy}}(0) = \sigma_1^2 > \sigma^2_{\text{exact}}$ and $\sigma^2_{\text{greedy}}(1) = \sigma_1^2 e^{-2\pi\lambda\sigma_1} \leqslant \sigma^2_{\text{exact}}$ for $\lambda > 0$, so the intermediate value theorem yields a unique $t_{\text{stop}}(\lambda)$ with $\sigma^2_{\text{greedy}}(t_{\text{stop}}(\lambda)) = \sigma^2_{\text{exact}}$. For the asymptotics, setting $\sigma^2_{\text{greedy}}(t_{\text{stop}}) = \sigma^2_{\text{exact}}$ and taking logarithms gives

$$\arctan\left(\frac{\sigma_1 t_{\text{stop}}(\lambda)}{1 - t_{\text{stop}}(\lambda)}\right) = \theta(\lambda), \qquad \theta(\lambda) := \frac{\log(1 + \pi\lambda\sigma_1)}{2\lambda\sigma_1}, \tag{98}$$

so that $t_{\text{stop}}(\lambda) = \tan\theta(\lambda)/(\sigma_1 + \tan\theta(\lambda))$. As $\lambda \to 0$, $\log(1 + \pi\lambda\sigma_1) = \pi\lambda\sigma_1 + O(\lambda^2)$, so $\theta(\lambda) \to \pi/2$, $\tan\theta(\lambda) \to \infty$, and $t_{\text{stop}}(\lambda) \to 1$. As $\lambda \to \infty$, $\theta(\lambda) \sim \log(\pi\lambda\sigma_1)/(2\lambda\sigma_1) \to 0$, so $\tan\theta(\lambda) \sim \theta(\lambda)$ and $t_{\text{stop}}(\lambda) \sim \theta(\lambda)/\sigma_1 \sim \log\lambda/\lambda$. $\square$

### C.5.3. EXACT OPTIMAL CONTROL

In the Gaussian to Gaussian case with a quadratic reward, since the dynamics are linear and both the reward and control cost are quadratic, (5) is a Linear-Quadratic Regulator (LQR) problem with an analytical solution.

**Proposition C.11.** *The value function has the quadratic form $V_t(x) = \frac{1}{2}P_t(x - x_t^{\text{OC}})^2$, where $P_t$ satisfies the Riccati equation*

$$-\dot{P}_t = \frac{\dot{C}_t}{C_t}P_t - \lambda P_t^2, \qquad P_1 = 2, \tag{99}$$

*and the reference trajectory is $x_t^{\text{OC}} = x_t^{\text{M}}$ given by (89). The optimal control is*

$$u_t^*(x) = -\lambda P_t(x - x_t^{\text{OC}}), \tag{100}$$

*and the Riccati equation has the explicit solution $P_t = 1/Q_t$ where*

$$Q_t = C_t\left(\frac{1}{2\sigma_1^2} + \lambda\int_t^1 \frac{d\tau}{C_\tau}\right). \tag{101}$$

*Proof.* Since the drift $b_t$ is affine in $x$, the control cost is quadratic, and the terminal cost is quadratic, standard linear-quadratic regulator theory (Bardi & Capuzzo-Dolcetta, 1997) guarantees that the HJB equation admits a quadratic value function

$$V_t(x) = \frac{1}{2}P_t(x - \xi_t)^2, \tag{102}$$

parameterized by a gain $P_t$ and a reference trajectory $\xi_t$ that are both determined by matching the HJB equation

$$-\partial_t V_t(x) = b_t(x)\,\partial_x V_t(x) - \frac{\lambda}{2}(\partial_x V_t(x))^2, \qquad V_1(x) = (x - a)^2. \tag{103}$$

Expanding in powers of $(x - \xi_t)$, the terminal condition gives $\xi_1 = a$ and $P_1 = 2$, the quadratic coefficient gives the Riccati equation

$$-\dot{P}_t = \frac{\dot{C}_t}{C_t}\,P_t - \lambda\,P_t^2, \tag{104}$$

which is (99), and the linear coefficient gives the ODE $\dot{\xi}_t = b_t(\xi_t)$. The reference trajectory $\xi_t$ is thus the state at time $t$ of the flow trajectory ending at $a$, which by the same Lagrangian-equation argument used in Proposition C.9 equals the greedy target $x_t^{\mathrm{M}}$ defined in (89). Setting $x_t^{\mathrm{OC}} := \xi_t$ gives $x_t^{\mathrm{OC}} = x_t^{\mathrm{M}}$.

To solve the Riccati equation in closed form, substitute $Q_t := 1/P_t$ to obtain the linear ODE $\dot{Q}_t = \frac{\dot{C}_t}{C_t} Q_t - \lambda$ with $Q_1 = 1/2$. Writing $Q_t = C_t q_t$ absorbs the homogeneous part and yields $\dot{q}_t = -\lambda/C_t$, which integrates to $q_t = \frac{1}{2\sigma_1^2} + \lambda \int_t^1 \frac{d\tau}{C_\tau}$ using $q_1 = Q_1/C_1 = 1/(2\sigma_1^2)$. This proves (101), and the optimal control follows from $u_t^*(x) = -\lambda \partial_x V_t(x) = -\lambda P_t (x - x_t^{\mathrm{OC}})$. $\qquad\square$

The exact and greedy solutions contract toward the *same* target $x_t^{\mathrm{OC}} = x_t^{\mathrm{M}}$, differing only in the gain. A computation analogous to the greedy case (Proposition C.9) yields the closed-form terminal distribution under the exact optimal control.

**Proposition C.12** (Terminal distribution under the exact optimal control). *Under the exact optimal control (100), the terminal distribution is* $\mathsf{N}(\mu_{\mathrm{exact}}, \sigma_{\mathrm{exact}}^2)$ *with the closed-form expressions*

$$\mu_{\mathrm{exact}} = a + \frac{\mu_1 - a}{1 + \pi\lambda\sigma_1}, \qquad \sigma_{\mathrm{exact}}^2 = \frac{\sigma_1^2}{(1 + \pi\lambda\sigma_1)^2}. \tag{105}$$

*Proof.* The exact optimal control (100) is linear in $x$, so the terminal distribution is Gaussian. The deviation argument proceeds identically to the greedy case in Proposition C.9, with $y_t := x_t - x_t^{\mathrm{OC}}$ and $x_t^{\mathrm{OC}} = x_t^{\mathrm{M}}$ by Proposition C.11, yielding the linear ODE

$$\dot{y}_t = \left( \frac{\dot{C}_t}{2C_t} - \lambda P_t \right) y_t =: \tilde{A}_t^* \, y_t. \tag{106}$$

The contraction factor $e^{\int_0^1 \tilde{A}_t^* \, dt} = \sigma_1/(1 + \pi\lambda\sigma_1)$ follows from (94) together with

$$\lambda \int_0^1 P_t \, dt = \log(1 + \pi\lambda\sigma_1), \tag{107}$$

which we now derive. From (101), $P_t = 1/(C_t q_t)$ with $q_t = \frac{1}{2\sigma_1^2} + \lambda \int_t^1 \frac{d\tau}{C_\tau}$, so that $\dot{q}_t = -\lambda/C_t$ and $\lambda P_t = -\dot{q}_t/q_t = -\frac{d}{dt} \log q_t$. Integrating gives $\lambda \int_0^1 P_t \, dt = \log(q_0/q_1)$, and using $q_1 = 1/(2\sigma_1^2)$ together with $q_0 = \frac{1}{2\sigma_1^2} + \lambda \cdot \pi/(2\sigma_1) = \frac{1 + \pi\lambda\sigma_1}{2\sigma_1^2}$ (the integral $\int_0^1 d\tau/C_\tau = \pi/(2\sigma_1)$ is (95) divided by $\sigma_1^2$) yields (107). Since $x_0 \sim \mathsf{N}(0,1)$ and $x_0^{\mathrm{OC}} = x_0^{\mathrm{M}} = (a - \mu_1)/\sigma_1$ by Proposition C.11, we have $y_0 \sim \mathsf{N}\big(-(a - \mu_1)/\sigma_1, \, 1\big)$, so

$$y_1 \sim \mathsf{N}\left( \frac{\mu_1 - a}{1 + \pi\lambda\sigma_1}, \, \frac{\sigma_1^2}{(1 + \pi\lambda\sigma_1)^2} \right). \tag{108}$$

Since $x_1 = a + y_1$, we obtain (105). $\qquad\square$

### C.5.4. COMPARISON OF GUIDANCE SCHEMES

We now compare the three guidance schemes using the closed-form terminal distributions established above. With the means and variances of all three terminal distributions in hand (Proposition C.6, Proposition C.9, and Proposition C.12), we can collect the corresponding expected rewards in closed form.

**Corollary C.13** (Closed-form expected terminal rewards). *Under the setting of Propositions C.6, C.9 and C.12, the expected terminal rewards are*

$$\mathbb{E}[r(X_1^{\mathrm{tilt}})] = -\frac{\sigma_1^2(1 + 2\lambda\sigma_1^2) + (\mu_1 - a)^2}{(1 + 2\lambda\sigma_1^2)^2},$$

$$\mathbb{E}[r(X_1^{\mathrm{greedy}})] = -\big(\sigma_1^2 + (\mu_1 - a)^2\big) e^{-2\pi\lambda\sigma_1}, \tag{109}$$

$$\mathbb{E}[r(X_1^{\mathrm{exact}})] = -\frac{\sigma_1^2 + (\mu_1 - a)^2}{(1 + \pi\lambda\sigma_1)^2}.$$

*Proof.* For any Gaussian $X \sim \mathsf{N}(\mu, \sigma^2)$, the quadratic reward $r(x) = -(x-a)^2$ has expectation

$$\mathbb{E}[r(X)] = -\big(\mathrm{Var}(X) + (\mathbb{E}[X] - a)^2\big) = -\big(\sigma^2 + (\mu - a)^2\big). \tag{110}$$

Applying this identity to the closed-form means and variances from (81), (91) and (105) yields (109). □

With these results in hand, we now interpret and compare the three guidance schemes.

**Controls.** Both trajectory-based schemes produce linear-in-$x$ controls that drive trajectories toward a common reference path ending at the reward maximum $a$:

$$
\begin{aligned}
\text{Greedy:} \quad & u_t^{\mathrm{M}}(x) \propto -M_t^2\left(x - x_t^{\mathrm{M}}\right), & \quad & x_t^{\mathrm{M}} := t\mu_1 + \frac{a - \mu_1}{M_t}, \\
\text{Exact OC:} \quad & u_t^{\mathrm{OC}}(x) \propto -P_t\left(x - x_t^{\mathrm{OC}}\right), & \quad & x_t^{\mathrm{OC}} = x_t^{\mathrm{M}}.
\end{aligned}
\tag{111}
$$

The greedy and exact controls share the same reference trajectory, differing only in the gain: $M_t^2$ versus $P_t$.

**Variances.** The closed-form variances collected from Propositions C.6, C.9 and C.12 are

$$
\begin{aligned}
\text{Tilt:} \quad & \sigma_{\mathrm{tilt}}^2 = \frac{\sigma_1^2}{1 + 2\lambda\sigma_1^2} & \quad & \sim \frac{1}{\lambda}, \\
\text{Greedy:} \quad & \sigma_{\mathrm{greedy}}^2 = \sigma_1^2 e^{-2\pi\lambda\sigma_1} & \quad & \sim e^{-2\pi\lambda\sigma_1}, \\
\text{Exact OC:} \quad & \sigma_{\mathrm{exact}}^2 = \frac{\sigma_1^2}{(1 + \pi\lambda\sigma_1)^2} & \quad & \sim \frac{1}{\lambda^2},
\end{aligned}
\tag{112}
$$

where the asymptotics hold as $\lambda \to \infty$. Reward tilting concentrates at the polynomial rate $O(1/\lambda)$, exact optimal control at the faster $O(1/\lambda^2)$, and greedy guidance contracts exponentially. This exponential scaling arises because greedy guidance applies an instantaneous contraction rate proportional to the gain without accounting for future control effort, while the exact gain $P_t$ is softened by the Riccati equation. Although greedy and exact gains agree to leading order as $\lambda \to 0$ (cf. Proposition C.4), they diverge substantially for moderate and large $\lambda$, with greedy guidance collapsing more aggressively. The practical implication is that $\lambda$ must be scaled differently between the frameworks to achieve comparable behavior, a consideration relevant to prevent reward hacking. Figure 7 illustrates these differences.

**Expected reward.** The closed-form expected rewards from Corollary C.13 are

$$
\begin{aligned}
\text{Tilt:} \quad & \mathbb{E}[r(X_1^{\mathrm{tilt}})] = -\frac{\sigma_1^2(1 + 2\lambda\sigma_1^2) + (\mu_1 - a)^2}{(1 + 2\lambda\sigma_1^2)^2} & \quad & \sim -\frac{1}{\lambda}, \\
\text{Greedy:} \quad & \mathbb{E}[r(X_1^{\mathrm{greedy}})] = -(\sigma_1^2 + (\mu_1 - a)^2)e^{-2\pi\lambda\sigma_1} & \quad & \sim -e^{-2\pi\lambda\sigma_1}, \\
\text{Exact OC:} \quad & \mathbb{E}[r(X_1^{\mathrm{exact}})] = -\frac{\sigma_1^2 + (\mu_1 - a)^2}{(1 + \pi\lambda\sigma_1)^2} & \quad & \sim -\frac{1}{\lambda^2},
\end{aligned}
\tag{113}
$$

where the asymptotics hold as $\lambda \to \infty$. Greedy guidance achieves exponentially higher reward (closer to zero) compared to the polynomial rates for exact optimal control and reward tilting. This quantifies the reward-diversity trade-off: greedy guidance achieves higher expected reward at the cost of more aggressive variance reduction.

**Early stopping.** One practical strategy to mitigate variance collapse is *early stopping*: applying guidance only for $t \in [0, t_{\mathrm{stop}}]$ and then following the uncontrolled flow for $t \in [t_{\mathrm{stop}}, 1]$. In our numerical experiments in the main text, we found this immensely useful for reducing the effect of reward hacking. This truncates the contraction integral, leading to the closed-form expression in (97). Figure 8 demonstrates this effect: at early stopping times (e.g., $t_{\mathrm{stop}} = 0.3$), the greedy guidance maintains variance comparable to the target, while still shifting the mean toward the reward maximum. This provides a tunable trade-off between reward optimization and diversity preservation.

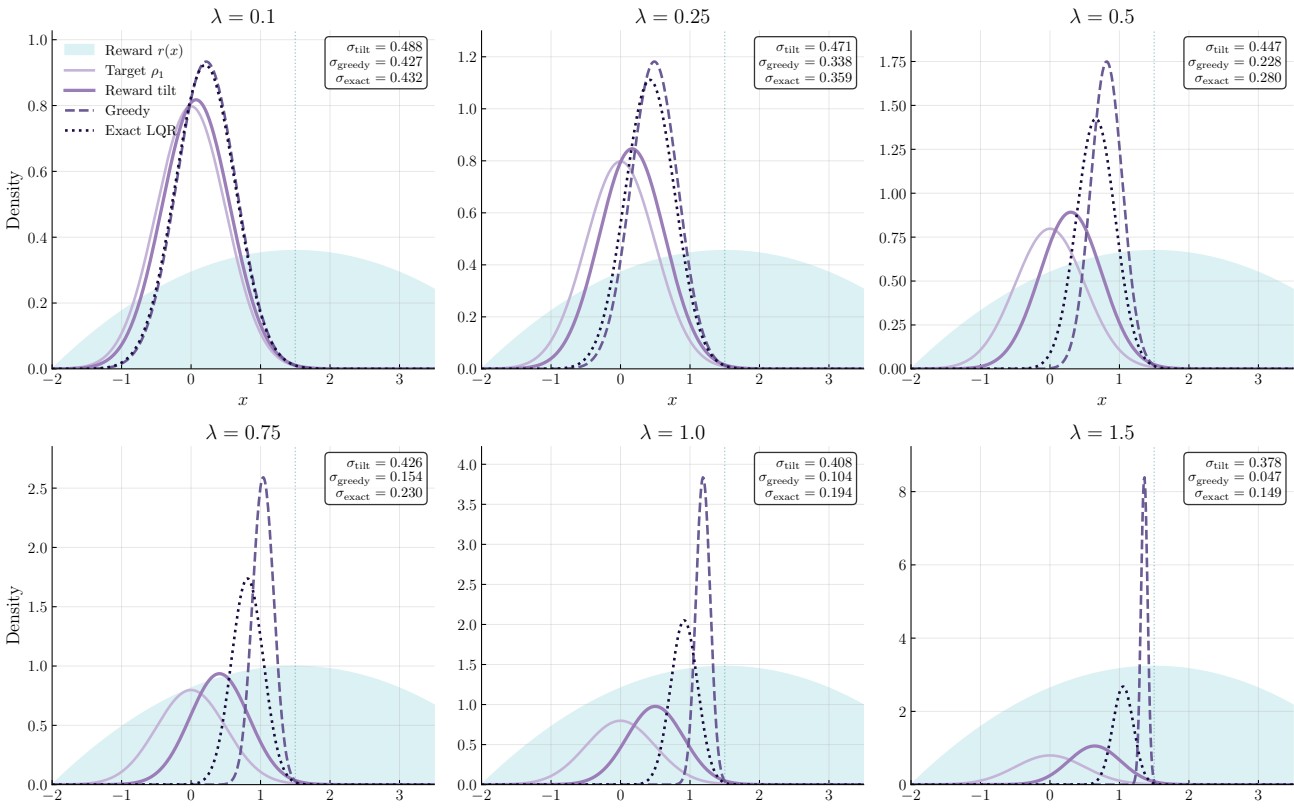

*Figure 7.* **Effect of guidance strength** $\lambda$ **on terminal distributions.** We compare the reward-tilted distribution (solid), greedy guidance (dashed), and exact LQR (dotted) for a Gaussian target $\rho_1 = \mathsf{N}(0, 0.5^2)$ with quadratic reward $r(x) = -(x - 1.5)^2$. As $\lambda$ increases, greedy guidance exhibits exponential variance collapse, while the exact LQR contracts at the slower rational-squared rate established in Proposition C.12. Both trajectory-based methods concentrate more aggressively than the reward tilt. These results highlight a need to carefully tune $\lambda$ for trajectory-level guidance, and that its choice requires a different scaling than the temperature parameter in typical exponential reward tilting.

### C.6. Variance recovery via early stopping

**Proposition C.14** (Variance recovery via early stopping)**.** *Under the setting of Proposition 2.5, there exists a unique* $t_{\text{stop}}(\lambda) \in (0, 1]$ *for which the terminal variance under greedy guidance on* $[0, t_{\text{stop}}(\lambda)]$ *followed by the uncontrolled flow on* $[t_{\text{stop}}(\lambda), 1]$ *matches* $\sigma_{\text{exact}}^2$.

The proof, along with the closed-form variance and the asymptotic behavior of $t_{\text{stop}}(\lambda)$, follows directly from the explicit greedy-guidance variance formula in (13) by setting the truncated trajectory variance equal to $\sigma_{\text{exact}}^2$ and inverting in $t_{\text{stop}}$; uniqueness follows from the monotonicity of the exponential factor $\exp(-2\pi\lambda\sigma_1 t_{\text{stop}})$ in $t_{\text{stop}}$.

### C.7. Proof of Proposition 2.4

**Proposition 2.4** (Tangent space projection)**.** *Suppose the data distribution* $\rho_1$ *is supported on a smooth manifold* $\mathcal{M} \subset \mathbb{R}^d$. *Let* $x_1 = X_{t,1}(x_t) \in \mathcal{M}$ *be the endpoint of the flow, and let* $T_{x_1}\mathcal{M}$ *denote the tangent space at* $x_1$. *Then, for any* $v \in \mathbb{R}^d$ *with* $v = v_\parallel + v_\perp$ *where* $v_\parallel \in T_{x_1}\mathcal{M}$ *and* $v_\perp \perp T_{x_1}\mathcal{M}$,

$$\nabla X_{t,1}(x_t)^\mathsf{T} v = \nabla X_{t,1}(x_t)^\mathsf{T} v_\parallel. \tag{12}$$

*Proof.* We first show that the image of the Jacobian is contained in the tangent space: $\text{Im}(\nabla X_{t,1}(x_t)) \subseteq T_{x_1}\mathcal{M}$. Fix an arbitrary direction $w \in \mathbb{R}^d$ with $\|w\| = 1$ and consider the one-parameter family $x_t + \epsilon w$ for $\epsilon \in \mathbb{R}$. Since $X_{t,1}$ maps $\mathbb{R}^d$ into $\mathcal{M}$, the curve $\gamma(\epsilon) := X_{t,1}(x_t + \epsilon w)$ lies entirely on $\mathcal{M}$ and passes through $x_1 = \gamma(0)$. Its tangent vector at $\epsilon = 0$ is therefore an element of the tangent space $T_{x_1}\mathcal{M}$, by the standard characterization of tangent vectors on a smooth manifold

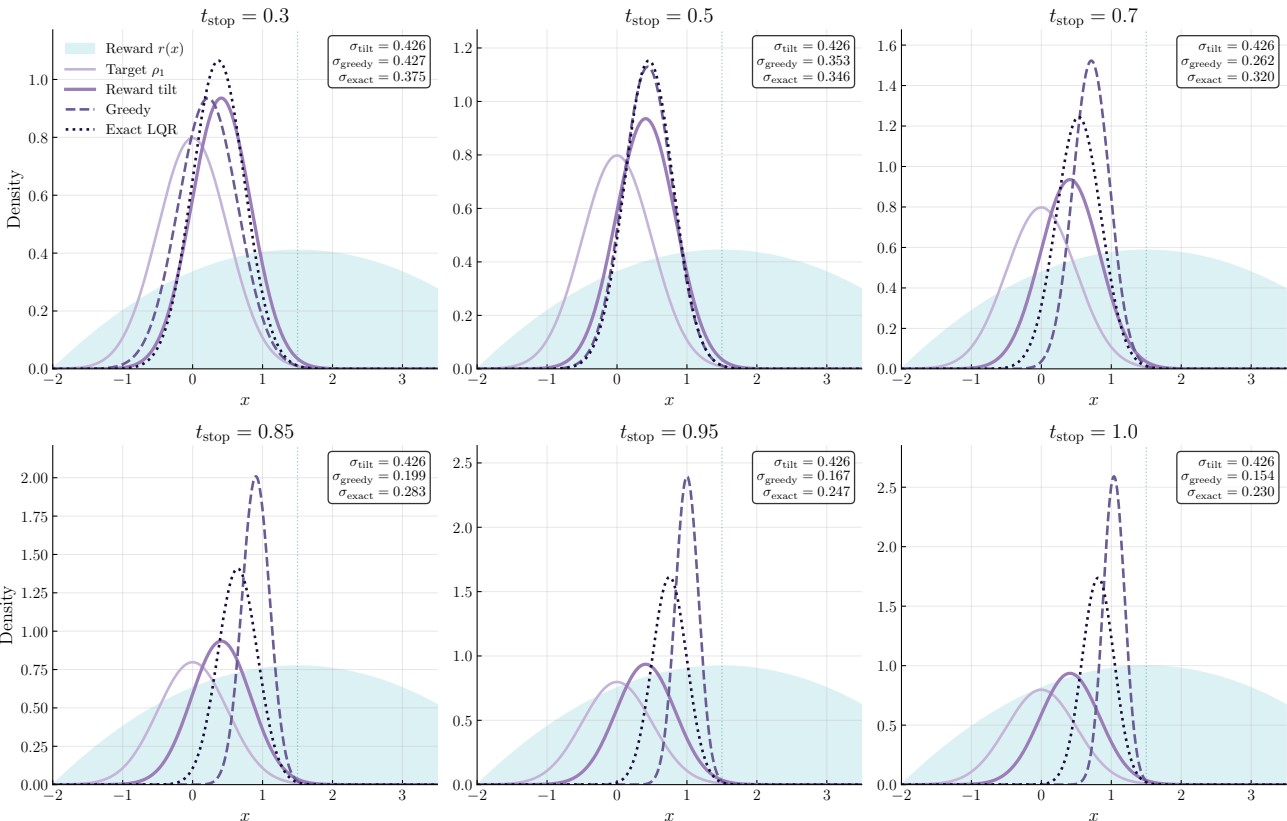

*Figure 8.* **Early stopping prevents variance collapse.** Same setup as Figure 7 with fixed $\lambda = 0.75$, but guidance is applied only until time $t_{\text{stop}}$, after which trajectories follow the uncontrolled flow. At $t_{\text{stop}} = 0.3$, greedy guidance preserves nearly all the target variance ($\sigma \approx 0.43$ vs $\sigma_1 = 0.5$) while still shifting the mean toward the reward. As $t_{\text{stop}} \to 1$, we recover the full-guidance behavior with significant variance collapse. This provides a practical mechanism to tune the reward–diversity trade-off.

as the velocities of smooth curves through the point. By the chain rule, this tangent vector is

$$\gamma'(0) = \nabla X_{t,1}(x_t)\, w. \tag{114}$$

Since $w$ was arbitrary and the image $\text{Im}(\nabla X_{t,1}(x_t))$ is spanned by $\{\nabla X_{t,1}(x_t)w \;:\; w \in \mathbb{R}^d\}$, we conclude $\text{Im}(\nabla X_{t,1}(x_t)) \subseteq T_{x_1}\mathcal{M}$.

Now, for any $v \in \mathbb{R}^d$ and $w \in \mathbb{R}^d$:

$$\langle \nabla X_{t,1}(x_t)^\mathsf{T} v, w \rangle = \langle v, \nabla X_{t,1}(x_t)w \rangle. \tag{115}$$

Since $\nabla X_{t,1}(x_t)w \in T_{x_1}\mathcal{M}$ for all $w$, this inner product depends only on the projection of $v$ onto $T_{x_1}\mathcal{M}$. Writing $v = v_\parallel + v_\perp$ with $v_\parallel \in T_{x_1}\mathcal{M}$ and $v_\perp \perp T_{x_1}\mathcal{M}$, we have

$$\langle v, \nabla X_{t,1}(x_t)w \rangle = \langle v_\parallel, \nabla X_{t,1}(x_t)w \rangle, \tag{116}$$

since the component $v_\perp$ is orthogonal to anything in $T_{x_1}\mathcal{M}$. Thus $\nabla X_{t,1}(x_t)^\mathsf{T} v = \nabla X_{t,1}(x_t)^\mathsf{T} v_\parallel$, completing the proof. $\qquad\square$

## D. Existing methods as special cases of FMRG

We now show that the FMRG framework provides a unifying lens through which many existing guidance methods can be understood as special cases or approximations. In Section 2, we showed that DPS arises as a coarse approximation to the greedy guidance (8) by replacing the flow map with a single Euler step. Here, we extend this analysis to several other prominent methods, summarized in Table 3, and provide detailed derivations showing how FlowDPS and FlowChef reduce to FMRG-E with different effective guidance weights.

*Table 3.* **Existing methods as special cases of FMRG.** Each method can be understood through three design choices: how the endpoint $X_{t,1}$ is approximated, how the base dynamics are integrated, and the guidance weight. Here $t_+ > t$ denotes the next timestep.

| Method | Endpoint | Dynamics | Jacobian? | Weight | Guidance at |
|---|---|---|---|---|---|
| FMRG-J (ours) | Flow map | Flow map | Yes | $\lambda_t \Delta t$ | Every step |
| FMRG-E (ours) | Flow map | Flow map | No | $\lambda_t \Delta t$ | Every step |
| DPS (Chung et al., 2024) | Euler step | Euler step | Yes | const | Every step |
| FlowDPS (Kim et al., 2025) | Euler step | Euler step | No | $(1-t)\,t_+$ | Every step |
| FlowChef (Patel et al., 2025) | Euler step | Euler step | No | const | Every step |
| MPGD (He et al., 2023) | Euler + proj. | Euler step | No | $t_+$ | Every step |
| ReNO (Eyring et al., 2024) | One-step gen. | — | Yes | — | $t = 0$ only |
| D-Flow (Ben-Hamu et al., 2024) | ODE solver | — | Yes | — | $t = 0$ only |

## D.1. DPS as a one-step approximation

We formalize the reduction summarized in Section 2 as the following corollary.

**Corollary D.1** (DPS as a single-step approximation of greedy guidance). *The DPS guidance signal*

$$u_t^{DPS}(x) = \lambda \left(I + (1-t)\,\nabla b_t(x)\right)^{\mathsf{T}} \nabla r\big(\hat{x}_1(x)\big),$$
$$\hat{x}_1(x) := \mathbb{E}[x_1 \mid x_t = x] = x + (1-t)\,b_t(x), \tag{117}$$

*is obtained from the greedy guidance (8) by replacing the exact flow-map endpoint $X_{t,1}(x)$ with the one-step Euler approximation $\hat{x}_1(x)$, and correspondingly approximating the Jacobian $\nabla X_{t,1}(x)^{\mathsf{T}}$ by its one-step Euler form $\left(I + (1 - t)\,\nabla b_t(x)\right)^{\mathsf{T}}$.*

We first give a brief review of DPS (Chung et al., 2024), which has been motivated as approximating (3) through two primary approximations. First, the expectation over trajectories is replaced by a *single* deterministic path. Second, the unknown endpoint $x_1$ is replaced by the posterior mean $\hat{x}_1 = \mathbb{E}[x_1 \mid x_t]$, also called the *denoiser*. Together, these yield the guidance scheme

$$\dot{x}_t = b_t(x_t) + \lambda_t(x_t)\nabla_{x_t} r\left(\hat{x}_1(x_t)\right), \quad \hat{x}_1(x_t) = \mathbb{E}\left[x_1 \mid x_t\right], \tag{118}$$

where $\lambda : [0, 1] \times \mathbb{R}^d \to \mathbb{R}$ is the guidance strength. While (118) has been observed to work well in practice and is efficient, these approximations are heuristic: there is no principled sense in which samples from (118) approximate the reward-tilted distribution $\tilde{\rho}_1$. Here, we prove that the posterior mean (denoiser) used in DPS-style guidance is a one-step approximation to the flow map, establishing that DPS emerges as a coarse approximation to the greedy guidance (8). That is, in the limit that the denoiser approximation is systematically refined, we recover (8).

We formalize the connection between the denoiser used in DPS and the flow-map endpoint in the following proposition, which shows that the posterior mean is determined by a single evaluation of the probability-flow velocity, and coincides with the one-step Euler approximation of the flow map under the linear interpolant.

**Proposition D.2** (Denoiser as a one-step flow-map approximation). *Let $I_t = \alpha_t x_0 + \beta_t x_1$ be a stochastic interpolant with boundary conditions $\alpha_0 = \beta_1 = 1$ and $\alpha_1 = \beta_0 = 0$, and let $b_t(x) = \mathbb{E}[\dot{I}_t \mid I_t = x]$ denote the associated probability-flow velocity. Then the posterior mean $\hat{x}_1(x) := \mathbb{E}[x_1 \mid I_t = x]$ admits the closed form*

$$\hat{x}_1(x) = \frac{\alpha_t\,b_t(x) - \dot{\alpha}_t\,x}{\alpha_t\dot{\beta}_t - \dot{\alpha}_t\beta_t}. \tag{119}$$

*In particular, for the linear interpolant ($\alpha_t = 1 - t$, $\beta_t = t$), the posterior mean reduces to the one-step Euler approximation of the flow map,*

$$\hat{x}_1(x) = x + (1 - t)\,b_t(x) =: X_{t,1}^{\mathrm{Euler}}(x). \tag{120}$$

Proposition D.2 makes precise the sense in which DPS-style guidance is a coarse approximation of the greedy guidance (8): both signals backpropagate $\nabla r$ through an estimate of the endpoint $X_{t,1}(x_t)$, but DPS uses the posterior mean, which by (120) coincides with a *single* Euler step of the pre-trained velocity, while FMRG uses the full flow map, which is the limit of infinitely many Euler steps. The derivation of (119) and (120) is standard (see, e.g., (Albergo et al., 2023)); we include it here for completeness.

*Proof.* By definition of the interpolant, $\dot{I}_t = \dot{\alpha}_t x_0 + \dot{\beta}_t x_1$, so that

$$b_t(x) = \mathbb{E}[\dot{I}_t \mid I_t = x] = \dot{\alpha}_t \, \hat{x}_0(x) + \dot{\beta}_t \, \hat{x}_1(x), \tag{121}$$

where $\hat{x}_0(x) := \mathbb{E}[x_0 \mid I_t = x]$ and $\hat{x}_1(x) := \mathbb{E}[x_1 \mid I_t = x]$. Taking conditional expectations of the interpolant identity $I_t = \alpha_t x_0 + \beta_t x_1$ also gives the constraint $\alpha_t \, \hat{x}_0(x) + \beta_t \, \hat{x}_1(x) = x$, which eliminates $\hat{x}_0$ in favor of $\hat{x}_1$:

$$\hat{x}_0(x) = \frac{x - \beta_t \, \hat{x}_1(x)}{\alpha_t}. \tag{122}$$

Substituting back into the expression for $b_t(x)$ yields

$$b_t(x) = \frac{\dot{\alpha}_t}{\alpha_t} \, x + \left( \dot{\beta}_t - \frac{\dot{\alpha}_t \, \beta_t}{\alpha_t} \right) \hat{x}_1(x), \tag{123}$$

and solving for $\hat{x}_1(x)$ gives (119). For the linear interpolant, $\dot{\alpha}_t = -1$ and $\dot{\beta}_t = 1$, so the denominator in (119) equals $\alpha_t \dot{\beta}_t - \dot{\alpha}_t \beta_t = (1 - t) + t = 1$, and the numerator simplifies to $(1 - t) \, b_t(x) + x$, yielding (120). $\square$

### D.2. FlowDPS as an approximation of FMRG-E

FlowDPS (Kim et al., 2025) is a guidance scheme for inverse problems built on top of a pre-trained flow model. We show here that, when rewritten in our convention ($t = 0$ noise, $t = 1$ data, linear interpolant $I_t = (1 - t)x_0 + tx_1$), it is recovered from FMRG-E by replacing both flow map velocities that appear in the operator-splitting update (16) with the pre-trained instantaneous velocity $b_t = v_{t,t}$.

**FlowDPS formulation.** FlowDPS treats the noisy state $x_t$ as a convex combination of a data prediction and a noise prediction obtained from Tweedie's formula:

$$\hat{x}_1 = x_t + (1 - t) \, b_t(x_t), \qquad \hat{x}_0 = x_t - t \, b_t(x_t), \tag{124}$$

where $b_t = v_{t,t}$ is the instantaneous velocity of the pre-trained flow (24). By Proposition D.2, the data prediction $\hat{x}_1 = x_t + (1 - t) \, b_t(x_t)$ is exactly the one-step Euler estimate of the flow-map endpoint $X_{t,1}(x_t) = x_t + (1 - t) \, v_{t,1}(x_t)$, obtained by freezing the velocity at the current time $t$. The guided data prediction is obtained by blending the Tweedie estimate with a reward-optimized estimate $\hat{x}_1^{\mathrm{opt}}$ that is computed via gradient descent on the measurement loss applied to $\hat{x}_1$:

$$\tilde{x}_1 = t \, \hat{x}_1 + (1 - t) \, \hat{x}_1^{\mathrm{opt}}. \tag{125}$$

Finally, the deterministic FlowDPS update to the next timestep $t_+ > t$ recomposes $\tilde{x}_1$ and $\hat{x}_0$ using the linear interpolant at time $t_+$:

$$x_{t_+} = t_+ \, \tilde{x}_1 + (1 - t_+) \, \hat{x}_0. \tag{126}$$

**Reduction to FMRG-E.** We now unpack the FlowDPS update (126) to exhibit its relationship with the FMRG-E operator splitting (16). In the unguided case $\hat{x}_1^{\mathrm{opt}} = \hat{x}_1$, the update reduces to the standard Euler step $x_{t_+} = x_t + \Delta t \cdot b_t(x_t)$, since $t\hat{x}_1 + (1 - t)\hat{x}_1 = \hat{x}_1$ and $t_+\hat{x}_1 + (1 - t_+)\hat{x}_0 = x_t + \Delta t \, b_t(x_t)$. The guidance correction is therefore the extra term produced by $\hat{x}_1^{\mathrm{opt}} - \hat{x}_1$:

$$x_{t_+}^{\mathrm{guided}} - x_{t_+}^{\mathrm{unguided}} = t_+(1 - t)(\hat{x}_1^{\mathrm{opt}} - \hat{x}_1). \tag{127}$$

For a single gradient step of size $\eta$ on the FlowDPS measurement loss (with $r$ taken as the negative loss), we have $\hat{x}_1^{\mathrm{opt}} - \hat{x}_1 = -\eta \, \nabla_{\hat{x}_1} r(\hat{x}_1)$, so that the full FlowDPS update becomes

$$x_{t_+} = x_t + \Delta t \cdot b_t(x_t) - (1 - t) \, t_+ \, \eta \, \nabla_{\hat{x}_1} r(\hat{x}_1). \tag{128}$$

Comparing term by term with the FMRG-E operator splitting (16), the FlowDPS update (128) is obtained from FMRG-E by two substitutions of the learned flow map by its one-step Euler approximation: the flow-map step velocity $v_{t,t_+}$ is replaced with the diagonal velocity $v_{t,t}$, and the endpoint-lookahead velocity $v_{t_+,1}$ is likewise replaced with $v_{t,t}$.

### D.3. FlowChef as an approximation of FMRG-E

**FlowChef formulation.** FlowChef (Patel et al., 2025) uses the same one-step Euler estimate of the flow-map endpoint, $\hat{x}_1 = x_t + (1-t)\, b_t(x_t)$, and applies a direct gradient correction to the Euler integration of the base flow (their Theorem 4.3):

$$x_{t_+} = x_t + \Delta t \cdot b_t(x_t) - s'\, \nabla_{\hat{x}_1} r(\hat{x}_1), \tag{129}$$

where $s'$ is a constant guidance scale. The gradient term in (129) is the reward gradient computed at the endpoint estimate $\hat{x}_1$ rather than at the current state $x_t$. FlowChef obtains this form by a procedure they term *gradient skipping*: their Lemma 4.2 derives the chain-rule expansion $\nabla_{x_t} r(\hat{x}_1) = \big(I + (1-t)\, J_{b_t}(x_t)\big)^{\mathsf{T}} \nabla_{\hat{x}_1} r(\hat{x}_1)$, where $J_{b_t}(x_t)$ denotes the Jacobian of the instantaneous velocity with respect to the state. The Jacobian factor captures how a perturbation at $x_t$ propagates forward to $\hat{x}_1$ through the Euler step, but FlowChef discards this factor entirely and uses only the terminal gradient $\nabla_{\hat{x}_1} r(\hat{x}_1)$ in the update.

**Reduction to FMRG-E.** As with FlowDPS, (129) is obtained from FMRG-E by the same two replacements of flow-map velocities with instantaneous velocities: $v_{t,t_+} \to b_t$ for the stepping term, and $v_{t_+,1} \to b_t$ for the endpoint lookahead used in the reward gradient.

### D.4. MPGD as an approximation of FMRG-E

Manifold Preserving Guided Diffusion (MPGD) (He et al., 2023) was originally formulated for DDIM-based diffusion models. Below, we cast it to the flow setting and show that it reduces to the same form as FlowDPS and FlowChef.

**MPGD formulation in the diffusion setting.** MPGD's key "shortcut" (their Equations 7–8) updates the Tweedie clean estimate $x_{0|t}$ directly rather than the noisy state $x_t$. Concretely, a single MPGD step takes the form

$$x_{0|t} \leftarrow x_{0|t} - c_t\, \nabla_{x_{0|t}} r(x_{0|t}), \tag{130}$$

$$x_{t-1} = \sqrt{\bar{\alpha}_{t-1}}\, x_{0|t} + \sqrt{1 - \bar{\alpha}_{t-1}}\, \epsilon_\theta(x_t, t), \tag{131}$$

where $c_t$ is a step size and the Tweedie estimate is $x_{0|t} = \frac{1}{\sqrt{\bar{\alpha}_t}}\big(x_t - \sqrt{1 - \bar{\alpha}_t}\, \epsilon_\theta(x_t, t)\big)$. The first line of (130) performs a reward-gradient descent on the clean estimate, and the second line re-noises $x_{0|t}$ back to the previous diffusion step $t-1$ using the learned noise prediction $\epsilon_\theta$. MPGD has two variants. The default variant, which we call MPGD without projection, applies the gradient update directly to $x_{0|t}$. A second variant, MPGD-AE, additionally passes $x_{0|t}$ through an autoencoder $D \circ E$ before computing the gradient, with the intent of enforcing that the gradient is computed at an on-manifold point. In what follows we focus on the variant without projection, since it provides the cleanest comparison to FMRG-E.

**Casting MPGD to a flow model.** We translate the MPGD update to the deterministic flow setting. For the linear interpolant, the Tweedie estimate becomes $\hat{x}_1 = x_t + (1-t)\, b_t(x_t)$ and the DDIM renoising reduces to an Euler step of the probability flow. After the reward-gradient update (130), the guided data estimate is

$$\hat{x}_1^{\text{guided}} = \hat{x}_1 - c_t\, \nabla_{\hat{x}_1} r(\hat{x}_1). \tag{132}$$

Recomposing $\hat{x}_1^{\text{guided}}$ with the noise prediction $\hat{x}_0 = x_t - t\, b_t(x_t)$ at the next timestep $t_+$ yields

$$x_{t_+} = t_+\, \hat{x}_1^{\text{guided}} + (1 - t_+)\, \hat{x}_0 = \underbrace{x_t + \Delta t \cdot b_t(x_t)}_{\text{Euler step}} - t_+\, c_t\, \nabla_{\hat{x}_1} r(\hat{x}_1). \tag{133}$$

This has the same structure as the FlowDPS and FlowChef updates (128) and (129), with an effective guidance weight $w_t^{\text{MPGD}} = t_+\, c_t$. For a constant MPGD step size $c_t$, the weight grows linearly in $t_+$, in contrast to the FlowDPS weight $w_t = (1-t)\, t_+$ and the constant FlowChef weight $w_t = s'$.

**Relation to FMRG.** Like FlowDPS and FlowChef, MPGD without projection makes the same two flow-map approximations, replacing $v_{t,t_+}$ with the diagonal velocity $b_t$ in the stepping term and $v_{t_+,1}$ with $b_t$ in the endpoint-lookahead term. MPGD-AE addresses the off-manifold issue at the endpoint using an explicit autoencoder projection. FMRG addresses the same issue more naturally. FMRG-E uses the on-manifold flow-map endpoint $X_{t,1}(x_t)$ instead of the off-manifold denoiser, and FMRG-J further projects reward gradients onto the tangent space via the flow-map Jacobian $\nabla X_{t,1}^{\mathsf{T}}$ (Proposition 2.4).

### D.5. Summary of approximations

Recall from (23) that the flow map is parameterized as $X_{s,t}(x) = x + (t - s)\, v_{s,t}(x)$. A single FMRG-E step ((16), Algorithm 1) uses the flow map in two ways:

1. **Step**: $\tilde{x}_{t_+} = X_{t,t_+}(x_t) = x_t + \Delta t\, v_{t,t_+}(x_t)$,

2. **Lookahead**: $\hat{x}_1 = X_{t_+,1}(\tilde{x}_{t_+}) = \tilde{x}_{t_+} + (1 - t_+)\, v_{t_+,1}(\tilde{x}_{t_+})$.

FlowDPS, FlowChef, and MPGD replace *both* off-diagonal velocities with $b_t = v_{t,t}$:

1. **Euler step**: $v_{t,t_+} \to b_t$,    so $\tilde{x}_{t_+} \approx x_t + \Delta t\, b_t(x_t)$,

2. **Denoiser**: $v_{t_+,1} \to b_t$,    so $\hat{x}_1 \approx x_t + (1 - t)\, b_t(x_t)$.

Both approximations reduce to the same operation: *replacing the learned off-diagonal flow map velocity $v_{s,t}$ with the instantaneous velocity $b_t = v_{t,t}$ ((24))*. FMRG avoids this approximation entirely by using a pre-trained flow map, which provides exact trajectory integration and on-manifold endpoint predictions, enabling effective guidance in as few as 3 NFEs.

**Guidance weight.**    In addition to the flow map approximation, the effective guidance weight $w_t$ differs:

- **FlowDPS** (deterministic): $w_t = (1 - t) \cdot t_+$.

- **FlowChef**: $w_t = s'$ (constant).

- **MPGD**: $w_t = t_+ \cdot c_t$.

- **FMRG-E**: $w_t = \lambda \cdot \Delta t$, from the optimal control formulation (Proposition 2.3).

### D.6. ReNO and D-Flow

ReNO (Eyring et al., 2024) and D-Flow (Ben-Hamu et al., 2024) take a different approach than the methods discussed above. Rather than guiding at each step, they optimize the initial noise seed $x_0$ to maximize $r(G(x_0))$, where $G$ denotes the full generative process. In our framework, this corresponds to applying FMRG guidance exclusively at $t = 0$, recovering the seed-optimization special case noted in Section 3 (where we highlight that taking multiple gradient steps before the first flow step recovers seed optimization). The two methods differ in how they evaluate the terminal gradient $\nabla_{x_0} r(X_{0,1}(x_0))$. ReNO uses a one-step generator as a fast approximation to the flow map, while D-Flow backpropagates through a multi-step ODE solver. Both methods are therefore instances of FMRG-J restricted to $t = 0$ that approximate the flow map $X_{0,1}$ by different mechanisms, computing $\nabla X_{0,1}(x_0)^\mathsf{T} \nabla r(X_{0,1}(x_0))$ under each respective approximation. The key limitation of this family of approaches is that they cannot correct the trajectory during generation, so that all guidance must be front-loaded into the initial condition.

## E. Experimental details

### E.1. Flow map model

All experiments use a flow map distilled from FLUX.1-Dev (Black Forest Labs, 2024) via Lagrangian distillation (Boffi et al., 2025b), parameterized as a LoRA adapter (rank 16, alpha 16) trained for 43,000 steps. The pre-trained checkpoint is publicly available at https://huggingface.co/gabeguofanclub/flux-1-dev-flowmap-lsd. All methods use FLUX.1's embedded guidance scale of 3.5. We follow the convention $t = 0$ (noise) to $t = 1$ (data) throughout.

### E.2. Reinitialization

Because greedy optimization depends on the initial noise sample, we may run a small number of independent reinitializations. We find at most 4 suffice even for complex neural-network-based rewards. Each reinitialization triggers a fresh single-trajectory guided run, in contrast to SMC, which maintains many particles with resampling along the trajectory.

*Table 4.* **Method parameters for inverse problems.**

| Method | Grad. target | Velocity | $n_{\mathrm{opt}}$ | Grad. norm | Early stop |
|---|---|---|---|---|---|
| FMRG-E | $z_0$ | flow map | 5 | No | $t_{\mathrm{stop}} = 0.67$ |
| FMRG-J | $z_t$ | flow map | 1 | Yes | No |
| FlowDPS | $z_0$ | instantaneous | 5 | No | No |
| FlowChef | $z_0$ | instantaneous | 5 | No | No |
| DPS | $z_t$ | instantaneous | 1 | No | No |

### E.3. Practical implementation

We define the key algorithmic parameters that appear in the per-task configurations below. $n_{\mathrm{opt}}$ denotes the number of inner gradient steps per guided step (see (17)). $t_{\mathrm{stop}} \in (0, 1]$ is the early stopping time: guidance is applied for $t \in [0, t_{\mathrm{stop}}]$, after which the remaining trajectory is completed in a single unguided flow map step $X_{t_{\mathrm{stop}},1}$ (smaller $t_{\mathrm{stop}}$ = earlier stopping). $\eta$ is the step size, which enters through $\lambda_t$ as described below.

**Flow map stepping.** Each guided step requires evaluating the flow map endpoint $X_{t,1}(x_t)$ for the reward gradient. As described in Section 3, stepping can use either a separate evaluation $X_{t,t_+}(x_t)$ (2 NFEs per step) or reuse the endpoint $X_{t,1}(x_t)$ for both the reward gradient and stepping (1 NFE per step). We find that reusing the endpoint performs better at low NFEs, where the savings allow more guided steps within the same budget. At higher NFEs, the approximation error accumulates and separate evaluations perform better. In practice, we use 1 NFE per step for low-NFE configurations and 2 NFEs per step for high-NFE configurations; the specific choice is noted in each task's configuration table.

**Guidance strength $\lambda_t$.** Recall from Algorithm 1 that each gradient update takes the form $x \leftarrow x + \lambda_t \cdot u$, where $u$ is the guidance signal. In practice, we set $\lambda_t$ as follows; $\eta$ is the single tuned constant in each case.

**FMRG-E.** We set $\lambda_t = \eta \cdot t \cdot (1 - t_+)$, where $t_+$ is the next timestep. This weighting, which downweights the update near the endpoints, works well in practice.

**FMRG-J.** We normalize the guidance signal to unit norm and rescale to the flow map velocity magnitude, giving $\tilde{u} = (u/\|u\|) \cdot \|v_{t,t_+}(x)\|$, and set $\lambda_t = \eta \cdot \Delta t$. This keeps $\eta$ invariant to the choice of reward model, which is important in practice since different rewards (e.g., human preference models vs. $\ell_2$ losses) exhibit vastly different gradient scales.

### E.4. Latent-space inverse problems

**Reward.** The reward is the negative $\ell_2$ reconstruction loss $r(x) = -\|Ax - y\|^2$, where $A \in \mathbb{R}^{d_o \times d}$ is the degradation operator and $y \in \mathbb{R}^{d_o}$ is the observation. Since the generative model operates in a latent space of dimension $d_e < d$, we decode the latent via the VAE decoder $D : \mathbb{R}^{d_e} \to \mathbb{R}^d$ and apply $A$ in pixel space:

$$r(z) = -\|A\,D(z) - y\|^2. \tag{134}$$

The degradation operators we consider are $4\times$ average-pooling for super-resolution, a $61 \times 61$ motion blur kernel with intensity 0.5 for deblurring, and a centered $64 \times 64$ binary mask for inpainting. Gaussian noise with $\sigma = 0.03$ is added to all observations.

**Datasets.** We evaluate on AFHQ-Dog (Choi et al., 2020) and FFHQ (Karras et al., 2019), using 1000 images at $256 \times 256$ resolution for each. Text prompts are "a photo of a dog" for AFHQ and "a photo of a person" for FFHQ.

**Method configurations.** Table 4 summarizes the method-specific parameters. FlowDPS and FlowChef compute gradients with respect to $z_0$ using the instantaneous velocity from the base flow model rather than the flow map. For FMRG-E at NFE $\leqslant 30$, we reuse the endpoint evaluation $X_{t,1}$ for stepping; at higher NFEs we use separate evaluations for lookahead and stepping (Section 3).

**Hyperparameter selection.** Step sizes, number of steps, and $n_{\mathrm{opt}}$ are selected via grid search on a held-out set of 50 images, with the best configuration evaluated on 1000 separate images. For FMRG-E and FMRG-J, we fix the number of

*Table 5.* **Selected hyperparameters for inverse problems.**

| Task | Method | AFHQ | | | | FFHQ | | | |
|---|---|---|---|---|---|---|---|---|---|
| | | NFE | $\eta$ | $n_{\text{opt}}$ | Steps | NFE | $\eta$ | $n_{\text{opt}}$ | Steps |
| SR | DPS | 200 | 10.0 | 1 | 200 | 200 | 50.0 | 1 | 200 |
| | FlowChef | 100 | 1.0 | 5 | 100 | 100 | 1.0 | 3 | 100 |
| | FlowDPS | 200 | 100.0 | 1 | 200 | 100 | 50.0 | 1 | 100 |
| | FMRG-E | 100 | 6 | 5 | 50 | 100 | 6 | 5 | 50 |
| | FMRG-J | 400 | 5.0 | 1 | 200 | 400 | 3.0 | 1 | 200 |
| Deblur | DPS | 100 | 10.0 | 1 | 100 | 200 | 200.0 | 1 | 200 |
| | FlowChef | 200 | 3.0 | 5 | 200 | 200 | 3.0 | 5 | 200 |
| | FlowDPS | 200 | 50.0 | 1 | 200 | 200 | 50.0 | 1 | 200 |
| | FMRG-E | 100 | 6 | 5 | 50 | 100 | 6 | 5 | 50 |
| | FMRG-J | 400 | 5.0 | 1 | 200 | 400 | 3.0 | 1 | 200 |
| Inpaint | DPS | 200 | 50.0 | 1 | 200 | 200 | 200.0 | 1 | 200 |
| | FlowChef | 200 | 10.0 | 1 | 200 | 100 | 50.0 | 5 | 100 |
| | FlowDPS | 200 | 200.0 | 1 | 200 | 200 | 200.0 | 1 | 200 |
| | FMRG-E | 100 | 6 | 5 | 50 | 100 | 6 | 5 | 50 |
| | FMRG-J | 400 | 5.0 | 1 | 200 | 400 | 3.0 | 1 | 200 |

*Table 6.* **Early stopping ablation (inverse problems).** LPIPS↓ / KID↓ on super-resolution for FMRG-E.

| NFE | AFHQ | | FFHQ | |
|---|---|---|---|---|
| | No ES | ES | No ES | ES |
| 6 | .228 / .010 | .231 / .009 | .248 / .049 | .234 / .047 |
| 12 | .220 / .011 | .204 / .006 | .217 / .050 | .191 / .043 |
| 30 | .206 / .015 | .184 / .008 | .197 / .059 | .180 / .046 |

steps to 50 and 200 respectively, and search only over step sizes. Table 5 reports the selected hyperparameters for each method and task.

**Early stopping ablation.** Table 6 shows the effect of early stopping on FMRG-E for super-resolution. Early stopping helps at low NFEs but has diminishing returns at higher NFEs.

**Text prompting ablation.** We ablate the effect of the text prompt on inverse problem performance. FMRG exhibits minimal degradation when varying or removing the text prompt, indicating robustness to prompt choice. In contrast, FlowDPS and FlowChef degrade noticeably. Figures 11 and 12 show full metrics when the text prompt is removed entirely (unconditional generation): the performance gap between FMRG and the baselines widens substantially in this setting, with FlowDPS and FlowChef degrading significantly while FMRG remains robust.

**Full metric plots and qualitative results.** Figures 9 and 10 show full metrics across NFEs for AFHQ and FFHQ with text-conditional generation. Figures 13 to 15, 17 and 18 show non-cherry-picked qualitative comparisons on randomly selected images.

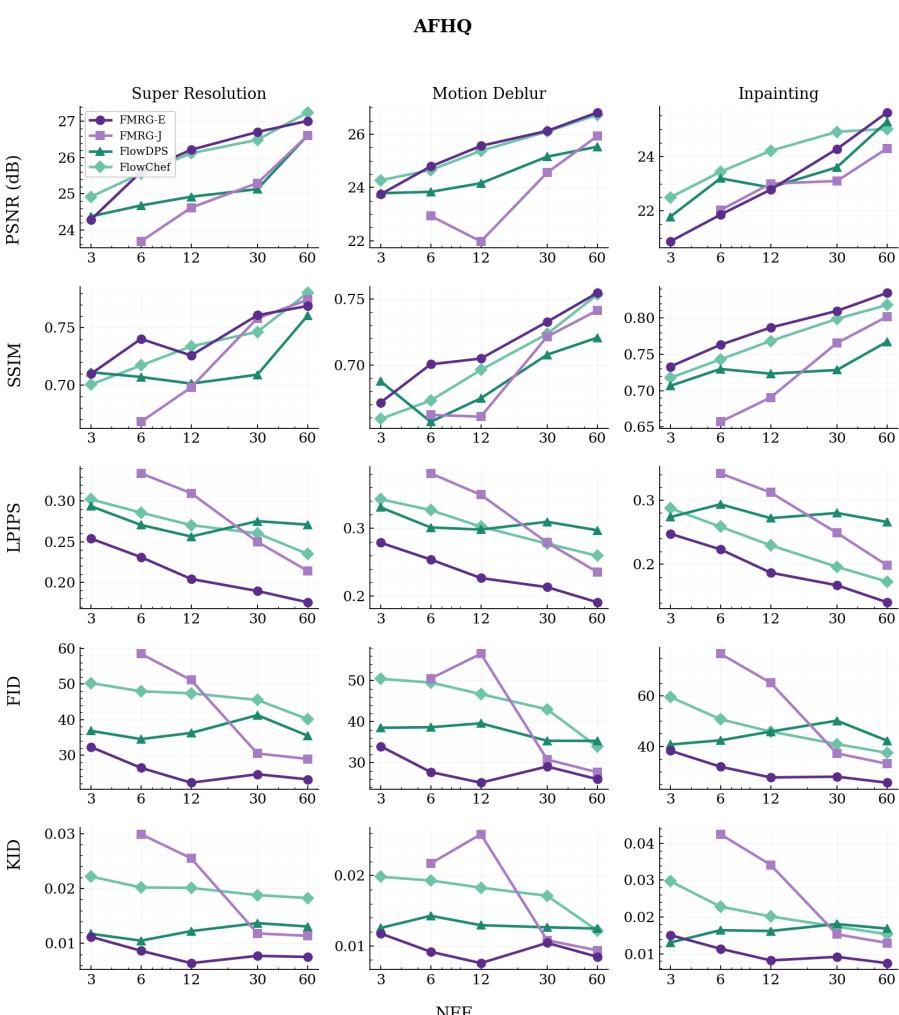

*Figure 9.* **Full AFHQ results across NFEs.** FMRG-E and FMRG-J dominate the distortion and perception metrics at every NFE budget, with the gap largest in the low-NFE regime.

**FFHQ**

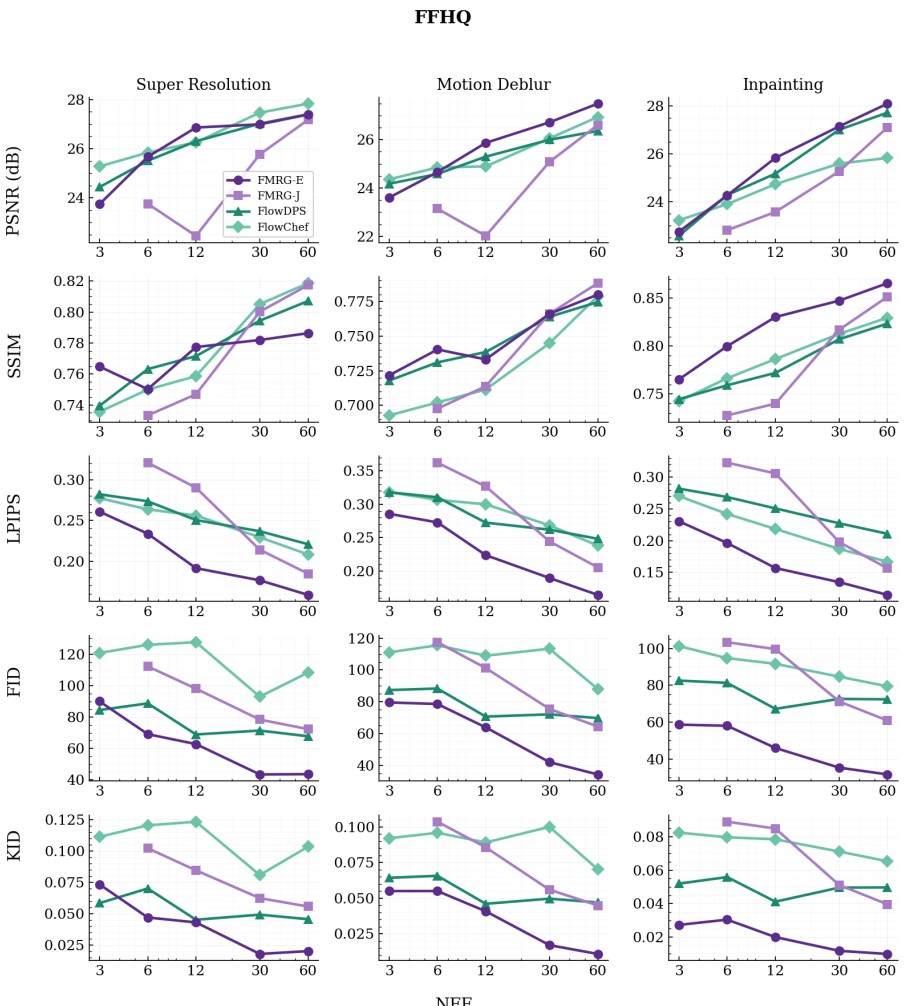

*Figure 10.* **Full FFHQ results across NFEs.** Same pattern as Figure 9: FMRG variants match or outperform all baselines across metrics at every NFE budget.

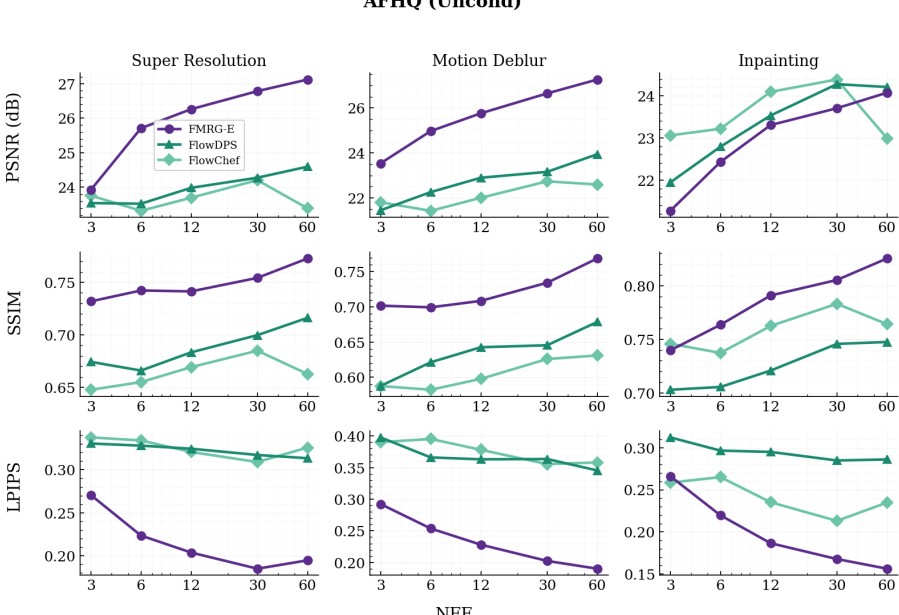

*Figure 11.* **AFHQ results under unconditional (prompt-free) generation.** Removing the text prompt widens the gap between FMRG and the baselines: FlowDPS and FlowChef degrade sharply, while FMRG variants remain robust.

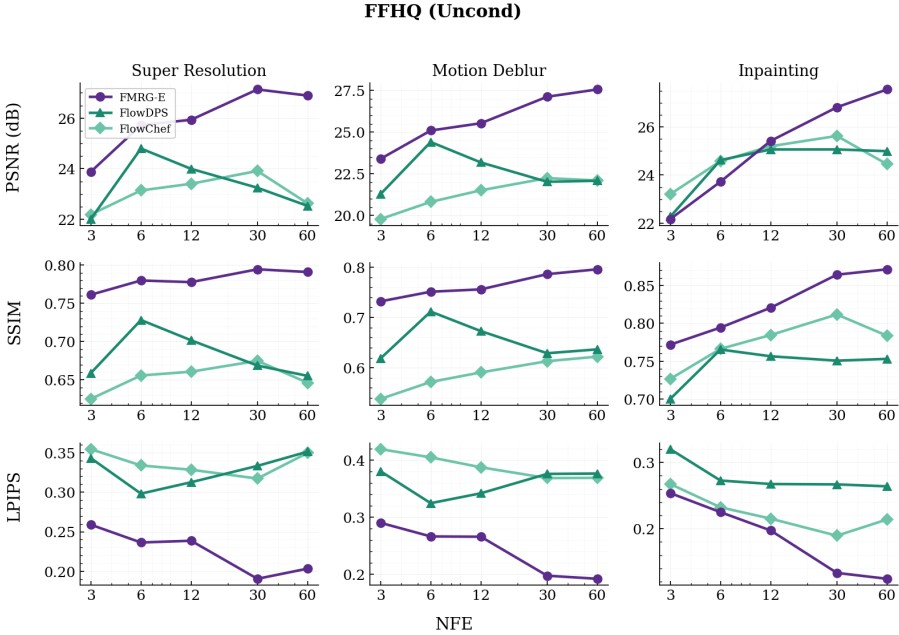

*Figure 12.* **FFHQ results under unconditional (prompt-free) generation.** Same pattern as Figure 11: FMRG is robust to prompt removal while baselines degrade.

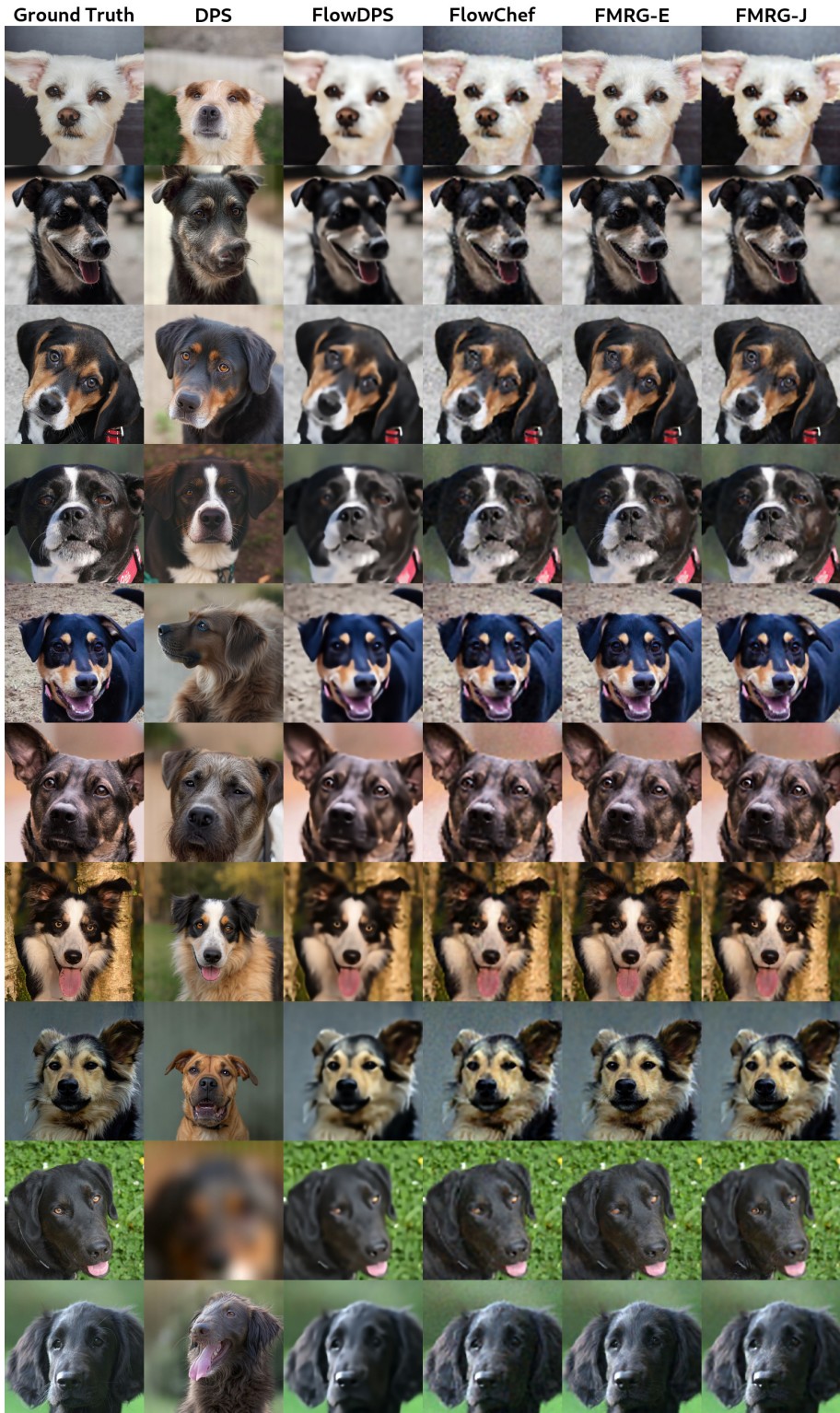

*Figure 13.* **AFHQ** $4\times$ **super-resolution, randomly selected examples (non-cherry-picked).** FMRG variants recover sharper detail than FlowDPS/FlowChef and avoid DPS's severe blurring artifacts.

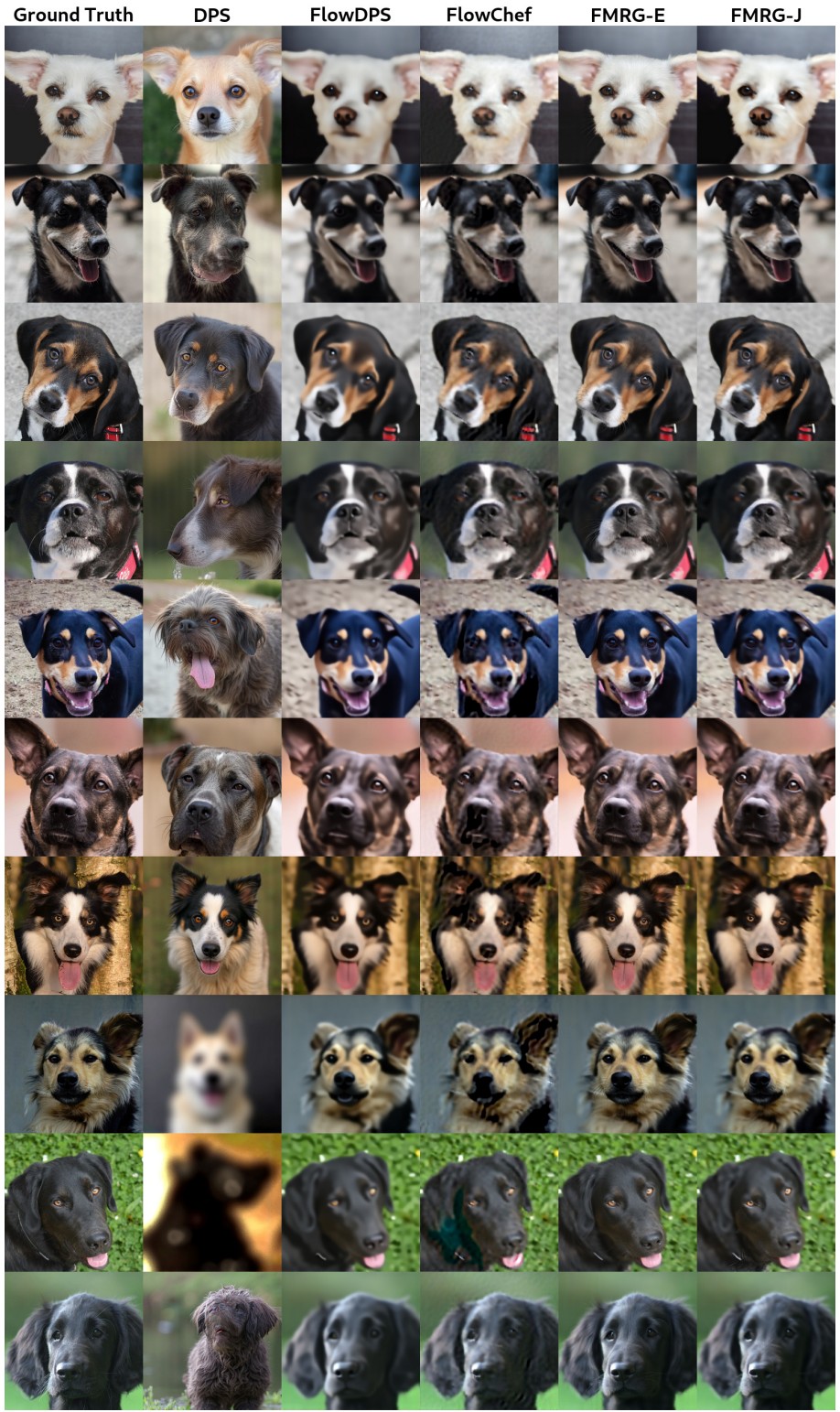

*Figure 14.* **AFHQ motion deblurring, randomly selected examples (non-cherry-picked).** FMRG reconstructions are consistent with the degraded observation and preserve fine texture.

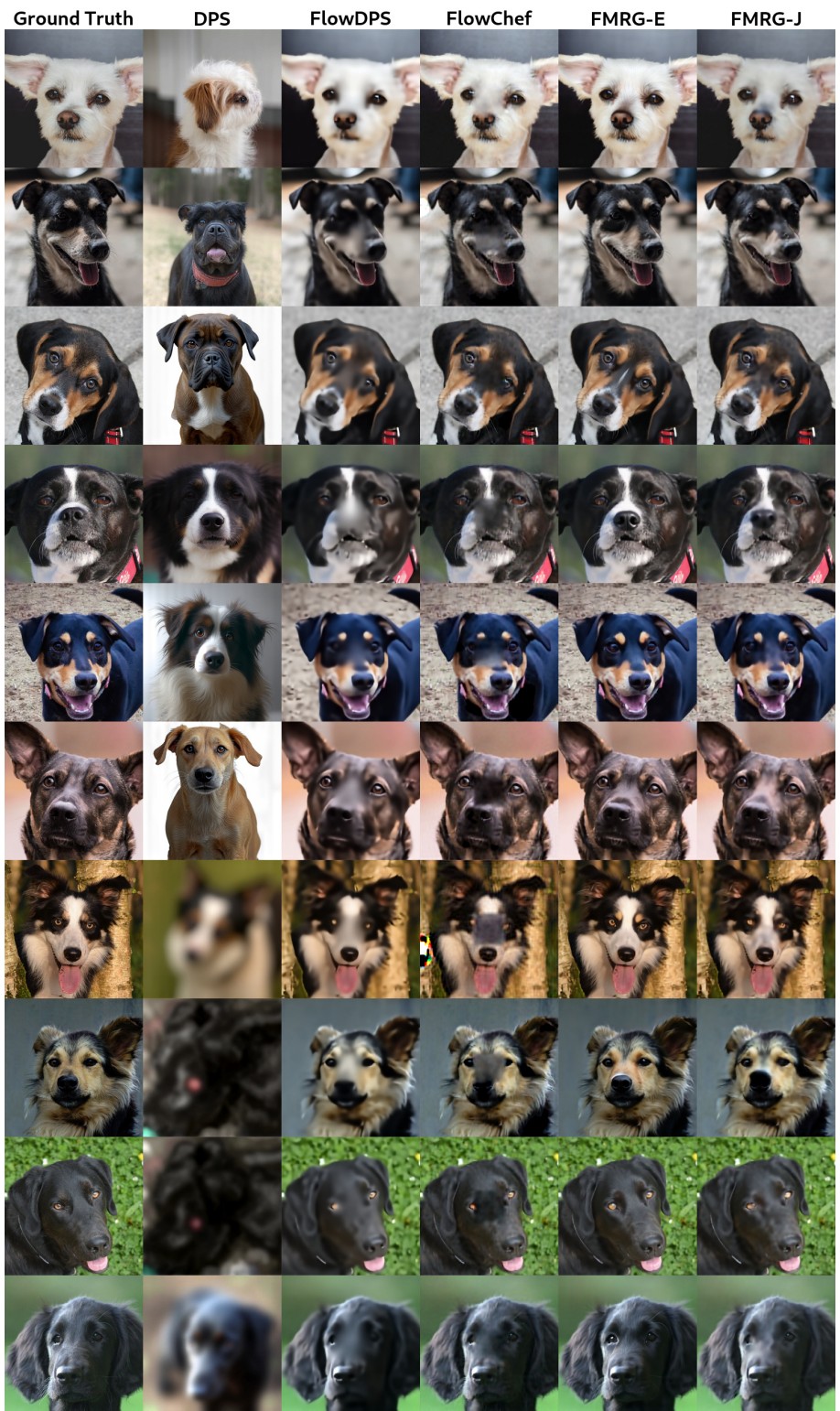

*Figure 15.* **AFHQ box inpainting, randomly selected examples (non-cherry-picked).** FMRG fills the masked region with content consistent with the visible context.

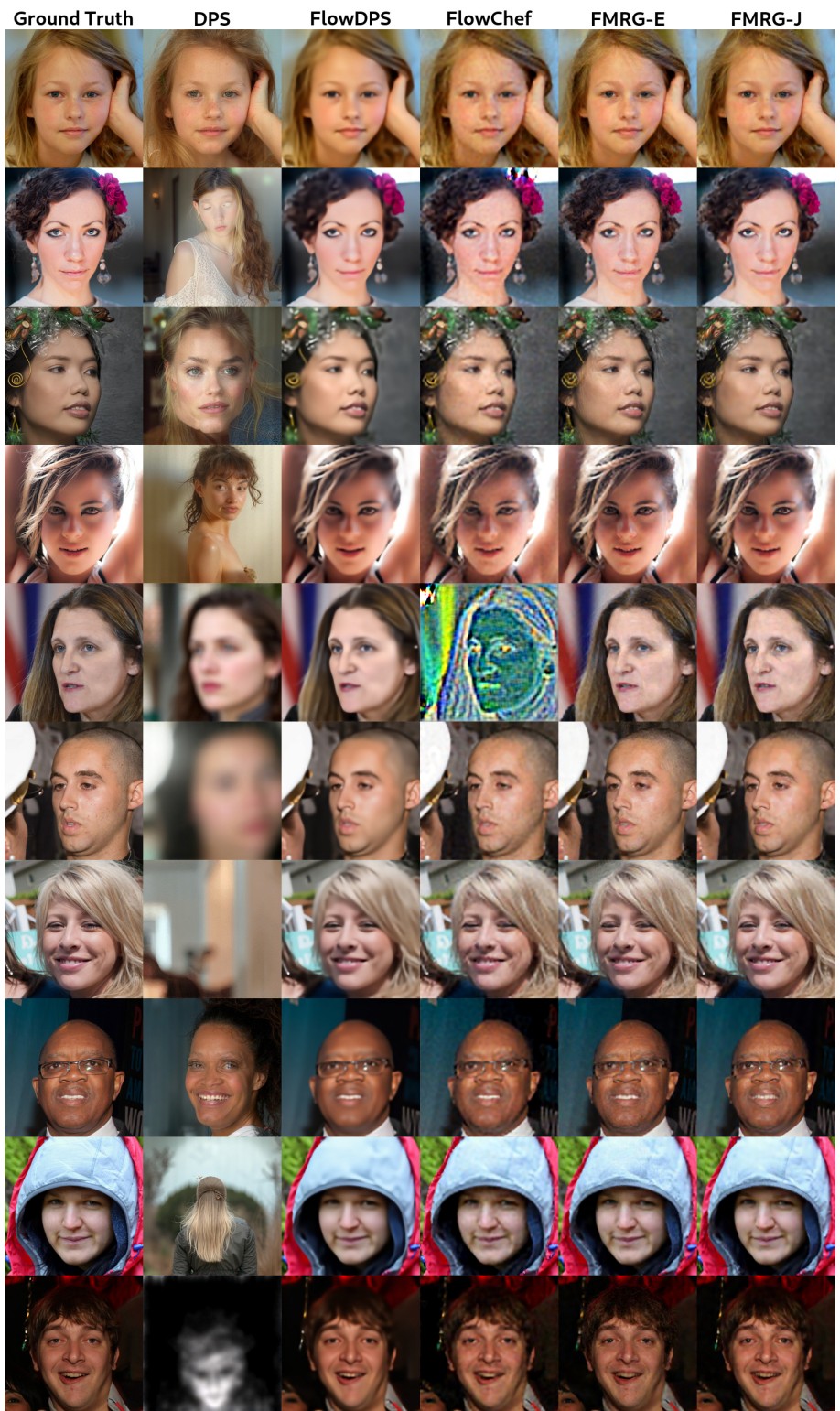

*Figure 16.* **FFHQ** $4\times$ **super-resolution, randomly selected examples (non-cherry-picked).** Same pattern as Figure 13: FMRG recovers sharper facial detail than the baselines.

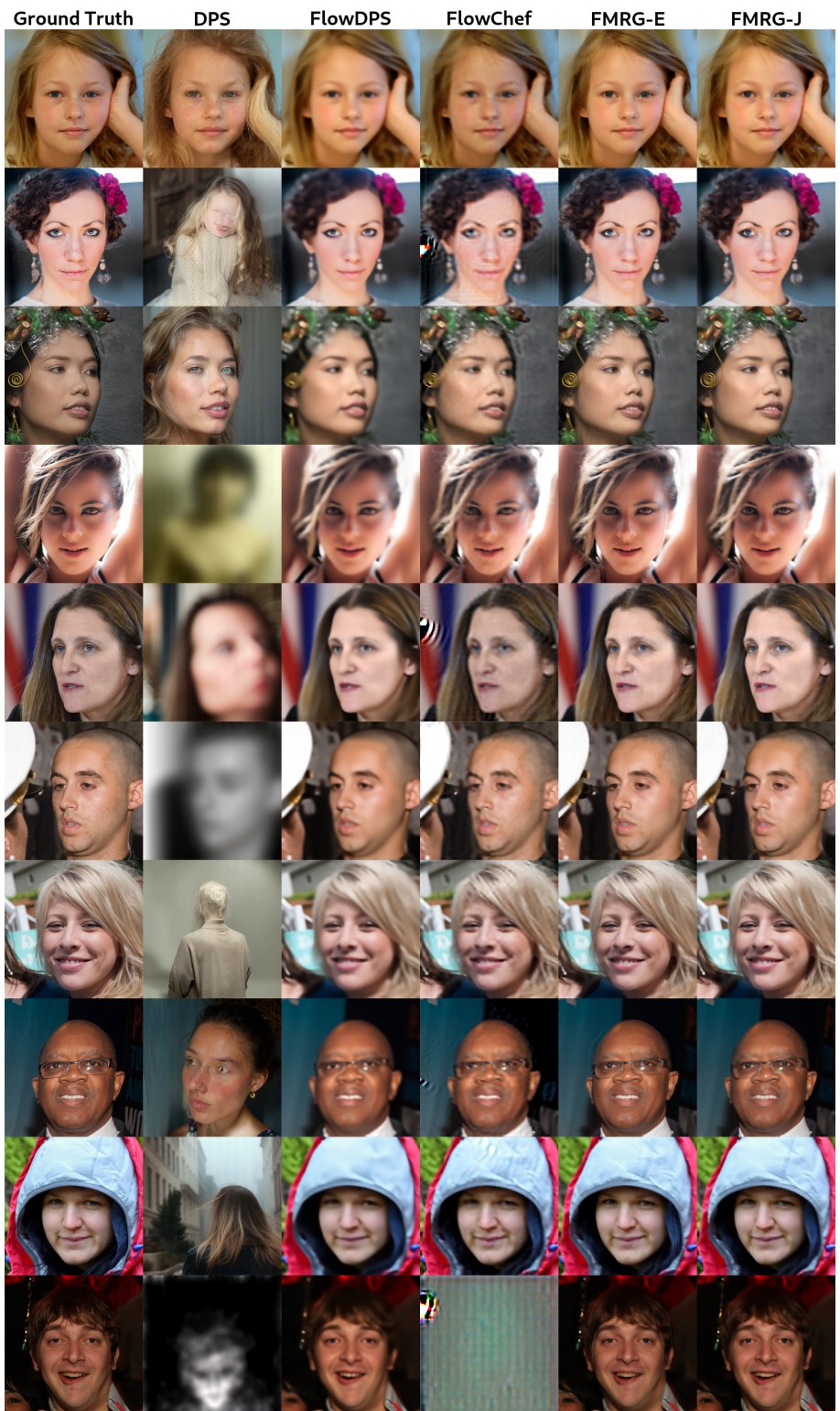

*Figure 17.* **FFHQ motion deblurring, randomly selected examples (non-cherry-picked).** FMRG recovers clean facial structure with minimal artifacts.

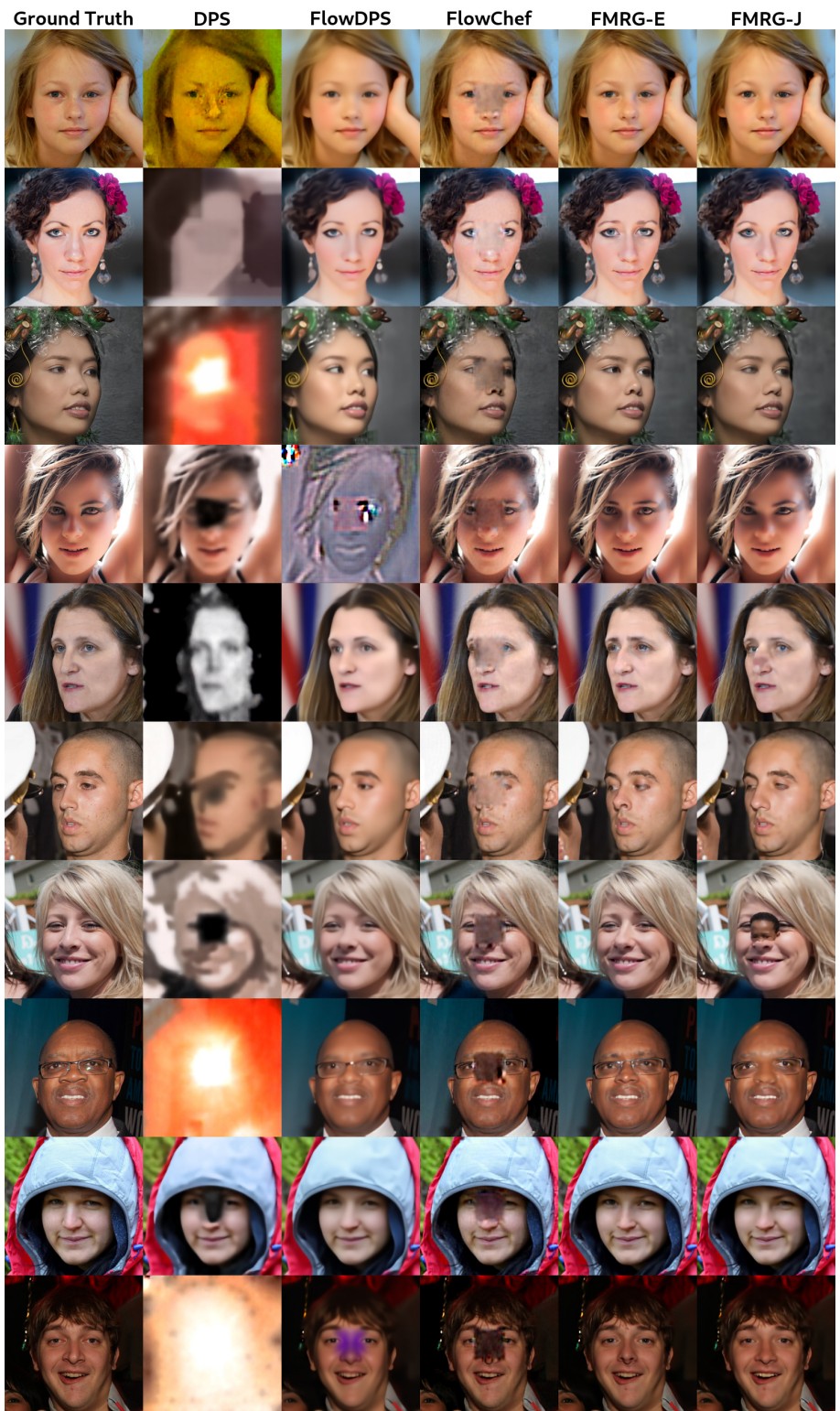

*Figure 18.* **FFHQ box inpainting, randomly selected examples (non-cherry-picked).** Same pattern as Figure 15: FMRG fills masked regions with content consistent with the visible face.

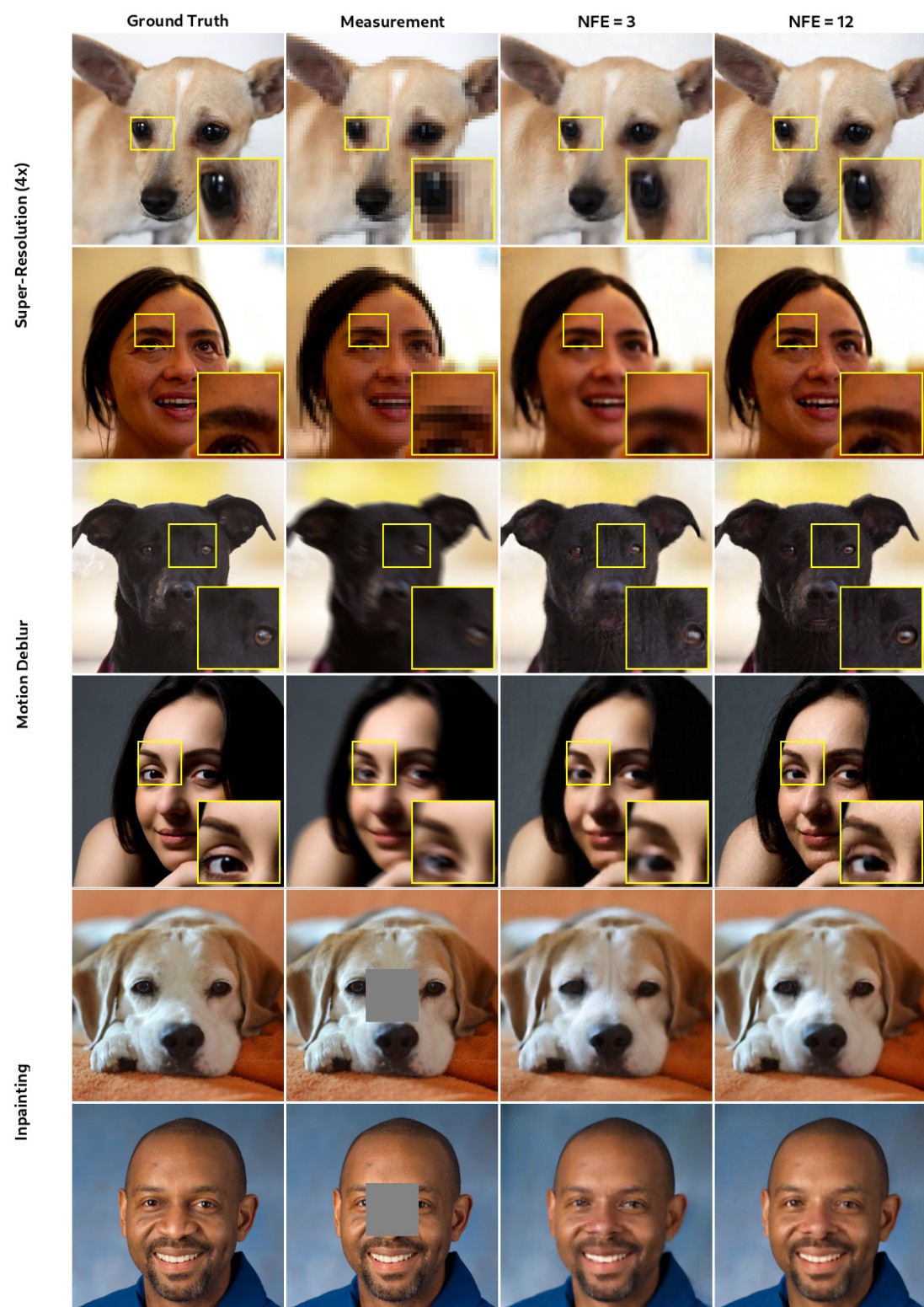

*Figure 19.* **Additional inverse problem qualitative results** with ground truth. Six examples each for super-resolution (4×), motion deblurring, and inpainting on AFHQ and FFHQ.

*Table 7.* **Style transfer with Gram matrix loss.** CLIP-I↑ measures style similarity; CLIP-T↑ measures text-prompt alignment. **Bold** indicates best.

| Method | CLIP-I↑ | CLIP-T↑ |
|---|---|---|
| DPS | 0.625 | 0.262 |
| FlowChef | 0.632 | 0.282 |
| FlowDPS | 0.631 | 0.294 |
| FMRG-E (ours) | 0.646 | 0.290 |
| FMRG-J (ours) | **0.649** | **0.299** |

## E.5. Style transfer

**Reward.** The style loss is the Frobenius norm between the Gram matrices of CLIP ViT-B/16 layer-2 features extracted from the reference and generated images, following MPGD (He et al., 2023).

**Implementation.** We evaluate all methods (FMRG-J, FMRG-E, FlowChef, FlowDPS, DPS) at different numbers of steps, step sizes, and $n_{\mathrm{opt}}$ within a budget of 100 NFEs and report the best configuration for each method.

**Evaluation.** We generate 48 text prompts using GPT-4o by providing 16 style reference images and requesting 3 prompts per style at increasing difficulty levels. All methods are evaluated at $256 \times 256$ resolution. We report CLIP-I (cosine similarity between CLIP image embeddings of the generated image and style reference) and CLIP-T (cosine similarity between the text prompt embedding and the generated image embedding) in Table 7. FMRG-J achieves the best CLIP-I (style similarity) and competitive CLIP-T (text alignment).

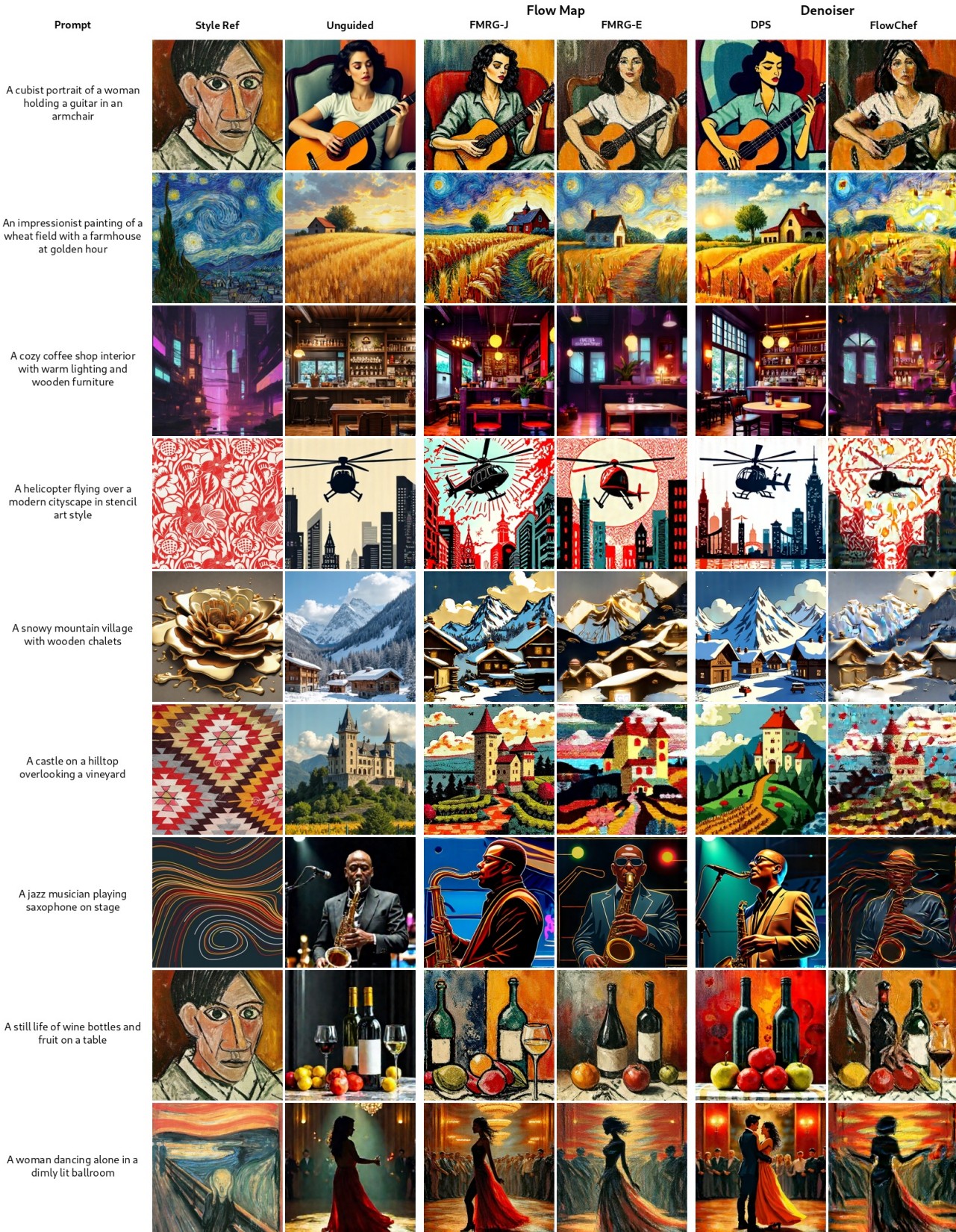

*Figure 20.* **Style transfer hierarchy.** Extended hierarchy comparison across multiple style references. FMRG-J consistently captures the target style while preserving semantic content.

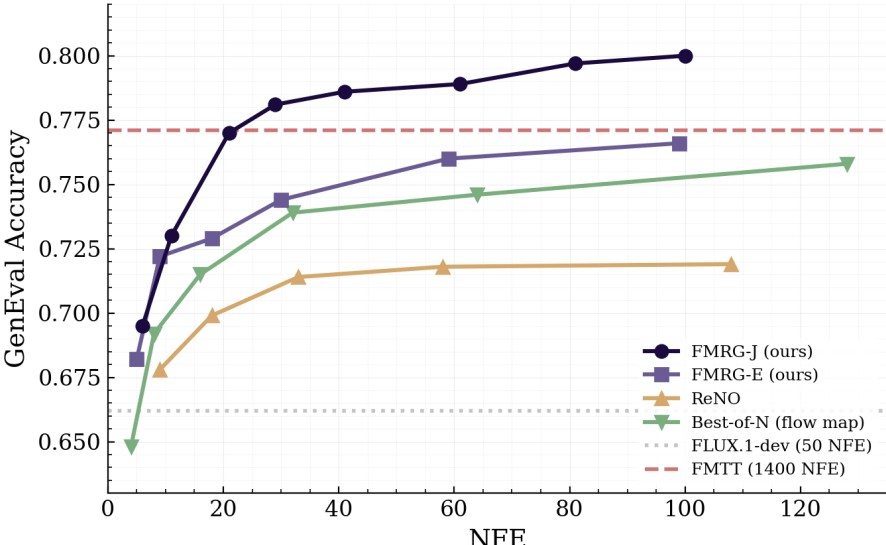

*Figure 21.* **GenEval accuracy vs. NFE.** FMRG-J dominates the Pareto frontier across all NFE budgets, matching FMTT (0.77) at NFE 20 with a 70× reduction in compute.

*Table 8.* **Configurations for GenEval results** (Table 2). All use $K=3$ warmup particles.

| Method | NFE | $\eta$ | $n_{\text{opt}}$ | $t_{\text{stop}}$ |
|--------|-----|--------|------------------|-------------------|
| FMRG-E | 100 | 0.7 | 3 | 0.25 |
| FMRG-J | 20 | 3.0 | 1 | 0.25 |
| FMRG-J | 100 | 5.0 | 1 | 0.25 |

### E.6. Reward-guided generation

**Reward ensemble.** Following Eyring et al. (2024), we use a weighted linear combination of four human-preference reward models:

$$r(x) = 5.0 \cdot \text{HPSv2}(x) + 1.0 \cdot \text{ImageReward}(x) + 0.01 \cdot \text{CLIP}(x) + 0.05 \cdot \text{PickScore}(x). \tag{135}$$

**Evaluation protocol.** The GenEval benchmark (Ghosh et al., 2023) consists of 553 prompts spanning six categories: single object, two objects, counting, colors, position, and color attribution. Object detection uses Mask2Former with a ResNet-50 backbone pretrained on COCO. The overall score is the unweighted mean of per-category accuracies. All methods in Table 2 are evaluated with 4 seeds, with samples aggregated across seeds before computing per-category accuracies.

**FMRG configurations.** Table 8 reports the full configuration for each result in Table 2. Both variants use early stopping at $t_{\text{stop}} = 0.25$. FMRG-J at NFE 20 uses flow map reuse (1 NFE per step), while FMRG-J at NFE 100 and FMRG-E use separate evaluations (2 NFEs per step).

**Warmup particle selection.** Since FMRG guides a single trajectory via greedy optimal control, its performance depends on the quality of the initial noise sample: an unfavorable initialization may land in a region of the reward landscape from which single-trajectory guidance cannot recover. To mitigate this, we adopt a lightweight warmup procedure. We initialize $K = 3$ particles from independent noise seeds and run each through approximately half of the total guided steps. We then evaluate the reward at each particle's predicted endpoint $X_{t,1}(z_t)$ and select the particle with the highest reward. The remaining guided steps continue from the selected particle only. This combines the exploration benefits of best-of-$N$ initialization with the optimization benefits of trajectory guidance at modest additional cost. We find that $K = 3$ restarts is sufficient to consistently improve performance.

*Table 9.* **Early stopping ablation on GenEval (FMRG-J).** Best GenEval accuracy across hyperparameters at each $t_{\mathrm{stop}}$. **Bold** indicates best.

| $t_{\mathrm{stop}}$ | **NFE 30** | **NFE 100** |
|---|---|---|
| 1.0 (none) | 0.748 | 0.779 |
| 0.75 | 0.745 | 0.787 |
| 0.50 | 0.762 | 0.796 |
| 0.25 | **0.768** | **0.800** |

**Baseline configurations.** **FMTT** (Singhal et al., 2025): We re-evaluate on our flow map backbone with 7 particles, 2 clones per particle, 20 sampling steps, lookahead depth 4, and a reward scaling factor of 4.5, totaling 1400 NFEs, as reported in Singhal et al. (2025). The reward is the same ensemble used by FMRG. **ReNO** (Eyring et al., 2024): Optimizes the initial noise latent via 50 gradient steps, using 8 sampling steps per reward evaluation and a learning rate of 5.0, for ∼58 NFEs. ReNO peaks around 50 iterations and degrades with longer optimization. **Best-of-**$N$: Generates $N$ samples independently with the 4-step flow map (no guidance) and selects the highest-reward sample. The main result uses $N$=32 (128 NFEs).

**Early stopping ablation.** Table 9 reports the effect of early stopping on FMRG-J for GenEval at NFE 30 and 100 (both without flow map reuse). $t_{\mathrm{stop}} = 0.25$ consistently outperforms all other levels, corroborating the Gaussian analysis in Appendix C.5. Figure 22 shows qualitative examples: as $t_{\mathrm{stop}}$ decreases from 0.75 to 0.25, image quality improves; removing early stopping ($t_{\mathrm{stop}} = 1$) leads to over-optimization artifacts.

**Additional aesthetic enhancement results.** Figures 23 and 24 show additional qualitative comparisons between FMRG and ReNO for reward-guided aesthetic enhancement.

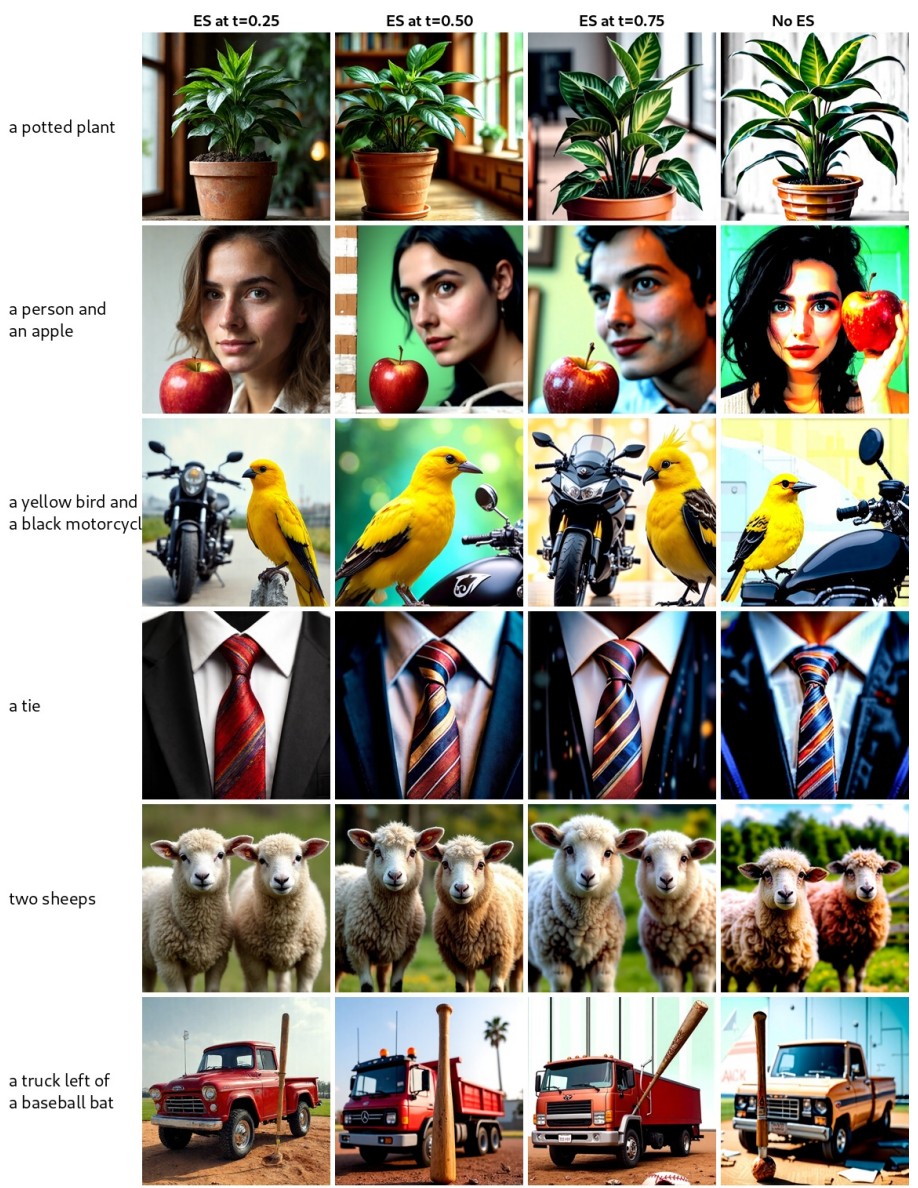

*Figure 22.* **Effect of early stopping on generation quality.** Each row shows a different prompt guided by the reward ensemble with FMRG-J at varying $t_{\text{stop}}$. $t_{\text{stop}} = 0.25$ consistently yields the best quality; removing early stopping ($t_{\text{stop}} = 1$) produces over-optimized artifacts.

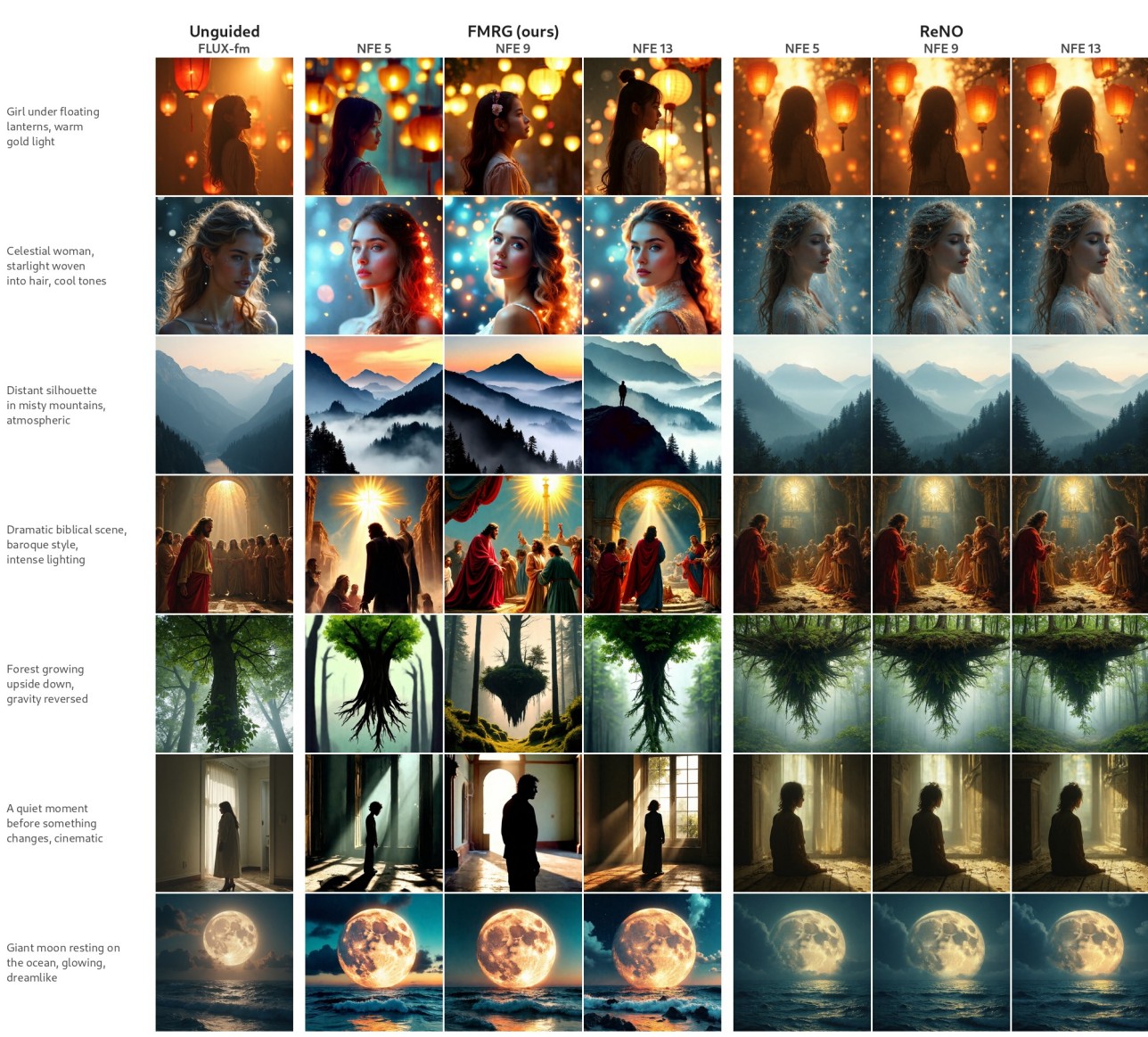

*Figure 23.* **Additional aesthetic enhancement comparisons (1/2).** At matched low NFE budgets, FMRG produces more dramatic aesthetic enhancements than ReNO.

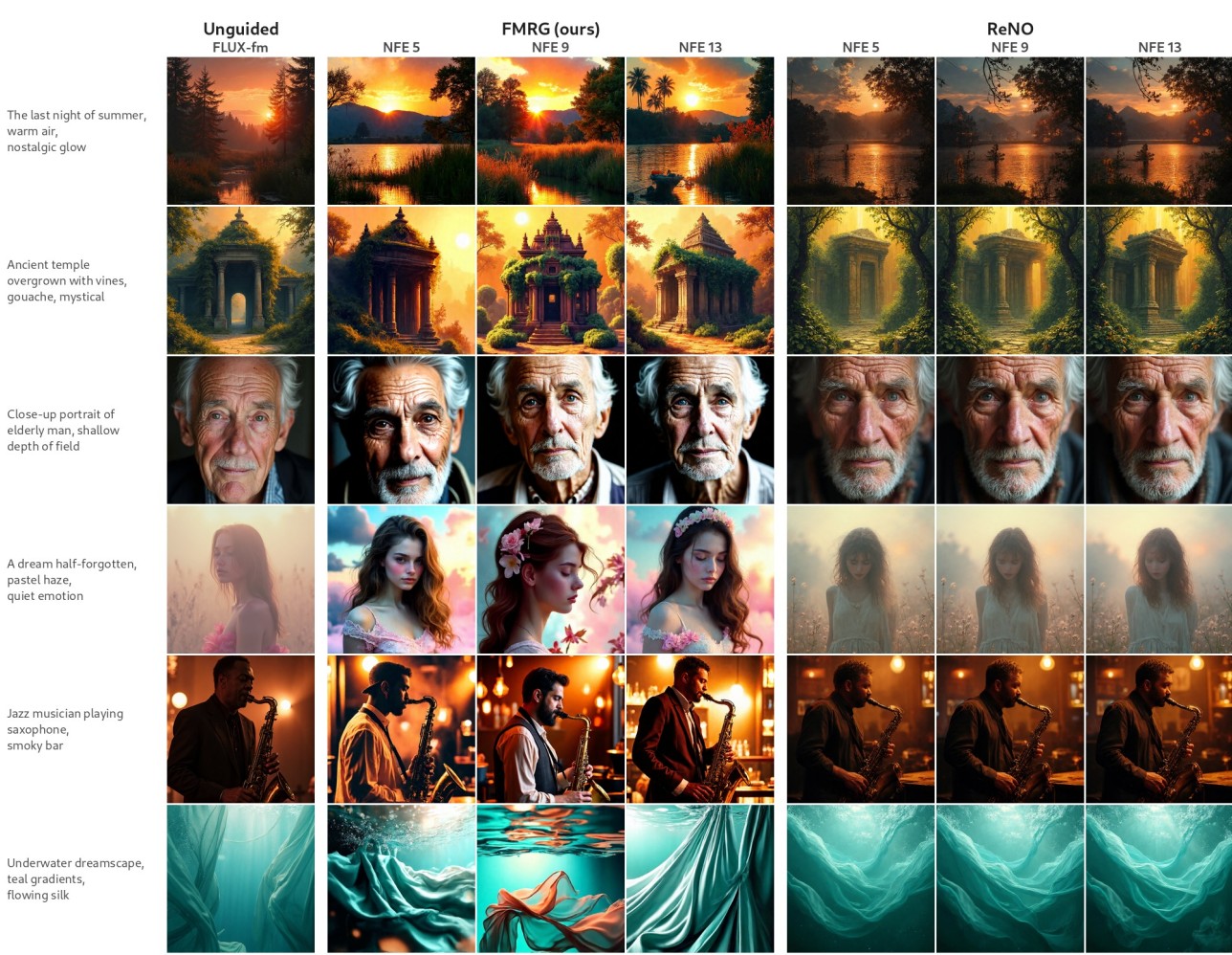

*Figure 24.* **Additional aesthetic enhancement comparisons (2/2).** At matched low NFE budgets, FMRG produces more dramatic aesthetic enhancements than ReNO.

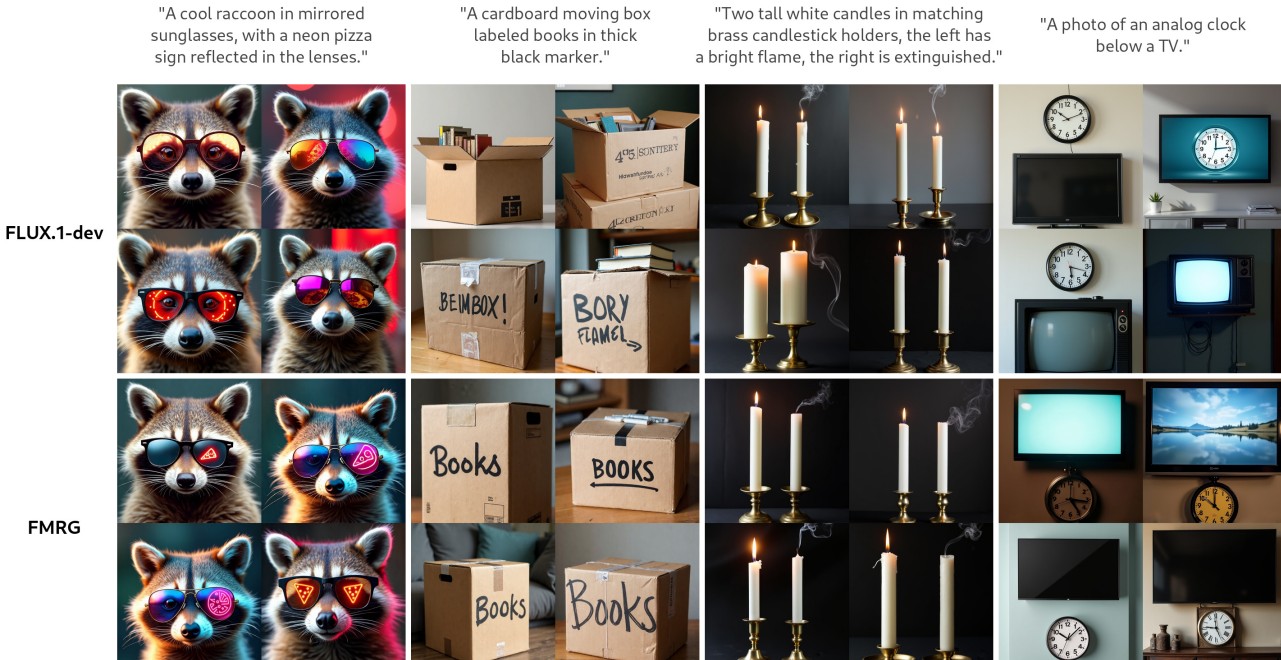

*Figure 25.* **VLM reward guidance.** Unguided FLUX generations (top) fail to follow complex compositional prompts. FMRG (bottom) steers generation toward prompt-faithful outputs.

### E.7. VLM guidance

For complex compositional prompts (Figure 25), we use Skywork-VL-Reward-7B (Wang et al., 2025) as a reward model. Given an image $x$ and text prompt $p$, the reward is $r(x,p) = \sigma(\text{logit}_{\text{Yes}}(x,p) - \text{logit}_{\text{No}}(x,p))$, where $\text{logit}_{\text{Yes/No}}$ are the VLM's last-token logits at the "Yes"/"No" vocabulary positions after the chat-template query "Is `"{prompt}"` a correct caption for the image?" We use FMRG-J with 20 guided steps, step size $\eta = 1.0$, and 4 warmup particles.

**Stochastic renoising.** We observe that for some images, VLM-guided generation can produce minor artifacts. We find that injecting a small amount of stochastic noise toward the end of the generation process is effective at reducing these artifacts and sharpening image quality. Concretely, given the current latent $x_t$ and the flow map velocity $v_{t,t_+}(x_t)$, we decompose the next-step prediction into a clean-sample estimate $\hat{x}_1$ and a noise estimate $\hat{x}_0$ via the linear interpolant:

$$\hat{x}_1 = x_t + (1-t)\, v_{t,t_+}(x_t), \qquad \hat{x}_0 = x_t - t\, v_{t,t_+}(x_t). \tag{136}$$

We then form a renoised prior by mixing the estimated noise with fresh Gaussian noise $\epsilon \sim \mathsf{N}(0, I)$:

$$\tilde{x}_0 = (1-c)\, \hat{x}_0 + c\, \epsilon, \tag{137}$$

and reconstruct the current latent using the renoised prior:

$$\tilde{x}_t = (1-t)\, \tilde{x}_0 + t\, \hat{x}_1. \tag{138}$$

The guided step then proceeds from $\tilde{x}_t$. The constant $c \in [0, 1]$ controls the amount of injected stochasticity. This lightweight procedure adds diversity without significantly altering the guided trajectory, and we find it effective in practice for VLM rewards.

### E.8. Wall-clock time and VRAM

We report wall-clock time per image and peak VRAM for all methods in Tables 10 and 11.

*Table 10.* **Wall-clock time and VRAM for reward-guided generation** (512px, H100).

| Method | NFE | Time/img (s) | VRAM (GB) |
|---|---|---|---|
| FMTT | 1400 | 337.5 | 73.5 |
| Best-of-$N$ ($N{=}32$) | 128 | 13.1 | 35.9 |
| ReNO (seed optim) | 58 | 21.9 | 48.2 |
| FMRG-E (ours) | 100 | 34.5 | 39.8 |
| FMRG-J (ours) | 20 | 5.5 | 48.2 |
| FMRG-J (ours) | 100 | 24.8 | 48.2 |

*Table 11.* **Wall-clock time and VRAM for inverse problems** (256px, L40S).

| Method | NFE | $n_{\mathrm{opt}}$ | Time/img (s) | VRAM (GB) |
|---|---|---|---|---|
| FlowChef | 30 | 3 | 5.10 | 23.3 |
| FlowDPS | 30 | 3 | 4.86 | 23.3 |
| FMRG-E | 30 | 3 | 4.85 | 23.3 |
| FMRG-J | 30 | 1 | 5.68 | 27.5 |
| FlowChef | 200 | 3 | 32.19 | 23.3 |
| FlowDPS | 200 | 3 | 32.23 | 23.3 |
| FMRG-E | 200 | 3 | 27.89 | 23.3 |
| FMRG-J | 200 | 1 | 35.62 | 27.5 |

# F. Extended related work

**Dynamical measure transport.** Flow- (Albergo et al., 2023; Lipman et al., 2022; Liu et al., 2022b) and diffusion-based (Song et al., 2021) generative models have achieved state of the art performance across diverse continuous modalities by learning to transport a simple reference distribution such as a Gaussian to a complex target. More recently, consistency models (Kim et al., 2024; Geng et al., 2024; Song et al., 2023; Heek et al., 2024) and flow maps (Sabour et al., 2025b; Geng et al., 2025a;b; Frans et al., 2024; Boffi et al., 2025a;b; Zhou et al., 2025) have been introduced to learn to map directly along a dynamical measure transport process, dramatically increasing sampling efficiency. While these models have become increasingly capable, principled guidance methods native to the few-step, single-trajectory regime of flow maps remain underexplored, motivating the present study.

**Guidance.** Guidance, or inference-time alignment, adjusts the output of a pre-trained generative model to better match downstream user-specified rewards without additional training (Uehara et al., 2025). The prevailing theoretical framework casts guidance as approximate sampling from a reward-tilted distribution, motivating SMC-based approaches that maintain particle populations and perform resampling (Wu et al., 2023a; Ren et al., 2025; Uehara et al., 2024a). While principled, these methods sacrifice the efficiency of single-trajectory inference. An alternative line of work applies guidance heuristically to single trajectories, including DPS (Chung et al., 2024), FlowDPS (Kim et al., 2025), FlowChef (Patel et al., 2025), MPGD (He et al., 2023), and RB-Modulation (Rout et al., 2024). Our framework provides a principled foundation for this class of methods, subsuming DPS and related single-trajectory approaches as coarse approximations to the greedy guidance signal derived from deterministic optimal control; the full reduction of each method to FMRG is given in Appendix D. More generally, seed optimization methods such as ReNO (Eyring et al., 2024) and D-Flow (Ben-Hamu et al., 2024) emerge as up-front optimization within the same framework. A complementary unifying perspective is given by Feng et al. (2025), who construct a general framework for flow-matching guidance whose velocity field samples the reward-tilted distribution $p' \propto p \, e^{-J}$. In their framework, DPS and related methods arise from a Taylor approximation of an expectation over the posterior $p(x_1 \mid x_t)$, whose error is controlled by the posterior variance and vanishes in the small-variance limit, for example as $t \to 1$. FMRG does not target the reward tilt and uses no posterior or small-variance approximation. Its guidance is a deterministic optimal-control signal defined through the exact flow-map endpoint, so DPS-type methods are recovered by the single approximation of replacing that endpoint with one Euler step (Appendix D), a more direct reduction that introduces no small-variance assumption. Flow map trajectory tilting (Sabour et al., 2025a) also uses the flow map to evaluate the reward, but steers a multi-particle diffusion process towards the reward tilt. In contrast, FMRG uses the flow map for both integration and guidance along a single deterministic trajectory, achieving up to a $70\times$ reduction in NFE at matched quality

(Section 5).

**Reward fine-tuning.**   An alternative to inference-time guidance is to bake reward alignment into the model via fine-tuning, which requires an additional training phase. This can be formalized as an instance of stochastic optimal control (Uehara et al., 2024b;a; Domingo-Enrich et al., 2025), which gives an interpretation of reward tilting guidance as an inference-time approximation to the optimal solution of this control problem. Here we pursue a *deterministic* optimal control problem and construct an efficient approximate solution to design a guidance signal. Reinforcement learning approaches such as DDPO (Black et al., 2024) and DPOK (Fan et al., 2023) treat the sampler as a Markov decision process and optimize model parameters to maximize expected reward, while gradient-based methods such as DRaFT (Clark et al., 2024) backpropagate through the full sampling process. Such approaches can achieve strong reward alignment but require expensive retraining for each new reward model and are prone to over-optimization that degrades base-model quality. In concurrent work, Diamond Maps (Holderrieth et al., 2026) and Meta Flow Maps (Potaptchik et al., 2026) design new stochastic flow maps to make reward-tilt sampling more computationally feasible; these approaches require training a new model or a more expensive posterior estimate, as well as multiple Monte Carlo rollouts, whereas we focus on a simpler deterministic setting that uses the existing pre-trained flow map without retraining. Variational Flow Maps (Mammadov et al., 2026) similarly require training an auxiliary noise adapter to enable reward alignment. In contrast, FMRG achieves reward alignment at inference without modifying the underlying flow, preserving the base model and allowing different rewards to be applied to a single checkpoint.

