# OpenReview forum: "How to Guide Your Flow: Few-Step Alignment via Flow Map Reward Guidance"
_ICML.cc/2026/Conference — ICML 2026 regular_

### Official Review · Reviewer_P8Pt · 2026-03-12

**Soundness:** 3
**Presentation:** 3
**Significance:** 3
**Originality:** 3
**Overall Recommendation:** 5
**Confidence:** 2

**Summary:**

Instead of trying to sample from a new guided distribution, they directly optimize one generation trajectory so that the final sample gets high reward while not moving too far away from the original flow model.

**Compliance With Llm Reviewing Policy:**

Affirmed.

**Final Justification:**

My issues have been resolved in the rebuttal.

**Key Questions For Authors:**

NA

**Limitations:**

Yes

**Strengths And Weaknesses:**

### Strengths

1. The paper has been well written and the motivation seems clear to me.
2. The theory established is pretty good. I particularly liked the hierarchy of approximations that the authors have established (line 172; 2nd column).

### Weakness

1. For Table-1, the datasets used are AFHQ and FFHQ. Both these datasets consists of closely-cropped and aligned images of faces. Is there a reason why these datasets were chose ? It would be interesting to see the results/evaluation metrics on a more diverse and general dataset.

---

> ### Author Rebuttal · Authors · 2026-03-31
>
> We thank the reviewer for their time and are delighted to hear they believe our paper is well-written with established theory (supplementary tables: https://drive.google.com/file/d/1mcQ1IWwwVpj5ShyqGdcpKvCMxsAoparg/view?usp=drive_link). We agree that the AFHQ/FFHQ evaluation in Table 1 is limited in diversity. We chose these examples as standard datasets for evaluating inverse problem solvers so that we can directly compare to baseline methods such as FlowDPS and FlowChef.
>
> To improve the comprehensiveness of our evaluation beyond these standard benchmarks, in the revision we have expanded our experiments to human preference, aesthetic, and text-image alignment rewards. We now also evaluate on GenEval [2], a compositional text-to-image benchmark with 553 diverse prompts spanning single objects, two objects, counting, colors, spatial relationships, and attribute binding. In this new round of experiments, we find strong improvements of FMTG over all baselines in terms of both quality and computational complexity, measured *either* in terms of NFEs or wall clock time and memory requirements:
>
> GenEval accuracy (4-seed). FMTT (SMC baseline) evaluated on our flow map.
>
> | Method | NFE | GenEval $\\uparrow$ |
> | --- | --- | --- |
> | FLUX.1-dev (unguided) | 50 | 0.662 |
> | Best-of-$N$, reward-selected | 400 | 0.751 |
> | ReNO [3], seed optim | 58 | 0.716 |
> | FMTT [1] | 1400 | 0.771 |
> | FMTG (ours) | 30 | 0.768 |
> | FMTG (ours) | 50 | 0.782 |
> | FMTG (ours) | 100 | **0.791** |
>
> FMTG matches FMTT [1] (an SMC baseline) with $47\\times$ fewer NFEs, and surpasses reward-selected Best-of-$N$ with $13\\times$ fewer NFEs. At NFE 50, FMTG outperforms all baselines. We additionally provide quantitative results on style transfer (Supplementary Table 5), comprehensive ablations on all design decisions (Supplementary Tables 6–8), and wall-clock/VRAM measurements (Supplementary Table 4).
>
> We appreciate the reviewer's positive assessment of the writing, motivation, and theory, as well as their constructive feedback. We hope the expanded evaluation addresses the remaining concern, and we would be grateful if the reviewer could consider raising their assessment in light of the improved experimental breadth and results.
>
> [1] Sabour et al., "Test-time scaling of diffusions with flow maps," arXiv:2511.22688, 2025.
>
> [2] Ghosh et al., "GenEval: An Object-Focused Framework for Evaluating Text-to-Image Alignment," NeurIPS 2024.
>
> [3] Eyring et al., "ReNO: Enhancing One-step Text-to-Image Models through Reward-based Noise Optimization," arXiv:2406.04312, 2024.

---

> > ### Author Rebuttal · Reviewer_P8Pt · 2026-04-02
> >
> > My concerns have been resolved. I am increasing my ratings. Thank you and best of luck.

---

### Official Review · Reviewer_RJNK · 2026-03-13

**Soundness:** 3
**Presentation:** 2
**Significance:** 2
**Originality:** 3
**Overall Recommendation:** 4
**Confidence:** 4

**Summary:**

This paper proposes a training-free method to guide generation results, thereby achieving conditional generation.

**Compliance With Llm Reviewing Policy:**

Affirmed.

**Final Justification:**

The authors have addressed my concerns. I have no further questions, and I have decided to raise my score.

**Key Questions For Authors:**

Please add discussions or experiments based on the suggestions in the weaknesses section.

**Limitations:**

I did not see a discussion of limitations from the authors. I believe there are three main limitations: 1) The cost is not low, as each NFE involves multiple optimizations; 2) The performance upper bound is likely far inferior to basic training-based methods; 3) It seems effective only on tasks related to conditional generation.

**Strengths And Weaknesses:**

1. Strengths:
    * This paper provides extensive theoretical derivations, examining the generation process from the perspective of a control problem.
2. Weaknesses:
    * From Algorithm 1, each forward pass requires multiple steps of gradient calculation and updates, so the computational cost seems high. It does not appear as lightweight as the 3NFE claimed by the authors. The authors should report the actual computational cost and compare it with other methods.
    * There is a lack of experiments related to early stopping to demonstrate the necessity and effectiveness of the strategy.
    * All experiments in this paper are conducted on conditional generation tasks, such as super-resolution and inpainting. Can it be used for tasks like improving aesthetics or image quality?
    * The authors should also compare with tuning-based conditional generation methods, because the results shown seem achievable without significant training costs. If so, the significance of this work would be much lower.

---

> ### Author Rebuttal · Authors · 2026-03-31
>
> We would first like to thank the reviewer for their time and helpful feedback! We hope to address some of the concerns raised below (supplementary tables: https://drive.google.com/file/d/1mcQ1IWwwVpj5ShyqGdcpKvCMxsAoparg/view).
>
> **Re: Computational Cost.**
> To ensure an equitable comparison with existing guidance methods, we report our results by counting the NFEs, which is standard practice in the literature. We agree that the actual computational cost is not always clearly illustrated with this metric. In our revised manuscript, we additionally report wall-clock time per image and peak VRAM (Supp. Table 4). FMTG-J at NFE 30 takes 7.9s per image, compared to 337.5s for FMTT [1].
>
> Regarding our 3 NFE claim: For FMTG-E, the inner optimization steps (n_opt) do not re-evaluate the 12B-parameter flow map — they operate only on the predicted clean sample (VAE decode + reward evaluation), costing approximately 1/10 of a flow map evaluation in wall-clock time. We use the same n_opt for all methods including FlowDPS and FlowChef, so the comparison is made at equivalent per-step compute. Importantly, FMTG still exhibits strong performance and outperforms baselines at higher compute budgets (Supp. Table 7). We agree that for more complex reward functions, FMTG will require more NFEs, and we will make this clearer in the revision.
>
> **Re: Early Stopping Experiments.**
>
> In the revision, we provide a new suite of extensive early stopping ablations. On inverse problems (Supp. Table 8), across 2 datasets × 3 tasks × 5 NFE levels, early stopping wins on FID in 29/30 comparisons (97%), SSIM in 27/30 (90%), and LPIPS in 22/30 (73%). On GenEval [2] (Supp. Table 2), early stopping also improves accuracy. Reward over-optimization ("reward hacking") is a well-known challenge in both guidance and reinforcement learning; our contribution is to formally analyze this phenomenon in the context of flow-based guidance (Section 2.4, Proposition 2.4) and provide a principled mitigation strategy through early stopping, which we prove recovers the correct scaling in the Gaussian setting. We find that the empirical performance matches well with our theory, which predicts that early stopping is effective and necessary to mitigate reward hacking.
>
> **Re: Conditional Generation Tasks.**
>
> In new experiments, we demonstrate that FMTG is effective on general reward-guided generation tasks beyond conditional generation. We evaluate on GenEval (Supp. Table 1), a compositional benchmark with 553 diverse prompts, guided by an ensemble of four human preference and aesthetic reward models (HPSv2, ImageReward, CLIP, PickScore). FMTG-J achieves 0.791 GenEval accuracy at NFE 100, outperforming FMTT (0.771 at 1400 NFE, 47× more compute), Best-of-$N$ (0.751 at 400 NFE), ReNO seed optimization (0.716 at 58 NFE), and the unguided baseline (0.662). Even at only NFE 30, FMTG-J (0.768) surpasses all baselines. We additionally provide quantitative style transfer results (Supp. Table 5), where FMTG outperforms FlowDPS and FlowChef on both style fidelity (CLIP-I) and text alignment (CLIP-T). Together, these results demonstrate that FMTG is effective for steering generation with a variety of differentiable rewards.
>
> **Re: Upper bound of guidance methods.**
>
> While both finetuning and guidance methods aim to generate samples aligned with a defined reward function, they tackle different aspects of the problem. Finetuning optimizes the model's learned distribution, while guidance optimizes individual generation trajectories. Importantly, the two are complementary: guidance methods can be applied on top of finetuned models. For example, FMTG further improves FLUX.1-dev, which has already undergone extensive post-training. Furthermore, principled finetuning frameworks have yet to be established for flow maps and remain a direction for future work.
>
> We also argue that the benefits of inference-time guidance can be seen as models scale: at FLUX scale (12B parameters), finetuning typically requires multi-GPU setups and must be retrained for each reward, while inference-time guidance can be applied immediately to any differentiable reward. Our method enables highly efficient guidance, often in seconds per image. We expand our experiments beyond conditional generation to aesthetic, human preference, and style transfer rewards, showing that we can improve FLUX.1-dev — despite its extensive post-training — at inference time using our approach, matching FMTT (SMC baseline) with 47× fewer NFEs on compositional evaluation (Supp. Tables 1, 4, 5). We additionally report wall-clock time and VRAM measurements (Supp. Table 4), supporting our claims of computational efficiency. Given this, we would be grateful if the reviewer could consider updating their assessment.
>
> [1] Sabour et al., "Test-time scaling of diffusions with flow maps," arXiv:2511.22688, 2025.
>
> [2] Ghosh et al., "GenEval: An Object-Focused Framework for Evaluating Text-to-Image Alignment," NeurIPS 2024.

---

> > ### Author Rebuttal · Reviewer_RJNK · 2026-04-03
> >
> > Thank you for the authors’ response. I have no further questions.

---

### Official Review · Reviewer_C5ix · 2026-03-13

**Soundness:** 3
**Presentation:** 3
**Significance:** 3
**Originality:** 3
**Overall Recommendation:** 5
**Confidence:** 2

**Summary:**

The paper formulates the inference-time guidance problem as a deterministic optimal control problem. The authors then characterize the optimal control solution as a function of the flow map for the optimally controlled dynamics. They then consider the heavily regularized setting in which the solution reduces to a function of the uncontrolled flow map making it tractable. They then propose an algorithm (FMTG) which guides the trajectory using this optimal control term. In experiments they show improvements across a diverse range of linear inverse problems and style guidance especially when using a small number of function evaluations.

**Compliance With Llm Reviewing Policy:**

Affirmed.

**Key Questions For Authors:**

1- Is there a theoretical explanation for why the advantages of FMTG are more apparent at smaller NFEs compared to baselines?

Also see weaknesses above.

**Limitations:**

yes.

**Strengths And Weaknesses:**

**Strengths**

1- The paper is very well-written, with the problem clearly motivated and the problem formulation and the theoretical results presented in an intuitive and easy to understand way.

2- The approach is theoretically well-justified, with the flow map arising naturally from the optimal solution of the problem formulation.

3- The empirical results are impressive and show consistent and significant improvement across a variety of tasks. The paper also presents thorough and convincing ablations. The low NFE regime performance is particularly impressive leading to a potentially very useful method in resource constrained settings where inference time needs to be kept small.

**Weaknesses**

1- I found the explanations regarding NFEs a bit confusing in parts. For example in 3.4 why does using $X_{t_k, t_{k+1}}$ require evaluating more NFEs per step than $X_{t_k, 1}$?

2- I also found the discussion regarding the early stopping and proposition 2.5 confusing. It is unclear to me which of the three variance scalings is preferable.

3- The practical implications of using Jacobian vs Euclidean gradient type in the algorithm seem a bit vague. Would we expect the Euclidean gradient to work well from the theory? Why or why not?

From my understanding, this is a strong paper with a novel well-justified approach that also seems to produce significant empirical gains. But I have not carefully checked the math/proofs and am not too familiar with related works in this area, as such I have chosen a low confidence score to reflect that.

---

> ### Author Rebuttal · Authors · 2026-03-31
>
> We thank the reviewer for their time and are delighted to hear they believe our paper is novel and well-written. We clarify the points of confusion below (supplementary tables: https://drive.google.com/file/d/1mcQ1IWwwVpj5ShyqGdcpKvCMxsAoparg/view).
>
> **Re: 1. Explanations about NFEs.**
>
> Each step of FMTG requires evaluating $X_{t_k,1}$ (reward) and $X_{t_k,t_{k+1}}$ (flow), leading to two NFEs. We find empirically we can reduce this to one by rescaling the endpoint evaluation via $X_{t_k,t_{k+1}}(x) \\approx x + \\tfrac{t_{k+1}-t_k}{1-t_k}(X_{t_k,1}(x) - x)$. This is exact for linear flows and works well when the guidance is sufficient to keep the sample near the data manifold. We also use early stopping, concluding with a *single* application of $X_{t_{stop},1}$. This is exact, enabling compelling reconstructions at NFE 3.
>
> **Re: 2. Variance Scalings.**
>
> The three scalings characterize how each guidance scheme affects sample diversity: the greedy approximation contracts variance exponentially in λ, while exact optimal control and reward tilting exhibit milder polynomial contraction. This analysis reveals a key insight: reward over-optimization is a well-known challenge across guidance and reinforcement learning, and our framework formally identifies when it occurs and how to prevent it. Specifically, early stopping limits the interval over which the greedy guidance is applied. In the Gaussian setting, this provably recovers the polynomial scaling of exact optimal control. Empirically, early stopping consistently improves sample quality, e.g., improving FID in 97% of settings (Supp. Table 8).
>
> **Re: Euclidean Gradient Use.**
>
> From a theoretical perspective, the efficacy of the Euclidean gradient depends on the geometric alignment of the reward function with the underlying data manifold. The Jacobian gradient can be viewed as projecting the Euclidean gradient onto the tangent space of the manifold (Proposition 3.1). If the reward is not well aligned with this manifold, the gradient will contain components orthogonal to the data, yielding poor image fidelity (Figures 3 and 11). For many applications, however, such as inverse problems, the reward function has optima on the data manifold. This implies that its gradients automatically force the generated sample onto the manifold even without projection, enabling principled usage of the Euclidean variant. We provide both versions of FMTG, to be selected based on compute budget and reward-manifold alignment.
>
> **Re: Why FMTG outperforms at Lower NFE counts?**
>
> FMTG outperforms at low NFEs for two reasons. First, FMTG enables exact computation of the reward in a single function evaluation via application of $X_{t, 1}$, while most methods leverage a coarse denoiser-based estimate. This degrades guidance quality – particularly early in the trajectory where the denoiser is low-accuracy – necessitating more guidance steps (and hence NFEs) to correct. Second, FMTG operates on a single particle over an ODE, enabling large steps via the pre-trained flow map. By contrast, methods targeting the reward-tilted measure via diffusions often require many integration steps to resolve the Wiener process and many particles to ensure low-bias samples.
>
> To conclude, we have substantially strengthened our experimental evaluation since the original submission. We now evaluate on GenEval [2], a compositional text-to-image benchmark with 553 diverse prompts, guided by four human preference and aesthetic reward models. FMTG-J matches FMTT [1] (a recent SMC-based guidance method) while using 47× fewer NFEs and less than half the VRAM (Supp. Tables 1 and 4). We additionally provide style transfer results (Supp. Table 5) and ablations on all design choices (Supp. Tables 2, 3, 6–8).
>
> Within the broader context, our work offers a novel perspective on guidance for flow-based models, departing from the standard lens of reward tilting. We show that guidance is naturally formulated as deterministic optimal control over the flow map. This perspective sheds light on a line of prior heuristics — DPS and its derivatives (e.g., FlowDPS, FlowChef, MPGD) — revealing that they do not approximate the reward tilt, as originally motivated, but rather approximate our flow-map guidance objective with a coarse single-step denoiser estimate. Beyond unifying these prior methods, our framework predicts that early stopping should recover the correct variance scaling, that the Jacobian is needed when the reward is misaligned with the data manifold, and that the flow map enables dramatically more efficient guidance — all validated empirically (Supp. Tables 2, 3, 6–8). We hope the reviewer finds these contributions and new results compelling, and we would be grateful if they would consider raising their assessment.
>
> [1] Sabour et al., "Test-time scaling of diffusions with flow maps," arXiv:2511.22688, 2025.
>
> [2] Ghosh et al., "GenEval: An Object-Focused Framework for Evaluating Text-to-Image Alignment," NeurIPS 2024.

---

> > ### Author Rebuttal · Reviewer_C5ix · 2026-04-02
> >
> > I thank the authors for addressing my concerns and would encourage the authors to include the clarifications in their response, in the final version. I retain my position that this is a well-written paper with significant impact on flow-matching guidance. I have decided to keep my acceptance score.

---

### Official Review · Reviewer_TF5c · 2026-03-18

**Soundness:** 3
**Presentation:** 2
**Significance:** 2
**Originality:** 2
**Overall Recommendation:** 3
**Confidence:** 3

**Summary:**

This paper studies inference-time guidance for flow-based generative models in the few-step, single-trajectory regime. Instead of viewing guidance as sampling from a reward-tilted distribution, the paper reformulates it as a deterministic optimal control problem that balances reward maximization against deviation from the base flow. Within this framework, the authors show that the flow map naturally appears in the optimal guidance solution, and they argue that several existing flow-guidance methods can be interpreted as coarse approximations that replace the full flow map with a local Euler-step approximation.

Building on this perspective, the paper proposes **Flow Map Trajectory Guidance (FMTG)**, a training-free guidance framework that uses a pretrained flow map both to advance the trajectory and to compute a stronger guidance signal. The paper also provides theoretical analysis of the resulting method, including its connection to the optimal control objective, its behavior in analytically tractable settings, and the effect of practical design choices such as early stopping and backpropagation through the flow map. Empirically, the method is evaluated at text-to-image scale using a FLUX.1-distilled flow map on latent inverse problems and reward-guided image editing tasks, where it achieves competitive or improved performance relative to prior guidance baselines while using only a small number of function evaluations.

**Compliance With Llm Reviewing Policy:**

Affirmed.

**Final Justification:**

I acknowledge the authors’ thoughtful efforts in the rebuttal, which has adequately addressed my minor clarifying questions. However, the core quality limitations that informed my initial review, including limited technical novelty, insufficient incremental contribution to the field, and weak experimental support for the manuscript’s key claims, remain entirely unresolved. Despite more positive ratings from fellow reviewers, based on ICML’s strict acceptance criteria, I still maintain my original weak reject recommendation.

**Key Questions For Authors:**

1. **Theory-to-practice gap for the final algorithm.**
   The main guidance rule is theoretically motivated by the small-\(\lambda\) expansion and the myopic-control interpretation, but the practical algorithm additionally uses multiple inner gradient steps, early stopping, tuned step sizes, and both Jacobian- and Euclidean-gradient variants. Could the authors clarify more explicitly which parts of Algorithm 1 are theoretically justified versus empirically chosen, and provide evidence that the method remains stable and effective outside the small-\(\lambda\) / infinitesimal-window regime? For example, an ablation over \(\lambda\), number of inner steps, and stopping time—together with discussion of failure modes—would help.
   **Why this matters:** If the authors can show that the practical gains are robust and still well aligned with the theory, it would increase my confidence in the paper’s soundness. If instead the method is highly sensitive or only works in a narrow tuned regime, I would view the contribution as less principled than currently presented.

2. **Reproducibility of the flow-map backbone.**
   The experiments appear to rely on a FLUX.1-distilled flow map obtained via private communication. Could the authors clarify exactly what model/checkpoint is used, whether it will be released, and how a reader could reproduce the main results without access to private assets? If release is not possible, it would help to include either a public alternative or a smaller fully reproducible setup demonstrating the same trends.
   **Why this matters:** A clear answer here would substantially affect my evaluation of reproducibility and presentation quality. If the core empirical claims can be reproduced on public assets, my confidence in the work would increase noticeably.

3. **Stronger evaluation for reward-guided generation and editing.**
   The inverse-problem results are convincing, but the reward-guided editing section is comparatively qualitative. Could the authors provide more systematic quantitative evaluation for these tasks—for example, reward improvement, fidelity / reconstruction tradeoff, prompt-wise robustness, seed variance, or human preference results? It would also be useful to report whether FMTG remains consistently better across multiple reward types rather than a few hand-picked examples.
   **Why this matters:** A stronger answer here could raise my assessment of both soundness and significance. Right now, the paper clearly demonstrates value on inverse problems, but the broader claim of being a generally strong guidance framework would be more convincing with deeper evidence on reward-guided tasks.

4. **Actual efficiency beyond NFE counts.**
   A central claim of the paper is that FMTG achieves comparable or better performance with far fewer function evaluations. However, the Jacobian-based variant may incur nontrivial backpropagation and memory overhead. Could the authors report wall-clock runtime and memory usage, ideally at matched quality levels, for FMTG-J, FMTG-E, and the main baselines?
   **Why this matters:** If the low-NFE advantage translates into real end-to-end efficiency, that would strengthen the practical significance of the work. If not, then the paper’s efficiency claims would need to be interpreted more cautiously.

5. **Positioning relative to the closest related guidance methods.**
   The paper makes a compelling case that DPS-like methods can be understood as coarse Euler approximations of the proposed flow-map-centric guidance rule, but the distinctions from other nearby approaches—such as flow-map-based trajectory tilting or seed/noise optimization methods—could be made more explicit. Could the authors clarify both the conceptual and empirical differences to the most closely related prior methods, and explain why FMTG should be viewed as a genuinely new guidance framework rather than a refined implementation of existing ideas?
   **Why this matters:** A precise response would sharpen the paper’s originality and help resolve whether the contribution is mainly a new perspective, a new algorithm, or both. This could affect my final assessment of novelty.

**Limitations:**

No. The paper does not adequately discuss limitations or negative societal impact. Its impact statement is very brief and largely dismisses such concerns, so the authors should add a clearer discussion of practical limitations, failure cases, and potential misuse in image editing/generation.

**Strengths And Weaknesses:**

**Soundness:**
The paper is technically strong overall. Its main methodological contribution is grounded in a clear reformulation of inference-time guidance for flow-based models as a deterministic optimal control problem, rather than approximate sampling from a reward-tilted distribution. This framing is well motivated, and the derivation of the proposed guidance rule from the flow map gives the method a principled foundation rather than presenting it as a purely heuristic update. I also found the theory to be more than decorative: the paper provides a meaningful chain from the control formulation, to the small-\(\lambda\) analysis, to the interpretation of existing methods such as DPS as coarse one-step approximations, and then to practical algorithmic design choices such as early stopping and the Jacobian-vs.-Euclidean gradient tradeoff. The experiments on inverse problems are reasonably convincing and align with the main efficiency claim, especially in the low-NFE regime where the method appears strongest.

That said, there are still some gaps between theory and practice. The strongest theoretical guarantees are derived in the small-guidance or myopic regime, while the practical algorithm includes several additional choices—multiple inner optimization steps, early stopping, step-size tuning, and Euclidean-gradient variants—that are only partially justified by the analysis. This does not invalidate the method, but it does make the final algorithm feel more “theory-inspired” than strictly theory-derived. In addition, although the appendix is substantial, some derivations rely on regularity assumptions and simplifications that may be hard to verify in the large-scale latent image models used in the experiments. On the empirical side, the inverse-problem experiments are solid, but the reward-guided generation section is more qualitative than quantitative, so the evidence for broad superiority across guidance tasks is somewhat less conclusive than the inverse-problem results.

**Presentation:**
The paper is generally well written and has a strong high-level narrative. The central message is easy to follow: if modern generative modeling is moving toward few-step or one-step flow-map-based generation, then guidance should also be reformulated in a way that is native to that setting. This is a compelling and coherent framing, and the paper does a good job of carrying that idea from motivation to theory to experiments. I also appreciated that the paper tries to position itself relative to multiple lines of prior work, including reward-tilting methods, DPS-style methods, and recent flow-map models.

However, I think the paper could still be improved substantially in clarity and reproducibility. Some parts of the theory-to-algorithm transition move quickly, and it is not always fully clear which parts are rigorously justified and which parts are empirical design decisions. The reward-guided generation section is also somewhat compressed and would benefit from more detail about reward definitions, tuning, and evaluation methodology. A more explicit comparison table or paragraph explaining how FMTG differs from closely related flow-based guidance or flow-map guidance methods would also help. Reproducibility is another concern: an important ingredient of the system is a FLUX.1-distilled flow map obtained via private communication, which weakens the paper’s reproducibility story even if the algorithm itself is conceptually reproducible.

**Significance:**
I view the problem tackled by the paper as important and timely. Inference-time guidance is central to making generative models useful in practice, and the field is increasingly interested in reducing generation cost through few-step and one-step models. In that context, a guidance method that is native to flow maps and performs well at very low NFE is a meaningful contribution. Even if the immediate impact is mostly within flow-based image generation, the broader perspective—replacing reward-tilted stochastic sampling views with a deterministic control view in the single-trajectory regime—could influence future work on fast alignment, inverse problems, and efficient controllable generation.

At the same time, the demonstrated impact is somewhat specialized at this stage. The paper focuses primarily on image generation tasks, and the strongest quantitative results are on latent inverse problems. It is plausible that the core ideas will generalize more broadly, but the current submission does not yet establish that. So I would rate the significance as clearly above incremental, but not yet obviously field-defining.

**Originality:**
The paper is original in a way that I think is appropriate for ICML. The novelty is not merely the introduction of another guidance heuristic; rather, it lies in the perspective shift. Recasting guidance for flow models as deterministic optimal control, centering the flow map in the resulting solution, and interpreting existing DPS-style methods as coarse Euler approximations of a more principled flow-map guidance rule are all meaningful conceptual contributions. This is the kind of originality that comes from unifying and clarifying a space, not just from adding another engineering trick.

The main caveat is that some components of the final algorithm—operator splitting, gradient updates, and practical optimization heuristics—are individually familiar, so the empirical method is partly a principled assembly of existing ingredients. Still, I think the paper’s originality comes from how these pieces are derived, organized, and justified through the flow-map-centric control formulation, and that novelty is real.

**Overall assessment:**
Overall, I think the paper’s strengths outweigh its weaknesses. Its biggest strengths are the quality of the problem framing, the principled flow-map-centric derivation, the useful reinterpretation of prior guidance methods, and the strong low-NFE inverse-problem results. Its main weaknesses are the incomplete bridge between theory and the full practical algorithm, limited quantitative depth in the reward-guided generation experiments, and some reproducibility/presentation issues. In summary, this is a strong and interesting paper with real conceptual novelty and promising practical relevance, even if some parts of the empirical and expository story could be strengthened.

---

> ### Author Rebuttal · Authors · 2026-03-31
>
> We appreciate the reviewer's thoughtful feedback. We address each concern below (supplementary tables: https://drive.google.com/file/d/1mcQ1IWwwVpj5ShyqGdcpKvCMxsAoparg/view).
>
> **Theory-to-practice gap.**
>
> We have updated Sections 2 and 3 to reflect that all of our algorithmic choices are both theoretically motivated and empirically validated:
>
> *Multiple gradient steps.* We discretize the guided ODE $\dot{x}\_t = b_t(x_t) + \lambda_t\nabla X_{t,1}(x_t)^\top \nabla r(X_{t,1}(x_t))$ using operator splitting, separating exact integration of the base flow via the flow map from integration of the reward guidance term. When integrating the guidance term, we are free to take multiple smaller timesteps to match the size of the flow map step — a standard technique for improved numerical integration. FMTG achieves strong performance even with n_opt=1, and performance further improves with additional inner steps (Supp. Table 7).
>
> *Early stopping.* As analyzed in Section 2.4, myopic guidance over the entire interval over-optimizes the reward ("reward hacking"). In the Gaussian setting, we prove that early stopping recovers the polynomial scaling of exact optimal control — this is significant because it shows that a simple truncation of the greedy scheme is sufficient to match the variance contraction behavior of the intractable exact solution, avoiding the exponential over-optimization that would otherwise occur. Early stopping additionally allows us to advance the remaining dynamics in a single step via $X_{t_{stop},1}$. Empirically, ES consistently improves FID, SSIM, and LPIPS on inverse problems (Supp. Table 8), and improves accuracy on GenEval (Supp. Table 2).
>
> *Step size.* Different reward functions operate at different scales, thus the guidance strength λ in Eq. (4) is determined by the specific reward function and the desired effect of the guidance on the base flow. We sweep guidance strengths across 3 tasks at 4 NFE levels (Supp. Table 6): we find FMTG dominates and is more robust than FlowDPS and FlowChef.
>
> *Euclidean vs. Jacobian.* Through Proposition 3.1, we characterize the role of the Jacobian as projecting reward gradients onto the data manifold's tangent space, providing insight to when the Euclidean variant suffices (Supp. Table 3; see our response to Reviewer C5ix for details).
>
> **Reproducibility.**
> The flow map checkpoint will be released on HuggingFace upon publication of the work, along with our full code repository to ensure exact reproducibility.
>
> **Stronger reward-guided evaluation.**
> We address this with systematic quantitative evaluation across multiple reward types. On GenEval [2], a compositional benchmark with 553 diverse prompts, we guide generation with an ensemble of human preference and aesthetic reward models (HPSv2 + ImageReward + CLIP + PickScore). FMTG-J matches FMTT [1] with 47× fewer NFEs, and Best-of-$N$ at 13× fewer NFEs (Supp. Table 1). We additionally provide style transfer results (Supp. Table 5) and ablations on all design choices (Supp. Tables 3, 6–8).
>
> **Actual efficiency beyond NFE counts.**
> We report wall-clock time and peak VRAM for all methods in Supp. Table 4. On inverse problems, FMTG-E and FMTG-J have comparable per-step cost to FlowDPS and FlowChef. On reward-guided generation (512px), FMTG-J matches FMTT while being 47× faster in wall-clock and using less than half the VRAM. FMTG-E further reduces VRAM and can be run on a single L40S GPU.
>
> **Positioning.**
> Several existing methods are coarse approximations of FMTG: ReNO corresponds to early stopping at t_stop ≈ 0; FlowDPS and FlowChef approximate FMTG-E with a denoiser estimate (see Euclidean vs. Jacobian above). FMTT targets the reward tilt via SMC with many particles, while FMTG solves a deterministic control problem for an individual particle.
>
> **Limitations.**
> We have expanded the discussion: (1) increased VRAM requirements for the Jacobian variant; (2) myopic guidance can over-optimize without early stopping; (3) potential for misuse in controllable generation.
>
> We note the reviewer's comment that "the paper's strengths outweigh its weaknesses." Since submission, we have significantly improved both experiments and clarity: systematic evaluation on reward-guided generation showing order-of-magnitude efficiency gains over reward-tilting methods (Supp. Table 1), comprehensive ablations on all design choices (Supp. Tables 6–8), and wall-clock/VRAM measurements (Supp. Table 4). We believe these revisions strengthen both the experimental evidence and the clarity of our contribution — offering a deeper understanding of established heuristics such as DPS and demonstrating that this perspective leads to significant practical gains. We would be grateful if the reviewer could consider updating their assessment.
>
> [1] Sabour et al., "Test-time scaling of diffusions with flow maps," arXiv:2511.22688, 2025.
>
> [2] Ghosh et al., "GenEval: An Object-Focused Framework for Evaluating Text-to-Image Alignment," NeurIPS 2024.

---

> > ### Author Rebuttal · Reviewer_TF5c · 2026-04-02
> >
> > Thank you for the detailed rebuttal. My concerns are largely resolved. The authors substantially strengthen the paper by clarifying the theory-to-practice connection, adding quantitative evaluation for reward-guided generation, reporting wall-clock time and VRAM, and sharpening the positioning relative to closely related guidance methods. I also appreciate the clarification on reproducibility, including the commitment to release the flow-map checkpoint and code upon publication. While I still think the final paper would benefit from clearer presentation of some details, the rebuttal addresses my main concerns sufficiently and improves my confidence in both the empirical support and the practical relevance of the work.

---

> > > ### Author Response · Authors · 2026-04-02
> > >
> > > We sincerely thank the reviewer for engaging thoughtfully with our rebuttal and for acknowledging that the concerns have been fully resolved. We are glad that the additional experiments (GenEval evaluation, wall-clock/VRAM measurements, comprehensive ablations) and clarifications on reproducibility, theory-to-practice alignment, and positioning have strengthened the reviewer’s confidence in the work.
> > >
> > > We also appreciate the reviewer’s note that the final paper would benefit from clearer presentation of some details. We fully agree and commit to incorporating these improvements in the camera-ready version, including clearer delineation of theoretically motivated versus empirically chosen design decisions, expanded discussion of the reward-guided experiments, and improved overall exposition.
> > >
> > > We noticed that while the reviewer selected option (a) — “Fully resolved — My concerns have been adequately addressed. If you select this option, please consider adjusting your score accordingly” — the overall recommendation currently remains at 3 (weak reject). We want to gently flag this, as we believe the current score may not reflect the reviewer’s updated assessment. In particular:
> > >
> > > - The original review noted that “the paper’s strengths outweigh its weaknesses,” identifying the problem framing, flow-map-centric derivation, reinterpretation of prior methods, and low-NFE results as key strengths. This language closely matches the definition of 4 (weak accept): “Technically solid paper that advances at least one sub-area of AI, with a contribution that others are likely to build on, but with some weaknesses that limit its impact (e.g., limited evaluation).”
> > >
> > > - The main weaknesses identified (the theory-to-practice gap, limited quantitative evaluation on reward-guided tasks, reproducibility, efficiency beyond NFE counts, and positioning) have each been addressed with new results and clarifications that the reviewer has acknowledged as resolving their concerns. Notably, limited evaluation was the reviewer’s central concern and is explicitly cited in the weak accept definition as a characteristic weakness of that category.
> > >
> > > Since the original assessment already aligned with weak accept language even before the rebuttal, and the identified weaknesses — including the limited evaluation — have now been fully resolved, we respectfully ask whether the reviewer would consider updating the score to 5 (accept) to reflect the reviewer’s own stated assessment. We would be very grateful for the consideration.

---

### Decision · Program_Chairs · 2026-04-30

**Decision:**

Accept (regular)

**Comment:**

This submission introduces Flow Map Trajectory Guidance (FMTG), a strategy for guidance in which the authors compute the gradient of the reward using a flow map lookahead instead of a lookahead based on the posterior mean of the probability path. This approach is natural in the context of recent advances in learning flow maps. Empirical evaluations demonstrate its effectiveness. This idea is accompanied by a framing in terms of a distinction between standard tilting approaches, which the authors frame as a stochastic optimal control, and the proposed approach, which the authors frame as deterministic optimal control. This perspective then informs the small-$\lambda$ expansion for Jacobian variant, which the authors compare to a Euclidean variant that is more heuristic and derives from existing approaches in diffusion.

The reviews for this submission were generally positive. The one reviewer who is more mixed is TF5c. On the one hand this reviewer indicates that the author response addressed some (or possibly all) of their concerns, on the other hand they maintain their score and cite "limited technical novelty, insufficient incremental contribution to the field, and weak experimental support for the manuscript’s key claims". The other reviewers were more positive, but also less detailed.

Given this context, the area chair has read the submission. Overall this paper demonstrates that flow map based guidance does not require SMC (which is very expensive in terms of the number of function evaluations). The area chair agrees with the positive reviewers that this is enough of a contribution to merit acceptance. At the same time the area chair is also inclined to agree with TF5c that there is something of a gap between the generality of the claimed contributions and the specific technical contributions demonstrated.

A specific point that would need to be addressed is the central framing of the paper. The authors write in the introduction that they seek to answer the question

> Is there a principled framework for single-trajectory guidance that leads to high-performing algorithms?

This framing does not hold up; it is not defensible to say that there were previously no single-trajectory guidance strategies, or that all previous methods were not principled, or not performant. A framing that "learned flow maps improve guidance" is perfectly defensible, but the framing as written overstates contributions.

One particular piece of related work that should be cited, and which should reflect positioning is:

> On the Guidance of Flow Matching, Ruiqi Feng, Chenglei Yu, Wenhao Deng, Peiyan Hu, Tailin Wu, ICML 2025

Feng et al. define a general guidance correction for a guided velocity field and then consider a number of approximation strategies. There are no flow maps in this paper, but FMTG is arguably closer to a new instantiation within this framework than a completely new approach (see in particular eq 7, which the authors also relate to DPS).

In short, this submission makes sufficient contributions, but there are some issues with framing. Feng et al. need to be cited in the main text and the relationship of this work to their framework needs to be discussed explicitly. More broadly, the authors should tone down overly broad claims of novelty. If the authors can commit to both, then this submission could appear.